# Uncertainties, sensitivities and robustness of simulated water erosion in an EPIC-based global-gridded crop model

Tony W. Carr[1,*], Juraj Balkovič[2,3], Paul E. Dodds[1], Christian Folberth[2], Emil Fulajtar[4], Rastislav Skalsky[2,5]

[1]University College London, Institute for Sustainable Resources, London, United Kingdom

[2] International Institute for Applied Systems Analysis, Ecosystem Services and Management Program, Laxenburg, Austria

[3] Department of Soil Science, Faculty of Natural Sciences, Comenius University in Bratislava, Bratislava, Slovak Republic

[4] International Atomic Energy Agency, Joint FAO/IAEA Division of Nuclear Techniques in Food and Agriculture, Vienna, Austria

[5] National Agricultural and Food Centre, Soil Science and Conservation Research Institute, Bratislava, Slovak Republic

* Correspondence to: Tony Carr (tony.carr.16@ucl.ac.uk)

**Abstract.** Water erosion on arable land can reduce soil fertility and agricultural productivity. Despite the impact of water erosion on crops, it is typically neglected in global crop yield projections. Furthermore, previous efforts to quantify global water erosion have paid little attention to the effects of field management on the magnitude of water erosion. In this study, we analyse the robustness of simulated water erosion estimates in maize and wheat fields between the years 1980 to 2010 based on daily model outputs from a global gridded version of the Environmental Policy Integrated Climate (EPIC) crop model. Using the MUSS water erosion equation and country-specific and environmental indicators determining different intensities in tillage, residue handling and cover crops, we obtained the global median water erosion rates of 7 t ha$^{-1}$ a$^{-1}$ in maize fields and 5 t ha$^{-1}$ a$^{-1}$ in wheat fields. A comparison of our simulation results with field data demonstrates an overlap of simulated and measured water erosion values for the majority of global cropland. Slope inclination and daily precipitation are key factors in determining the agreement between simulated and measured erosion values and are the most critical input parameters controlling all water erosion equations included in EPIC. The many differences between field management methods worldwide, the varying water erosion estimates from different equations and the complex distribution of cropland in mountainous regions add uncertainty to the simulation results. To reduce the uncertainties in global water erosion estimates, it is necessary to gather more data on global farming techniques, to reduce the uncertainty in global land use maps and to collect more data on soil erosion rates representing the diversity of environmental conditions where crops are grown.

## 1 Introduction

Water erosion is widely recognized as a threat to global agriculture (den Biggelaar et al., 2004; Kaiser, 2004; Panagos et al., 2018; Pimentel, 2006). The removal of topsoil by surface runoff reduces soil fertility and crop

yields due to loss of nutrients, degradation of the soil structure, and decreasing plant-available water capacity (Våje et al., 2005). Water erosion is a natural process, but the impact of agricultural field management on surface cover and roughness is decisive for the magnitude of water erosion. High energy precipitation, steep slopes and lack of vegetation cover intensify water erosion. The most vulnerable areas are mountainous regions, due to steep slopes, the tropics and subtropics, due to abundant high energy precipitation, and arid regions, where precipitation events are rare but often intense and the vegetation cover is sparse. This global distribution of water erosion is indicated by suspended sediment in rivers (Walling and Webb, 1996). South America, Sub-Saharan Africa, South and East Asia have been identified as the most vulnerable regions to erosion on agricultural land by several prior studies (Borrelli et al., 2017; Pimentel et al., 1995).

Despite its importance for global agriculture, water erosion is usually not considered in global gridded crop model (GGCM) studies. Throughout the past decade, GGCMs - typically combinations of agronomic or ecosystem models and global gridded input data infrastructures - have become essential tools for climate change impact assessments, evaluations of agricultural externalities, and as input data providers for agro-economic models (Mueller et al., 2017). Few assessments have considered land degradation processes and found their inclusion and understanding crucial for evaluating climate change mitigation and adaptation strategies (Balkovič et al., 2018; Chappell et al., 2016). Beyond crop models, there is a need to improve the representation of agricultural management and soil-related processes in earth system models to better reflect carbon sinks and sources (Luo et al., 2016; McDermid et al., 2017; Pongratz et al., 2018). Moreover, improving the representation of water erosion in large-scale models is urgently needed to inform major environmental and agricultural policy programs such as the European Union's Common Agricultural Policy (CAP), the United Nations Sustainable Development Goals (SDGs), the United Nations Convention to Combat Desertification (UNCCD) and the Intergovernmental Science-Policy Platform on Biodiversity and Ecosystem Services (IPBES) (Alewell et al., 2019). Yet, the necessary algorithms to simulate water erosion are often not incorporated in such models. Exceptions among field-scale crop models, which are frequently used in GGCM ensemble studies, are the Environmental Policy Integrated Climate model (EPIC) and Agricultural Production Systems Simulator (APSIM). Compared to other commonly used crop models in GGCMs, EPIC stands out in its detailed representation of soil processes including water erosion and the impacts of tillage on soil properties (Folberth et al., 2019).

Recently, water erosion models such as the Universal Soil Loss Equation (USLE) and the Revised Universal Soil Loss Equation (RUSLE) have been used to estimate global water erosion. Annual global soil removal estimates and water erosion rates on cropland of recent studies range between 13 – 22 Gt and 11 - 13 t ha$^{-1}$ (Borrelli et al., 2017; Doetterl et al., 2012; van Oost et al., 2007). USLE and its modifications were developed in the Midwestern United States and should ideally be evaluated against soil erosion measurements when used for other agro-environmental zones (Evans and Boardman, 2016). However, the uneven distribution of field data around the world, the lack of long-term soil measurements in most global regions, and the great variability of the designs of erosion rate measurements hamper the evaluation of global soil loss estimates derived from models (Auerswald et al., 2004; Borrelli et al., 2017; García-Ruiz et al., 2015). In addition, model input data on topography, soil properties and land use are often aggregated over large areas and thus simulation results cannot be directly compared to single field measurements at specific locations.

Most global soil removal estimates using water erosion models are based on static observation approaches or on very coarse timescales that do not fall below annual time steps (Borrelli et al., 2017). Therefore, seasonal patterns of soil cover and precipitation intensities are neglected even though they are crucial factors for water erosion. The state of the soil and its cover is influenced by land management, such as the choice of crops, planting and harvest dates, tillage and plant residue management. Accordingly, neglecting the impact of seasonal changes in vegetation cover and field management practices constitutes large uncertainty in global water erosion estimates. Crop models usually simulate crop growth on a daily timescale, which allows attached water erosion models to account for daily changes in weather, soil properties and vegetation cover. However, uncertainty remains due to the increasing requirement of input data for daily simulations, which is especially challenging at a global scale.

The overall aim of this study is (i) to analyse the robustness of water erosion estimates in all global agro-environmental regions simulated with an EPIC-based global-gridded crop model and (ii) to discuss the main drivers affecting the robustness and the uncertainty of simulated water erosion rates on a global scale. We simulate global water erosion rates in maize and wheat fields using different empirical erosion equations in EPIC while accounting for the daily crop growth and development under different field management scenarios. Here, maize and wheat are used as representative crops of global agriculture, as they are grown under most environmental conditions and represent contrasting soil cover patterns. Our global simulations are carried out for a baseline crop management scenario based on a set of environmental and country-specific assumptions and indicators, which is a common practice in global gridded crop modelling. In addition, we quantify the uncertainties of simulated water erosion values stemming from (i) uncertain field management inputs, and (ii) water erosion calculation methods. We also evaluate the model's sensitivity to all inputs involved in the water erosion calculation to interpret the variability and uncertainties of the simulation results, and to discuss the differences between water erosion equations. Finally, we use field measurements from various locations world-wide to evaluate the robustness of estimated water erosion rates under different environmental conditions.

## 2 Methods

The simplified framework in Figure 1 illustrates the particular stages of the methodological procedure applied by this study and their relationships to input data and model outputs. Both, input and output data are used twofold. We use input data (i) to simulate daily maize and wheat growth and water erosion with EPIC, and (ii) to analyse the sensitivity of relevant model parameters to simulate global water erosion with all equations in EPIC. We use model outputs (i) to calculate a baseline global water erosion scenario, and (ii) to address the uncertainty of simulation results. The final step of this study consists of the robustness check of the model outputs using field data. A detailed description of each element of this study is described in the following sections.

### 2.1 Modelling water erosion and crop growth with EPIC

### 2.1.1 Global gridded crop model and input data

We use a global gridded version of the Environmental Policy Integrated Climate (EPIC) crop model, EPIC-IIASA (Balkovič et al., 2014), to simulate soil sediment loss with runoff from 1980 to 2010 while accounting for the daily growth of maize and wheat under different field management scenarios. EPIC can simulate the growth of a wide range of crops and has a sophisticated representation of carbon, nutrient and water dynamics as well as a

wide variety of possible field management options, including tillage operations and crop rotations (Izaurralde et
al., 2006; Sharpley and Williams, 1990). Originally EPIC was named Erosion-Productivity Impact Calculator and
was developed to determine the relationship between erosion and soil productivity. Due to its origin, EPIC has
several options to calculate water erosion caused by precipitation, runoff and irrigation (Williams, 1990).
EPIC-IIASA requires global soil and topography data and daily weather data. The basic spatial resolution of the
model is 5' x 5' at which soil and topographic data are provided. These are aggregated to homogenous response
units and further intersected with a 30' x 30' climate grid, the resolution at which global gridded climate data are
available. This results in a total of 131,326 grid cells with a spatial resolution ranging between 5' to 30' (about 9
km to 56 km near the equator) (Skalský et al., 2008). We use global daily weather data from the AgMERRA
dataset for the years 1980-2010 (Ruane et al., 2015), soil information from the Harmonized World Soil Database
(FAO/IIASA/ISRIC/ISSCAS/JRC, 2009), and topography from USGS GTOPO30 (USGS, 1997). Each grid cell
is represented by a single field characterized by the combination of topography and soil conditions prevailing in
this landscape unit. Each representative field has a defined slope length (20 – 200 m) and field size (1 - 10 ha)
based on a set of rules for different slope classes (Table S1). The slope of each representative field is determined
by the slope class covering the largest area in each grid cell (Table S1). Slope classes are taken from a global
terrain slope database (IIASA/FAO, 2012) and are based on a high-resolution 90 m SRTM digital elevation model.
In each grid cell, we consider reported growing seasons for maize and wheat (Sacks et al., 2010), and spatially
explicit nitrogen and phosphorus fertilizer application rates (Mueller et al., 2012).
**2.1.2 Water erosion equations**
EPIC includes seven empirical equations to calculate water erosion (Wischmeier and Smith, 1978). The basic
equation is:
$Y = R * K * LS * C * P$       (1)
where Y is soil erosion in t ha$^{-1}$ (mass/area), R is the erosivity factor (erosivity unit/area), K is the soil erodibility
factor in t MJ$^{-1}$ (mass/erosivity unit), LS is the slope length and steepness factor (dimensionless), C is the soil
cover and management factor (dimensionless) and P is the conservation practices factor (dimensionless).
The main difference between the water erosion equations available in EPIC is their energy components used to
calculate the erosivity factor. The USLE, RUSLE and RUSLE2 equations use precipitation intensity as an erosive
energy to calculate the detachment of soil particles. The Modified Universal Soil Loss Equation (MUSLE)
equation and its variations MUST and MUSS use runoff variables to simulate water erosion and sediment yield.
The Onstad-Foster equation (AOF) combines energy through rainfall and runoff (Table 1).
The erosion energy component is calculated as a function of either runoff volume Q (mm), peak runoff rate $q_p$
(mm h$^{-1}$) and watershed area WSA (ha), or via the rainfall erosivity index EI (MJ ha$^{-1}$). The latter determines the
detachment of soil particles through the energy of daily precipitation and a statistical estimate of the daily
maximum intensity of precipitation falling within 30 minutes. RUSLE2 is the only equation calculating soil
deposition. If the sediment load exceeds the transport capacity, determined by a function of flow rate and slope
steepness, soil is deposited, which is calculated by a function of flow rate and particle size (USDA-ARC, 2013).
The soil cover and management factor is updated for every day where runoff occurs using a function of crop
residues, biomass cover and surface roughness. The impact of soil erodibility on simulated water erosion is
calculated for the top-soil layer at the start of each simulation year as a function of sand, silt, clay and organic
carbon content. The topographic factor is calculated as a function of slope length and slope steepness. A detailed
description of the cover and management, soil erodibility and topographic factor is provided in the supporting
information (Text S1). The conservation practice factor is included in all equations as a static coefficient ranging
between 0 and 1, where 0 represents conservation practices that prevent any erosion and 1 represents no
conservation practices. Typical conservation practice factors can be derived from tables, which include values
ranging from 0.01 to 0.35 for terracing strategies and from 0.25 to 0.9 for different contouring practices (Morgan,
2005; Wischmeier and Smith, 1978). Alternatively, values can be derived from local field studies and remote
sensing (Karydas et al., 2009; Panagos et al., 2015), from equations using topographical data (Fu et al., 2005;
Terranova et al., 2009), or from economic indicators (Scherer and Pfister, 2015).

### 2.1.3 Field management scenarios

Field management techniques influencing soil properties and soil cover have a significant impact on the amount
of water erosion. However, these methods are very heterogenous around the world and data on different field
management techniques are sparse. Therefore, three tillage management scenarios – conventional tillage, reduced
tillage and no-tillage – were designed by altering parameters related to water erosion to analyse the impact of field
management on simulated water erosion and to draw conclusions on its impact on the quality of simulation results.
In the reduced and no-tillage scenarios, we decrease soil disturbance by reducing cultivation operations, tillage
depth and surface roughness, and we increase plant residues left in the field after harvest. In addition, we reduce
the runoff curve numbers, which indicate the runoff potential of a hydrological soil group, land use and treatment
class, with decreasing tillage intensification by using pre-defined values for the cover treatment classes presented
in Table 2 (Sharpley and Williams, 1990). By lowering the runoff curve numbers, the impact of reduced tillage
practices on the hydrologic balance can be taken into account (Chung et al., 1999). We simulate each tillage
scenario with and without green fallow cover in between growing seasons, leading to a total of six field
management scenarios.

### 2.2 Baseline scenario for estimating global water erosion in wheat and maize fields

We estimate the rate of water erosion globally by combining these six tillage and cover crop scenarios in different
regions of the world, using climatic and country-specific assumptions and indicators (Table 3). We chose maize
and wheat as two contrasting crop types for analysing water erosion in different cultivation systems. Maize is a
row crop with relatively large areas of bare and unprotected soil between the crop rows. The plant density in wheat
fields is much higher, which improves the protection of soils against water erosion.
We consider conventional and reduced tillage systems globally while considering no-tillage only for countries in
which the share of conservation agriculture is at least 5 %. In tropical regions, we simulate water erosion with a
green cover in between maize and wheat seasons to account for soil cover from a year-round growing season. In
temperate and snow regions, we simulate water erosion affected by both soil cover throughout the year and bare
soil in winter seasons. In arid regions, we do not simulate green cover in between growing seasons due to the
limited water supply.
On slopes steeper than 5 %, we consider only rainfed agriculture, as hilly cropland is irrigated predominantly on
terraces that prevent water runoff. To account for erosion control measures on steep slopes, we use a conservation
P-factor of 0.5 on slopes steeper than 16 %, and a P-factor of 0.15 on slopes steeper than 30 % to simulate
contouring and terracing based on the range of P-values presented by Morgan (2005). The threshold for slopes
that are cultivated with conservation practices is based on the slope classes used for the underlying structure of
slope information of EPIC-IIASA, from which the three highest slope classes (16–30 %, 30–45 %, >45 %) mark
slopes that are less likely to be cultivated without measures to prevent erosion. We choose the MUSS equation
for the baseline scenario as it generates the lowest deviation between simulated and measured water erosion as
discussed below. Table 3 summarises the field management assumptions of the baseline scenario used to aggregate
erosion rates in each grid cell and region.
**2.3 Uncertainty analysis of field management scenarios and water erosion equations**
Given the global scale of the analysis and the aggregated nature of available field management information, there
is much uncertainty about crop management strategies, which introduces uncertainty in the water erosion
estimates. In addition, each water erosion equation gives a different overall erosion estimate. To discuss the
uncertainty of simulation results, we evaluate the variance in simulated water erosion rates at grid level due to: (i)
different management assumptions, and (ii) the choice of water erosion equation. The variance of simulation
outputs is defined as the range between minimum and maximum simulated water erosion rates with all
combinations of tillage and cover crop scenarios and with each water erosion equation.
**2.4 Sensitivity analysis of model parameters**
We use a sensitivity analysis to identify the most essential input parameters to the factors in the seven water
erosion equations. We use the Sobol method (Sobol, 1990), which is a variance-based sensitivity analysis that is
popular in environmental modelling (Nossent et al., 2011). With this method, it is possible to quantify the amount
of variance that each parameter contributes to the total variance of the model output. These amounts are expressed
as sensitivity indices, which rank the importance of each input parameter for simulated water erosion. In addition,
the sensitivity indices can be used to determine the impact of parameter interactions on the model output.
We test 30 parameters directly connected to the water erosion equations in EPIC. In total, we assign 126,976
random values to all input parameters along a pre-defined triangular distribution or a range of discrete values
(Table S2). Water erosion is simulated with EPIC using the seven available equations for each random input
combination at 40 locations where wheat and maize are cultivated. To represent a heterogenous distribution of
global precipitation regimes, we use the natural break optimisation method to choose locations based on average
annual precipitation amounts from 1980 to 2010 (Jenks, 1967). For each location and equation, the most sensitive
parameters are ranked. To analyse the impact of precipitation regimes on the sensitivity of each parameter, we use
Spearman coefficients ($\rho$) to determine if positive or negative relationships exist between each parameter's
sensitivity and annual precipitation.
**2.4 Evaluation of simulated erosion against reported field measurements**
We compare our simulated water erosion rates with 606 soil erosion measurements on arable land from 36
countries representing plot and field scale. Most of the selected erosion rates are based on the [137]Cs method. In
addition, data from erosion plots and volumetric measurements of rills collected by Auerswald et al. (2009),
Benaud et al. (2020) and García-Ruiz et al. (2015) are used. In total, 315 records are derived by the [137]Cs method,
188 records from runoff plots, and 103 records from volumetric measurements of rills. An overview of the field
data is presented in Fig. S4-S7, and the full dataset is available in Table S5.
Guidance on the [137]Cs method is provided by Fulajtar et al. (2017); Mabit et al. (2014) and Zapata (2002). The
[137]Cs radionuclide was released by nuclear weapon tests and from the accident of the Chernobyl Nuclear Power
Plant to the atmosphere and subsequently deposited in the uppermost soil layer by atmospheric fallout. After its
deposition it was bind to soil colloids and can be moved only together with soil particles by mechanical processes
such as soil erosion. Its chemical mobility and uptake by plants is negligible (Mabit et al., 2014; Zapata, 2002). If
part of the topsoil contaminated by [137]Cs is removed by erosion, the [137]Cs concentrations in soil profiles can be
used to trace soil movements using mass balance equation (Walling et al., 2014). A major advantage of the [137]Cs
method is that it provides long term mean erosion rates (representing the period since [137]Cs fallout in the 1960s
until the time of sampling) and overcomes the problem of high temporal variability of erosion.
Bounded plots are the most commonly used method of erosion measurements. They were introduced in the USA
in the 1920s (Hudson, 1993) and were used for the development of USLE and WEPP models (Brazier, 2004).
Eroded soil material can be quantified with erosion plots in different ways (total collection of sediment, fractioned
collection of sediments using multislott divisors, measurement of discharge and sediment concentration by tipping
buckets and Coshocton wheels). The overview of this method is provided by Cerdan et al. (2010); Hudson (1993);
Mutchler et al. (1994); De Ploey and Gabriels (1980) and Zachar (1982).
The volumetric measurements of rill erosion are used since approximately the 1940s in the USA (Kaiser, 1978 in
Evans, 2013) and the 1950s in Europe (Lobotka, 1955), usually at field scale (Boardman, 1990, 2003; Boardman
and Evans, 2020; Brazier, 2004; Evans, 2002, 2013; Herweg, 1988; Zachar, 1982). The volume of erosion rills is
derived from their lengths and profile cross-section areas, which are measured in field or from terrestrial and aerial
photos (Evans, 1986, 1988; Watson and Evans, 1991).
The overwhelming effect of the experimental methodology on measured erosion rates, the lack of sufficient
metadata accompanying erosion measurements and the granular spatial resolution of our simulation setup hinders
a direct comparison between simulated and observed water erosion rates. Instead we compare aggregated
simulated and observed erosion values for different slope and precipitation classes to analyse the robustness of
simulated water erosion rates under different environmental conditions. Therefore, only measurements with
recorded slope steepness and annual precipitation are used. Where annual precipitation is not recorded, it is taken
from the WorldClim2 dataset (Fick and Hijmans, 2017). Due to the non-normal distribution of the simulated and
measured data, the median deviation (MD) is used as a measure to compare the agreement between simulated and
measured water erosion values.

**3 Results**

We estimate global median water erosion rates of 7 t ha$^{-1}$ and 5 t ha$^{-1}$ in maize and wheat fields, respectively. The
total removal of soil in global maize and wheat fields is estimated to be 5.3 Gt a$^{-1}$ and 1.9 Gt a$^{-1}$, respectively. The
map in Figure 2 illustrates the global distribution of simulated water erosion rates. Highest water erosion is
simulated in mountainous regions and regions with strong precipitation, especially in tropical climate zones. In
Asia, those regions are widespread in the east, south-east and the Himalaya region. In Africa, similar areas with

high water erosion values are spread around the continent and are most common at the west coast and in East Africa including broad areas in Guinea, Sierra Leone, Liberia, Ethiopia and Madagascar. In South America, highest water erosion is simulated in the south of Brazil and regions around the Andes mountain range and the Amazon river basin. The highest water erosion values on the American continent are simulated in tropical Central America and the Caribbean. In North America, highest water erosion occurs along the west coast and in the east. Water erosion in Europe is highest in Mediterranean areas and around the Alps.

Median annual water erosion values for the five largest wheat and maize producing countries demonstrate the strong impact of climate and topography on simulated water erosion. In Brazil, China and India, where a large proportion of cropland is in tropical areas, water erosion is relatively high with annual median values of 10 t ha$^{-1}$, 6 t ha$^{-1}$, and 37 t ha$^{-1}$, respectively. In Russia and the United States annual median values are much lower with 1 t ha$^{-1}$, and 2 t ha$^{-1}$, respectively. Overall, Figure 2 illustrates the large variation in simulated water erosion between tropical climate regions and regions with a large proportion of flat and dry land.

**3.1 Sources of model uncertainty related to management assumptions and method selection**

The uncertainty of the simulation results due to management scenarios and the choice of water erosion equations is highest in regions most vulnerable to water erosion (Figure 3). The annual median uncertainty range at each grid cell due to management is 30 t ha$^{-1}$. For 97 % of grid cells, the lowest erosion rates are simulated with management scenarios including no-tillage and cover crops. For 86 % of grid cells, maximum erosion rates are simulated under conventional tillage without cover crops. The annual median uncertainty range at each grid cell due to the choice of erosion equation is 23 t ha$^{-1}$. In 74 % of grid cells, the lowest erosion rates are simulated with the MUSS equation. The highest erosion values are simulated with the RUSLE equation (46 %), followed by the USLE equation (25%).

In most locations, the uncertainty due to field management exceeds the uncertainty caused by choice of erosion equation. For 46 % of grid cells, management scenarios cause the prevailing uncertainty, which we defined as the higher uncertainty range by at least 5 t ha$^{-1}$. The selected erosion equation causes higher uncertainty by at least 5 t ha$^{-1}$ in 14 % of grid cells. The map in Figure 4 illustrates the global distribution of prevailing uncertainty sources.

**3.2 Main drivers of the global erosion model**

We designed the sensitivity study to explain the large variability of simulated water erosion rates in different regions and to discuss the main differences between water erosion equations. Water erosion is highly sensitive to slope steepness (SLP) for all equations. The first-order sensitivity index of the slope parameter indicates that 46–54 % of the variance in the model output is attributable to the slope, without considering interactions between the input parameters (Table 4). Daily precipitation (PRCP) is the second most important parameter for calculating water erosion, with an individual contribution of around 9–20 % to the variance of the output. The remaining parameters contribute together 4–13 % to the output variance.

The first-order sensitivity indices do not include interactions between input parameters, which leads to the sum of all first-order sensitivity indices being lower than 1. The total-order sensitivity indices sum all first-order effects and interactions between parameters, which leads to overlaps in case of interactions and a sum greater than 1. The differences between the first-order and the total-order indices can be used as a measure to determine the impact of the interactions between a specific parameter with other parameters. The total-order sensitivity indices show

that slope steepness, including interactions to other parameters, contributes 63–75 % of the output variance from which 18–21 % are due to interactive effects with other parameters (Table 5). The total-order sensitivity indices from precipitation range from 21–36 %, from which 10–18 % is due to interactions with other parameters.

The high sensitivity of slope and precipitation is similar for all equations, but the most sensitive parameters after these can be different for each equation. Equations estimating erosion energy by surface runoff and the RUSLE2 equation are very sensitive to the hydrological soil group (HSG), which determines the soils infiltration ability. This parameter is used in the calculation of the curve number, which defines the partition of precipitation into runoff and infiltration. Also, the land use number (LUN), which is ranked among the most sensitive input parameters, is used for the calculation of the curve number. The most sensitive parameters of the USLE and RUSLE equation, following slope inclination and daily precipitation, are soil texture classes (SAND & SILT) followed by daily temperature changes (TMX). Crop residues (ORHI) are relatively important for all equations but especially important for equations based on rainfall-energy. Other parameters relevant for field management, such as surface roughness and mixing efficiency of the topsoil, have little influence on water erosion.

The sensitivity of slope steepness has a strong positive correlation with the amount of annual precipitation at each location ($\rho = 0.69$, $p<0.01$). The increase in the sensitivity of slope steepness with increasing annual precipitation is demonstrated in Figure 5, which illustrates substantially lower sensitivity indices at dry locations compared to wet locations. In contrast, the sensitivity indices of daily precipitation are negatively correlated to annual precipitation with a moderate strength ($\rho = 0.45$, $p<0.05$). Depending on the equation, strong positive or negative correlations between SIs and annual precipitation also exist for other parameters such as slope length, soil texture, soil organic carbon, channel length, channel slope and watershed area (Table S4).

### 3.3 Evaluation of simulation results against field data

The most recent estimated global water erosion rates on cropland of 11 - 13 t ha$^{-1}$ derived from a comparable method (Borrelli et al., 2017; Doetterl et al., 2012; van Oost et al., 2007) lie above our simulated median water erosion rates of 7 t ha$^{-1}$ and 5 t ha$^{-1}$ for maize and wheat fields, respectively. Similarly, our global water erosion estimates in maize and wheat fields are lower than the median value of 9 t ha$^{-1}$ from 606 water erosion measurements from cropland around the world.

To evaluate the agreement between simulated and observed data, we compare median values between simulated and measured erosion rates grouped by precipitation and slope classes, which are defined along the whole range of recorded slope inclinations and annual precipitation amounts of the field data (Figure 6a). Although slope and precipitation classes from the field are spread unevenly, they cover most climatic and topographic characteristics relevant to global agriculture. The comparison illustrates that the deviation between simulated and field data is highest for locations with steep slopes and high annual precipitation. Where slopes are steeper than 8 % and annual precipitation is higher than 1000 mm, the median of simulated water erosion exceeds the median of measured water erosion in most cases by at least 50 t ha$^{-1}$. With decreasing slope steepness and annual precipitation, the median deviation between simulated and measured data is decreasing. Where both slope steepness is below 8 % and annual precipitation is below 1000 mm, the median deviation is lower than 5 t ha$^{1}$ in most cases. A comparison of measured and simulated water erosion using other equations with the baseline scenario can be found in Fig. S8.

The boxplots in Figure 6b illustrate the range of water erosion values measured in the field and simulated with the
baseline scenario. The high deviation between observed and simulated values for grouped locations with slopes
steeper than 8 % and annual precipitation higher than 1000 mm can also be observed between the range of
simulated and measured water erosion values. Outside locations combining steep slopes and strong precipitation,
median deviation between simulated and measured data is lower than the variability within the field data. The
range of values at locations with lower precipitation and slope steepness demonstrates that simulated values are
mostly below measured values in those environments.
The uncertainty in the choice of management scenarios and water erosion equations included in our baseline
scenario leads to an uncertainty of the deviation between simulated and measured erosion values. This uncertainty
is demonstrated in Figure 6b by additional three bars illustrating the range of simulated medians due to contrasting
tillage management scenarios, cover crop scenarios and different water erosion equations. At locations with low
to moderate slope steepness and annual precipitation, the measured water erosion values agree best with the
simulation values generated under scenarios implying larger water erosion, such as high intensity tillage and low
soil cover. On the other hand, at locations with steep slopes and intensive precipitation, the measured values are
closer to the simulated values under scenarios with less intensive tillage and more soil cover. In addition, the
varying sensitivities of each water erosion equation lead to a different magnitude of water erosion values in
different environments. On low to moderate slopes, water erosion simulated with the MUSS equation is lowest,
whereas RUSLE generates the highest values. On steep slopes, the RUSLE equation generates the lowest water
erosion values, which agree best with the measured values. The options to increase and decrease simulated water
erosion with different field management scenarios and water erosion equations creates both uncertainty in the
model results, but also the possibility to closely match field data.
At locations combining steep slopes and intense precipitation, most management scenarios and equations generate
water erosion values that are higher than the measured values. However, those environmental conditions cover
only a small share of global cropland. Cultivation areas with slopes steeper than 8 % and annual precipitation
higher than 1000 mm represent only 7 % of global maize and wheat cropland in our grid cells. The map in Figure
7 illustrates that the highest concentration of these areas is in East and South-East Asia, followed by Central and
South America, and Sub-Saharan Africa.

## 4 Discussion

### 4.1 Varying robustness of simulated water erosion in different global regions

Global water erosion estimates generated with an EPIC-based GGCM and our baseline scenario overlap with
observed water erosion values under most of the climatic and topographic environments where maize and wheat
are grown. However, global maize and wheat land include locations where environmental characteristics differ
significantly from the Midwestern United States, where the data was collected to develop the water erosion
equations embedded in EPIC. The USLE model and its modification were developed with data for slopes of up to
20 %, which makes model application for steeper slopes uncertain (McCool et al., 1989; Meyer, 1984).
Furthermore, the relations between kinetic energy and rainfall energy in the American Great Plains differ from
other regions in the world (Roose, 1996). Similarly, the runoff curve number method, which is the key

methodology for the calculation of surface runoff, is based on an empirical analysis in watersheds located in the United States and might be less reliable in different regions of the world (Rallison, 1980). Due to the high sensitivity of slope steepness and daily precipitation for the calculation of water erosion, the reliability of the tested equations decreases in regions where typical slope and precipitation patterns differ from the Midwestern US. Although some studies have successfully used USLE and its modification under a different environmental context (e.g. Alewell et al., 2019; Almas and Jamal, 2009; Fischer et al., 2018; Sadeghi and Mizuyama, 2007), many studies have concluded that the accuracy of these models may be reduced outside the environments they were created without calibration and model adaptation (e.g. Cohen et al., 2005; Labrière et al., 2015).

The skewed distribution of simulated water erosion values influenced by extreme soil loss rates in few fields highly sensitive to water erosion results in a large difference between the global median value of 6 t ha$^{-1}$ a$^{-1}$ and the global average value of 19 t ha$^{-1}$ a$^{-1}$ (Fig. S9). Due to the strong influence of outliers on average values, we used median values to represent global and regional water erosion rates in wheat and maize fields. The high sensitivity of the simulation results to slope inclinations and precipitation suggests that a significant share of the estimated soil removal of 7.2 Gt a$^{-1}$ originates from small wheat and maize cultivation areas on steep slopes with strong annual precipitation.

**4.2 Sources of uncertainties in global water erosion estimates**

**4.2.1 Uncertain land use in mountainous regions**

Changing climatic conditions with increasing elevation and the variable soils in mountainous regions can favour crop cultivation in higher elevations over lower elevations (Romeo et al., 2015). However, upland farming without soil conservation measures can lead to exhaustive soil erosion and can become a critical problem for agriculture (Montgomery, 2007). Large areas of land have been abandoned due to high erosion rates as soils were no longer able to support crops (Figure *8*) (Romeo et al., 2015). As mountain agriculture is determined by various environmental and socio-economic factors, the cultivation of steep slopes can be very variable between regions. Regional erosion assessments in mountainous cropland suggested that areas with extreme water erosion rates are mainly limited to marginal steep land cultivated by smallholders (Haile and Fetene, 2012; Long et al., 2006; Nyssen et al., 2019). In some mountainous regions, efforts to remove marginal farmlands from agricultural production, and programs to improve land management on steep slopes have reduced high water erosion rates (Deng et al., 2012; Nyssen et al., 2015). On the contrary, recent pressure through increasing population and crop production demands has resulted in re-cultivation of hillslopes and a reduction of fallow periods, which limits the recovery of eroded soil (Turkelboom et al., 2008; Valentin et al., 2008).

To analyse the sustainability of simulated maize and wheat cultivation systems exposed to high erosion rates, we compare simulated annual eroded soil depth with a global dataset on modelled sedimentary deposit thickness (Pelletier et al., 2016). The comparison shows that at 4 % of grid cells permanent maize and wheat cultivation would not be sustainable as the whole soil profile would be eroded at the end of the simulation period (Fig. S18). Most of the unsustainable agriculture is simulated on steep slopes. Although we account for conservation techniques and cover crops, we do not imitate the highly complex farming practices involving intercropping techniques and fallow periods, which are common on hillslopes typically managed by smallholders (Turkelboom et al., 2008). Moreover, we assume that the slope class representing the largest area in each grid cell most likely represents the largest share of arable land. This builds on the idea that a spatially extensive and diverse landscape

can be represented by a single "representative field" characterized by the prevailing topography and soil conditions found in the landscape. On hilly terrain this setup simulates maize and wheat cultivation on steep slopes and thus mainly represents unsustainable agriculture. Although unsustainable maize and wheat cultivation can be observed in several mountain regions, cropland is very heterogeneously distributed in mountains and thus erosion rates from one representative field are highly uncertain.

The uncertainty in cropland distribution can partly be reduced by developing a higher resolution global gridded data infrastructure, which is currently not available for EPIC-IIASA. However, due to the large uncertainty in global land cover maps (Fritz et al., 2015; Lesiv et al., 2019), an explicit spatial link between cropland distribution and the corresponding slope category cannot be established without on-site observations. We test the impact of this uncertainty for erosion estimates in Italy, where large maize and wheat cultivation areas are distributed on both flat terrain in the north and mountainous regions in the south. In an ideal scenario where cropland is limited to flattest land available per grid cell, median simulated water erosion in Italy would be reduced to tolerable levels below 1 t ha$^{-1}$. However, in a scenario, where the most common slopes per grid cell are cultivated, median simulated water erosion increases to 14 t ha$^{-1}$ due to high water erosion simulated in Italy's mountainous regions (Fig. S19). This suggests a high uncertainty in global water erosion estimates due to uncertain spatial links between maize and wheat cultivation areas and different slope categories.

### 4.2.2 Uncertain field management

Simulated water erosion values are highly variable depending on the field management scenario. Simulating cover crop and no-tillage worldwide results in the lowest global soil removal of 2 Gt a$^{-1}$ with median water erosion rates of 1 t ha$^{-1}$ a$^{-1}$ and simulating no cover crops and conventional tillage worldwide results in the highest global soil removal of 13 Gt a$^{-1}$ with median water erosion rates of 17 t ha$^{-1}$ a$^{-1}$. These variations cause further uncertainties in the simulation results.

Indeed, a proper reconstruction of a business-as-usual field management is important to further narrow down the uncertainty in global crop modelling (Folberth et al., 2019). In this study we allocated prevailing field management using a set of environmental- and country-specific indicators, similarly to Porwollik et al. (2019). For example, we accounted for conservation agriculture only in countries where this management strategy is likely according to AQUASTAT (FAO, 2016). Furthermore, by assuming cover crops in between wheat and maize seasons we simulated more complex cropping systems in the tropics, where long and year-round growing seasons and frequent multi-cropping farm practices barely leave the soil uncovered. Hence, we did not simulate bare fallow in the tropics as erroneously high water erosion values would have been simulated at locations with heavy precipitation falling on bare soil. In addition, conservation practices such as contouring and terracing are crucial to reduce the simulation of high water erosion values on steep slopes. We simulated these practices for specific slope classes under the assumption that farmers around the world uniformly use conservation practices when cultivating on steep slopes. The most relevant parameters used for tillage scenarios are related to crop residues left in the field. In addition, equations directly connected to surface runoff are strongly influenced by the land use number used to determine the impact of cover type and treatment on soil permeability. While both crop residues and green fallow decrease water erosion significantly, especially in the tropics, their use varies widely between regions and even farms, based on a complex web of factors such as institutional factors, farm sizes, risk attitudes, interest rates, access to markets, farming systems, resource endowments, and farm management skills (Pannell et al., 2014).

Also, soil conservation measures such as terraces or contour farming significantly influence water erosion but are
very heterogeneously used between regions, farming systems and farmers. Our baseline scenario is a very rough
depiction of the complex patterns of field management around the world but attempts to represent these highly
influential practices with the limited available data.

### 4.2.3 Variable estimates from different water erosion equations

The water erosion equation chosen for the baseline scenario generates the lowest global soil removal estimate.
Different water erosion equations embedded in EPIC estimate a higher global soil removal of up to 11 Gt $a^{-1}$ as
well as higher median water erosion rates up to 19 t $ha^{-1}$ $a^{-1}$. The MUSS water erosion equation chosen for the
baseline scenario generates water erosion rates closest to the field data. The focus of equations on either rainfall
energy or runoff energy is relevant for the different simulation results under specific environmental conditions.
Equations based on rainfall-energy such as RUSLE and USLE simulate higher water erosion values than the other
equations at most locations. However, on steep slopes they generate the lowest water erosion values as runoff
becomes a greater source of energy than rain with increasing slope steepness (Roose, 1996). Also, the varying
sensitivities of other parameters to the equations such as soil properties and management parameters lead to a
varying agreement between simulated data and field data depending on the equation selection. Detailed field data
would facilitate the choice of an appropriate equation to simulate water erosion worldwide or for a specific region.

### 4.3 The difficulty of evaluating large-scale erosion estimates with field data

The selection of field data for evaluating simulated water erosion was limited by the low availability of suitable
water erosion observations covering the entire globe. The lack of reliable data on water erosion rates is a severe
obstacle for understanding erosion, developing and validating models and implementing soil conservation
(Boardman, 2006; Nearing et al., 2000; Poesen et al., 2003; Trimble and Crosson, 2000). The main reasons for
the low availability of suitable data to evaluate simulated water erosion rates are twofold: (i) erosion monitoring
is expensive, time consuming and labour demanding; and, (ii) primary data and metadata of measurement sites
accompanying final results are often not available and many older measurements are poorly accessible as they are
not available online (Benaud et al., 2020). A variety of factors influencing water erosion such as climate, field
topography, soil properties and field management need to be considered when modelling water erosion but are
often not reported in available field measurements (García-Ruiz et al., 2015). This hampers a direct comparison
between simulated and observed water erosion values. We demonstrated the varying match between measured
and simulated water erosion using different tillage and cover crop scenarios. Metadata on field management often
only provides the crop cultivated and therefore the conditions under which erosion was measured in the field are
not known sufficiently to evaluate erosion values simulated under different field management scenarios. Similarly,
information on field topography and soil properties is often not provided with recorded field measurements and
thus their use is limited in an evaluation of water erosion estimates simulated in different global environments.
Moreover, most data are concentrated in the United States, West Europe and the West Mediterranean (García-
Ruiz et al., 2015). In summary, there is a lack of field data representing all needed regions, situations and scenarios
(Alewell et al., 2019).
The appropriate selection of field data to evaluate model outputs needs to be considered as well. At different
spatial scales different erosion processes are dominant and consequently different erosion measurement methods
are suitable (Boix-Fayos et al., 2006; Stroosnijder, 2005). Most authors use very heterogeneous data sets to
evaluate their models, involving data generated by different methods at variable time and spatial scales and
variable quality. For example, Doetterl et al. (2012) used plot data, suspended sediments from rivers, and data
from RUSLE modelling. Borrelli et al. (2017) used soil erosion rates (measurement methods are not specified),
remote sensing, vegetation index (NDVI) and results of RUSLE modelling. In his review on erosion rates under
different land use, Montgomery (2007) used field data derived from erosion plots, field-scale measurements,
catchment-scale measurements using hydrological methods, $^{137}$Cs-method, soil profile truncation and elevated
cemetery plots.
Whilst all erosion measurement methods are open to criticism, we decided to use only data obtained by field
measurements from runoff plots, by $^{137}$Cs method and volumetric surveys as these methods are most suitable at
plot, slope and field scale. Geodetic methods such as erosion pins and laser scanner are also used at plot to field
scales, but their accuracy is much lower than the accuracy of plot measurements and $^{137}$Cs method. Furthermore,
erosion pins are mainly suitable for areas with extreme erosion rates (Hsieh et al., 2009; Hudson, 1993), and laser
scanners have difficulties to recognize vegetation (Hsieh et al., 2009). Other commonly used methods such as
hydrological method (measurements of discharge and suspended sediment load) and bathymetric method are more
suitable for larger scales and involve a significant portion of channel erosion, which is not related with agricultural
land (García-Ruiz et al., 2015). We did not consider plot experiments using rainfall simulators as they are usually
performed on small plots with artificially generated rainfalls, which mostly have very low energies and thus
generate low erosion rates (Boix-Fayos et al., 2006; García-Ruiz et al., 2015).
The $^{137}$Cs method was criticised by Parsons and Foster (2013), who questioned assumptions about the $^{137}$Cs
behaviour in the environment (variability of the $^{137}$Cs input by wet fallout, its microspatial variability at reference
sites, its possible mobility in certain soils, the $^{137}$Cs uptake by plants and other aspects of $^{137}$Cs behaviour in soil).
To confront the criticism against the $^{137}$Cs method, Mabit et al. (2013) discussed all objections raised by Parsons
and Foster (2013) and confirmed its accuracy by listing several studies, in which $^{137}$Cs based erosion rates are
compared with erosion rates derived from direct measurements. The $^{137}$Cs method is based on a set of
presumptions which should be met to produce useful results and thus careful interpretation of the obtained results
is needed (Fulajtar et al., 2017; Mabit et al., 2014; Zapata, 2002).
Similarly, erosion rates obtained by volumetric measurements require careful interpretation as they are exposed
to various potential sources of errors and do not account for interill erosion. Although the latter can be neglected
under certain circumstances, studies from Europe and semiarid areas of the USA have reported that interill erosion
contributed significantly to the amount of soil eroded in fields (Boardman and Evans, 2020; Parsons, 2019).
Further, measuring the lengths and cross-sections of rills during field surveys or on terrestrial and aerial photos
can be very subjective (Panagos et al., 2016). Different approaches used to detect and measure rills in fields can
cause variability in calculated erosion volumes up to a factor of two (Boardman and Evans, 2020; Casali et al.,
2006; Watson and Evans, 1991). In order to obtain soil erosion rates in weight units, soil volumes need to be
converted using the soil bulk density, which is often based on estimates (Evans and Brazier, 2005).
The shortcomings of erosion plot measurements were discussed by several authors (Auerswald et al., 2009;
Brazier, 2004; Evans, 1995, 2002; Loughran et al., 1988). Erosion plots have various sizes and shapes (few meters
to few hundreds of meters) and various approaches of sediment recording are used (total collection, multislot

divisors, tipping buckets, Coshocton wheels), which all involve significant uncertainties. Although some long-term plot experiments exist, many plot measurements fail to cover the whole year erosion cycle (Auerswald et al., 2009). Often, they have to be removed during land management operations such as seeding, ploughing, or they are too expensive and labour demanding.

Despite all the shortcomings of available soil erosion data, most data provide valuable information (Benaud et al., 2020). The evaluation against field measurements in this study provided a first indication of the robustness of results under specific topographic and climatic conditions. In most environments relevant for maize and wheat cultivation the deviation between simulated and measured water erosion values is lower than the variability within the field data. The reported data does not enable us to further narrow down the uncertainties addressed. Although the metadata accompanying the field measurements includes information on slope steepness and annual precipitation (or geographic coordinates allowing for overlay with climatic data), information on soil types or texture classes, crop type and tillage system implemented over time are provided only for few points. Also, the various methods used to measure erosion rates, their complex implementation and the bias of field studies towards locations sensitive to erosion lead to an uncertain representation of large-scale erosion rates based on field measurements. To facilitate in-depth evaluation of erosion models across different scales, it is crucial to provide detailed information on site characteristics and to harmonise approaches to measure erosion in the field. Moreover, the accessibility of field data should be improved as raw data is often not published or needs to be collected from numerous publications, grey literature and conference proceedings to obtain the large amount of data necessary for regional or global erosion studies. Therefore, we support recent efforts to collate erosion measurements and metadata from existing studies (Benaud et al., 2020) as we believe that the availability of field data through a single platform will greatly benefit future modelling studies and the understanding of soil erosion at all scales.

**5 Conclusion**

The simulation of water erosion with GGCMs is largely influenced by the resolution of global datasets providing topographic, soil, climate, land use and field management data, which is currently not available at the field scale. Yet, considering water erosion in global crop yield projections can provide useful outputs to inform assessments of the potential impacts of erosion on global food production and to identify soil erosion hotspots on cropland for management and policy interventions. To improve the quality of the estimates and to further develop these models, it is crucial to identify, communicate and address the existing uncertainties. Increasing the resolution of global soil, topographic and precipitation data is central for improving global water erosion estimates. In addition, this study provides an insight into the importance of considering field management. The numerous options to simulate the cultivation of fields result in a large range of possible water erosion values, which can only partly be narrowed down at a global scale. Further improvement of global water erosion estimates requires detailed and harmonized field measurements across all environmental conditions to validate and calibrate simulation outputs. Using existing field data, we were able to identify specific environmental characteristics for which we have lower confidence in the modelled erosion rates. These are mainly found in the tropics and mountainous regions due to the high sensitivity of simulated water erosion to slope steepness and precipitation strength, and the complexity of mountain agriculture. However, these areas represent only a small fraction of global cropland for maize and wheat. The overlap of simulated and measured water erosion values in most environments used to produce maize

and wheat underlines the robustness of an EPIC-based GGCM to simulate the differences in water erosion rates of major global crop production regions.

**Data availability**. Additional information on model outputs, methods, study design and field data are available in the supporting information file: TWCarr-si.zip.

**Author contributions.** TC, JB, CF and RS designed the study. TC, JB, CF, EF and RS collected and analysed the data. TC prepared the manuscript with contributions from all co-authors.

**Competing interests**. The authors declare that they have no conflict of interest.

**Acknowledgement**. This project has received funding from the Grantham Foundation and the European Union's Horizon 2020 research and innovation programme under grant agreement No 776810 (VERIFY) and No 774378 (CIRCASA). We would like to thank three anonymous reviewers for their help to improve this paper.

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

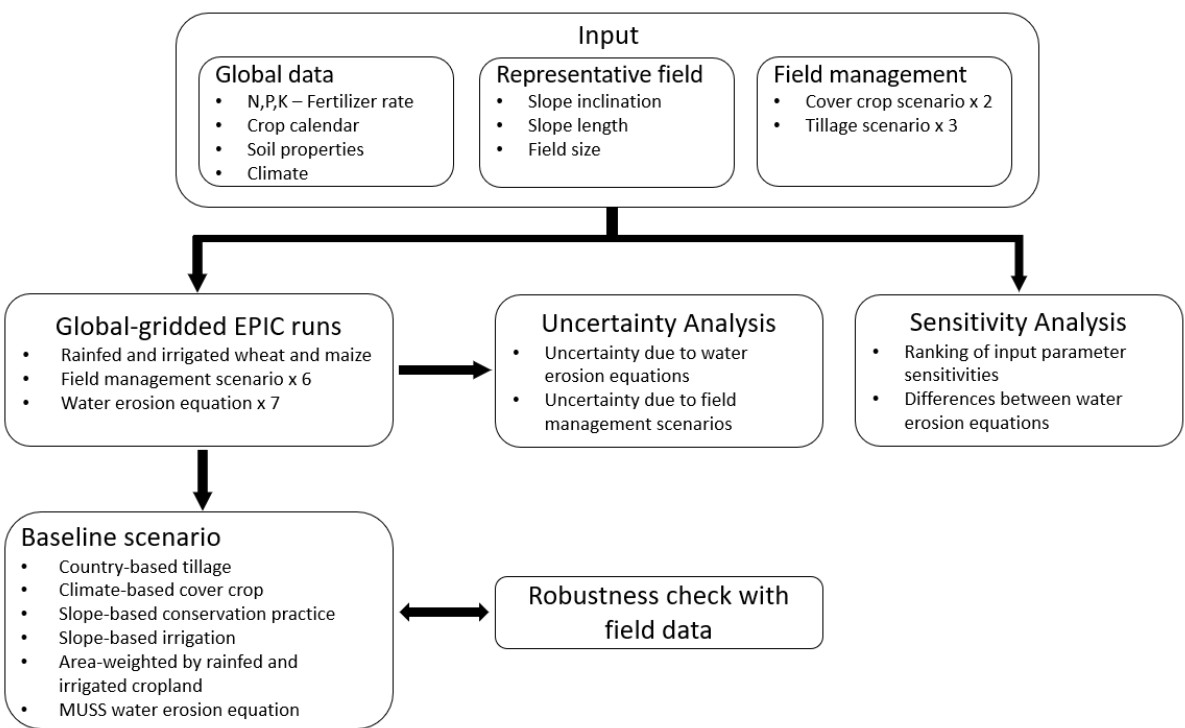


Figure 1: Scheme of procedure used for simulating global water erosion with EPIC-IIASA and for analysing the
uncertainty, sensitivity and robustness of our simulation setup.

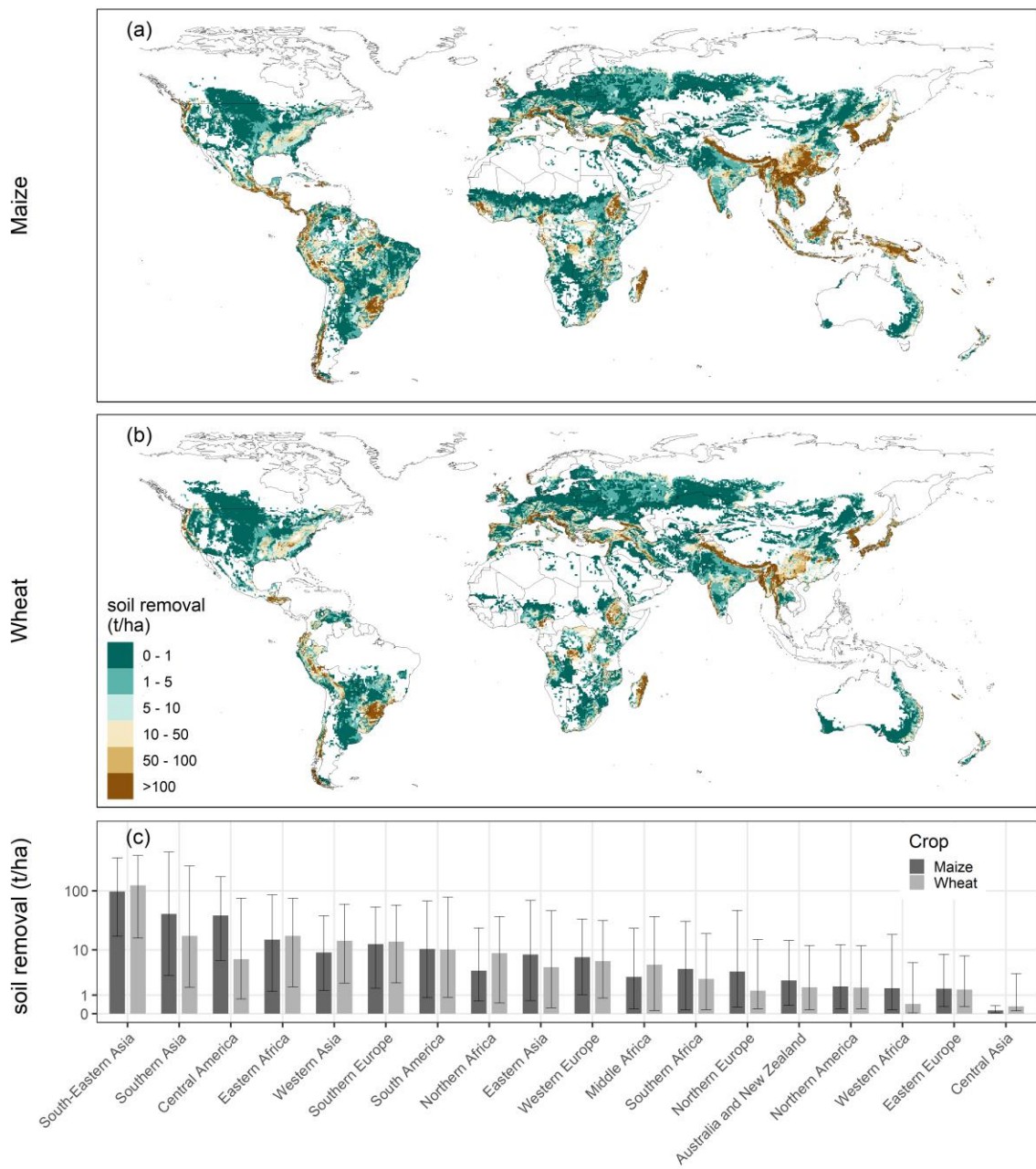


Figure 2: Soil loss due to water erosion in maize (a) and wheat (b) fields simulated with the baseline scenario.
Each pixel cell illustrates the median relative water erosion of one representative field. The extent of cropland
areas is not considered in pixel cell size. The bars in the bottom plot (c) illustrate median soil removal for major
world regions simulated under maize and wheat cultivation. The lines and whiskers illustrate 25th and 75th
percentile values. The classification of world regions is illustrated in Fig. S3. Due to the large gap between
aggregated values, all values in the bottom plot have been log-transformed to facilitate the visual comparison.

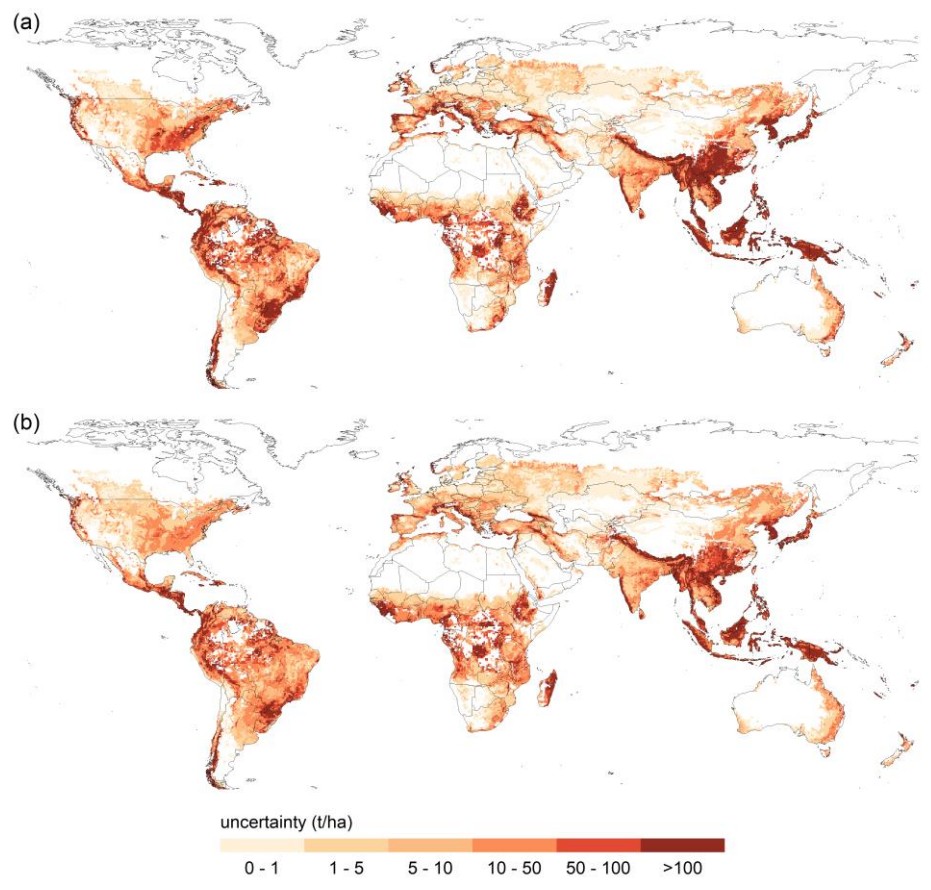


Figure 3: Water erosion uncertainty due to (a) field management assumptions and (b) water erosion equations.

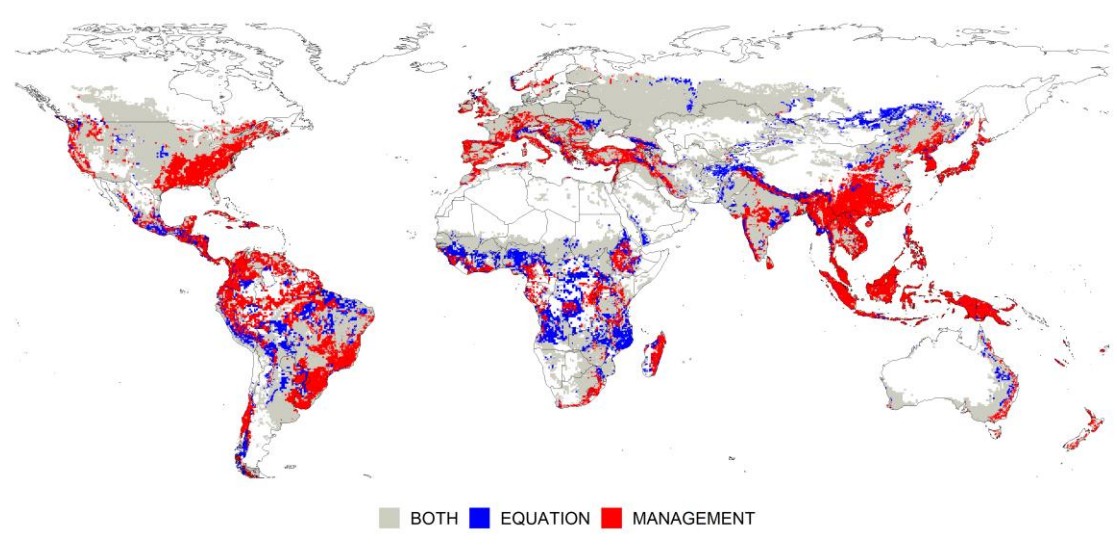


Figure 4: Prevailing uncertainty, defined as the higher uncertainty range by at least 5 t ha[-1].

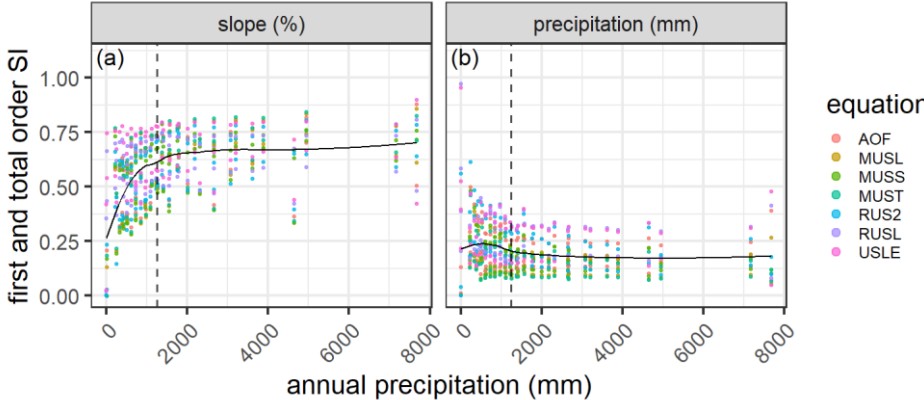


Figure 5: First-order and total-order sensitivity indices (SI) for (a) slope steepness (%) and (b) precipitation [mm]. The dashed vertical line illustrates median annual precipitation at all tested locations (1248 mm).

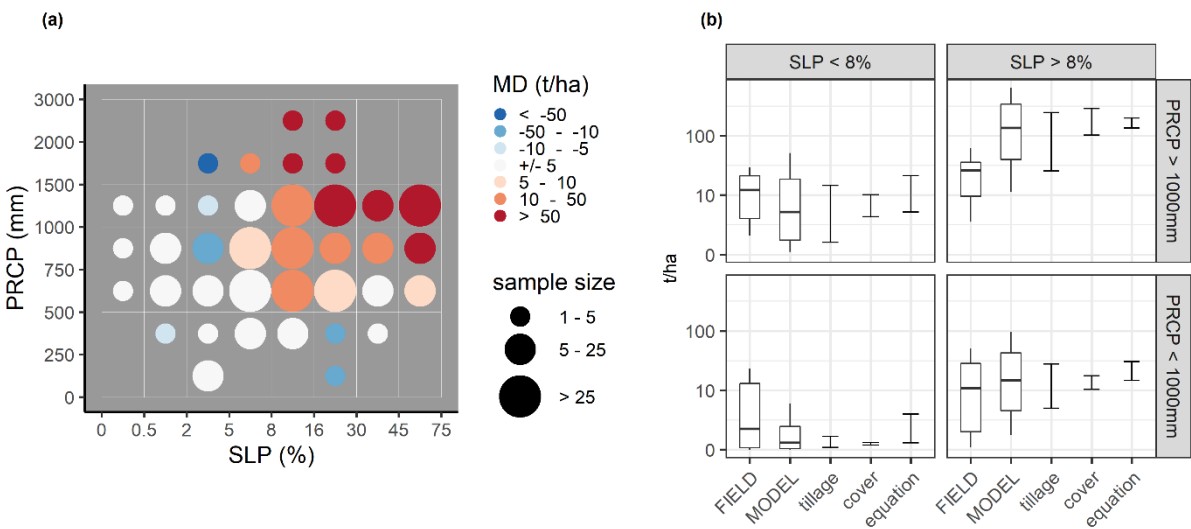


Figure 6: Comparison of simulated erosion with measured erosion. (a) Median deviation (MD) in t ha[1] between simulated erosion using the baseline scenario and measured erosion. Simulated and measured data is grouped into precipitation classes and slope classes used for the simulation setup. (b) Distributions of measured erosion rates, erosion rates simulated with the baseline scenario and uncertainty ranges for management assumptions and erosion equations. The boxplots are defined by the median, the 25[th] and the 75[th] percentile of simulated and measured erosion rates. Whiskers illustrate the 10[th] and 90[th] percentile. The three bars next to the boxplots illustrate minimum and maximum median erosion rates calculated with all tillage and cover crop scenarios and with all water erosion equations. The values have been log-transformed for better visualization.

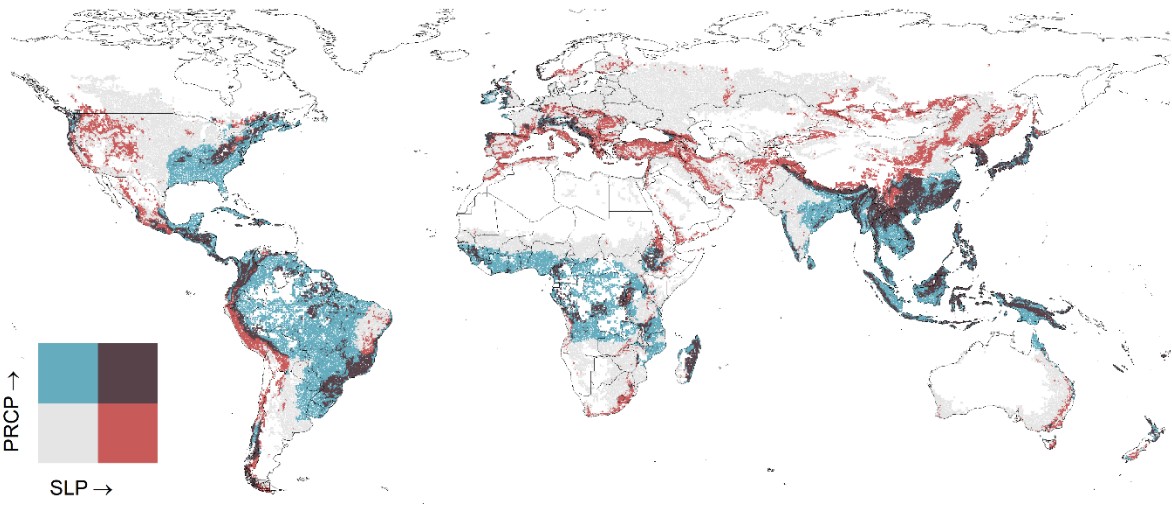

914

Figure 7: Distribution of low to high slope steepness (SLP) and annual precipitation (PRCP) in maize and wheat
fields. Dark areas illustrate grid cells where dominant slopes are steeper than 8 % and annual precipitation is above
1000 mm. Correspondingly, blue, red, and grey pixels are below one or both thresholds.

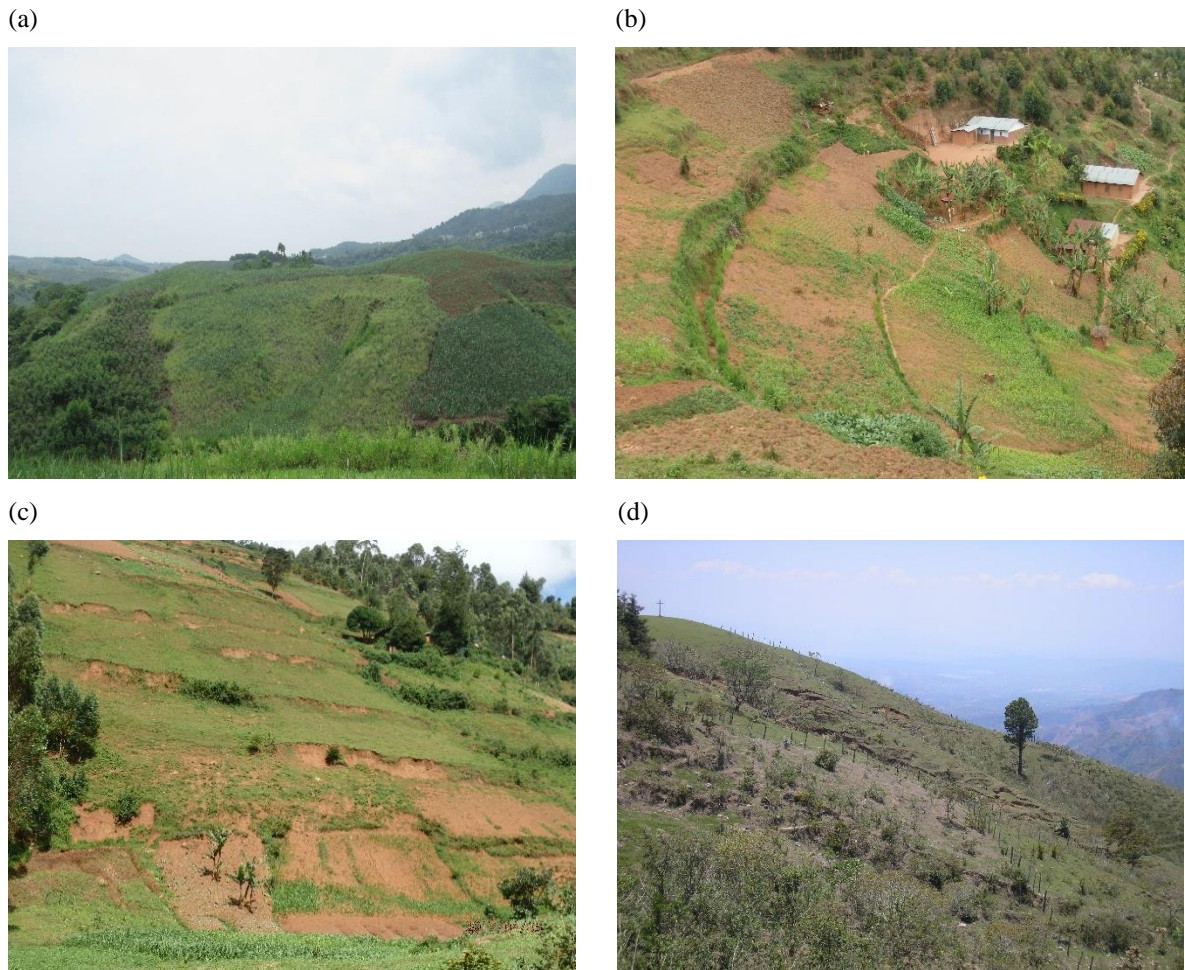

Figure 8: (a) Sugar cane cultivation on steep slopes in South China (Nanning, Guangxi Zhuang Autonomous
Region). The steepest slopes are already abandoned and reforested by eucalyptus trees. (b) Maize cultivation on
strongly eroded slopes (30 – 60 %) in South West Uganda (Kigwa, Kabale District). (c) Abandoned fields and
maize cultivation on a steep slope (30 – 60 %) in South West Uganda (Kigwa, Kabale District). (d) Degraded

and abandoned maize fields on steep slopes (20 – 60 %) in Northern El Salvador (San Ignacio, Chalatenango
Department). The photos and additional examples are provided in Fig. S10 – S17.

Table 1: Equations for calculating the erosivity factor in each water erosion equation available in EPIC.

| Erosivity factor | | Equation |
|---|---|---|
| R = EI | (2) | USLE, RUSLE, RUSLE2 (Renard et al., 1997; USDA-ARC, 2013; Wischmeier and Smith, 1978) |
| $R = 0.646 * EI + 0.45 * (Q * q_p)^{0.33}$ | (3) | AOF (Onstad and Foster, 1975) |
| $R = 1.586 * (Q * q_p)^{0.56} * WSA^{0.12}$ | (4) | MUSLE (Williams 1975) |
| $R = 2.5 * Q * q_p^{0.5}$ | (5) | MUST (Williams, 1995) |
| $R = 0.79 * (Q * q_p)^{0.65} * WSA^{0.009}$ | (6) | MUSS (Williams, 1995) |


Table 2: Tillage management scenarios for maize and wheat cultivation

| | Conventional tillage | Reduced tillage | No-tillage |
|---|---|---|---|
| total cultivation operations | 6 – 7 | 4 – 5 | 3 |
| max. surface roughness | 30 – 50 mm | 20 mm | 10 mm |
| max. tillage depth | 150 mm | 150 mm | 40 – 60 mm |
| plant residues left | 25 % | 50 % | 75 % |
| cover treatment class | straight | contoured | contoured & terraced |


Table 3: Management assumptions and erosion equation selected for the baseline scenario

| Option | Baseline | | | |
|---|---|---|---|---|
| TILLAGE | • Mix of conventional, reduced and no-tillage in regions where the national share of conservation agriculture is > 5 % according to the latest reported data in AQUASTAT (2007-2014) (FAO, 2016): Argentina, Australia, Bolivia, Brazil, Canada, Chile, China, Colombia, Finland, Italy, Kazakhstan, New Zealand, Paraguay, Spain, USA, Uruguay, Venezuela, Zambia and Zimbabwe.<br>• Mix of conventional and reduced tillage in the rest of the world. | | | |
| OFF-SEASON COVER | • Cultivation only with cover crops in tropics according to Koeppen-Geiger regions (Fig. S1) (Kottek et al., 2006).<br>• Mix of off-season cover with and without cover crops in temperate and cold zones.<br>• No cover crops in arid regions. | | | |
| CONSERVATION PRACTICE FACTOR | Slope | 0 – 16 % | 16 – 30 % | > 30 % |
| | P-Factor | 1.0 | 0.5 | 0.15 |
| CROP | Water erosion is simulated in wheat and maize fields based on the global crop distribution by MIRCA2000 (Fig. S2) (Portmann et al., 2010). | | | |

| IRRIGATION | • Only on slopes ≤ 5 %. |
|---|---|
| | • Weighted average of irrigated and rainfed cropland based on MIRCA2000 (Portmann et al., 2010). |
| METHOD | MUSS water erosion equation. |
| AGGREGATION | Median of all management scenarios per grid cell and region |


Table 4: First-order sensitivity indices (SI) ranking for the five most sensitive input parameters (PARM) for each
water erosion equation including slope steepness (SLP), daily precipitation (PRCP), soil hydrologic group (HSG),
land use number (LUN), soil silt content (SILT), soil sand content (SAND), curve number parameter (S301),
maximum air temperature (TMX) and crop residues left after harvest (ORHI). The sensitivity indices of the
remaining parameters are presented in Table S3.

| rank | AOF | | MUSL | | MUSS | | MUST | | RUSLE2 | | RUSLE | | USLE | |
|---|---|---|---|---|---|---|---|---|---|---|---|---|---|---|
| | PARM | SI | PARM | SI | PARM | SI | PARM | SI | PARM | SI | PARM | SI | PARM | SI |
| 1 | SLP | 0.47 | SLP | 0.47 | SLP | 0.46 | SLP | 0.48 | SLP | 0.46 | SLP | 0.50 | SLP | 0.54 |
| 2 | PRCP | 0.13 | PRCP | 0.10 | PRCP | 0.12 | PRCP | 0.09 | PRCP | 0.16 | PRCP | 0.20 | PRCP | 0.18 |
| 3 | HSG | 0.03 | HSG | 0.04 | HSG | 0.05 | HSG | 0.04 | HSG | 0.03 | SAND | 0.05 | SILT | 0.02 |
| 4 | SILT | 0.02 | LUN | 0.02 | LUN | 0.02 | LUN | 0.02 | SAND | 0.01 | TMX | 0.01 | TMX | 0.01 |
| 5 | LUN | 0.01 | SILT | 0.02 | S301 | 0.01 | SILT | 0.02 | LUN | 0.01 | ORHI | 0.01 | ORHI | 0.01 |
| … | … | … | … | … | … | … | … | … | … | … | … | … | … | … |
| sum | | 0.69 | | 0.68 | | 0.71 | | 0.69 | | 0.71 | | 0.78 | | 0.77 |



Table 5: Total-order sensitivity indices (SI) ranking for the five most sensitive input parameters (PARM) for each
water erosion equation including slope steepness (SLP), daily precipitation (PRCP), soil hydrologic group (HSG),
land use number (LUN), soil silt content (SILT), soil sand content (SAND), maximum air temperature (TMX)
and crop residues left after harvest (ORHI). The sensitivity indices of the remaining parameters are presented in
Table S3.

| rank | AOF | | MUSL | | MUSS | | MUST | | RUSLE2 | | RUSLE | | USLE | |
|---|---|---|---|---|---|---|---|---|---|---|---|---|---|---|
| | PARM | SI | PARM | SI | PARM | SI | PARM | SI | PARM | SI | PARM | SI | PARM | SI |
| 1 | SLP | 0.68 | SLP | 0.68 | SLP | 0.63 | SLP | 0.68 | SLP | 0.66 | SLP | 0.69 | SLP | 0.75 |
| 2 | PRCP | 0.28 | PRCP | 0.23 | PRCP | 0.22 | PRCP | 0.21 | PRCP | 0.32 | PRCP | 0.36 | PRCP | 0.36 |
| 3 | HSG | 0.09 | HSG | 0.12 | HSG | 0.13 | HSG | 0.12 | HSG | 0.08 | SAND | 0.12 | SILT | 0.05 |
| 4 | SILT | 0.07 | LUN | 0.07 | LUN | 0.07 | LUN | 0.07 | LUN | 0.05 | TMX | 0.02 | TMX | 0.02 |
| 5 | LUN | 0.05 | SILT | 0.07 | SILT | 0.05 | SILT | 0.07 | SAND | 0.04 | ORHI | 0.01 | SAND | 0.01 |
| … | … | … | … | … | … | … | … | … | … | … | … | … | … | … |
| sum | | 1.29 | | 1.30 | | 1.25 | | 1.27 | | 1.34 | | 1.27 | | 1.27 |
