# Peer review of "Uncertainties, sensitivities and robustness of simulated water erosion in an EPIC-based global-gridded crop model"

_Biogeosciences, 2020_

## Referee Comment (RC1) · Anonymous Referee #1 · 4 May 2020

Despite the decades of research, modelling spatially distibuted phenomena such as soil water erosion, still represents very challenging job. The biggest challenge lies in comparing modelled and measured soil eosion rates, especially in case of global scaled models such EPIC. The main added values of presented paper are: 1. Evaluation of simulation results against field data and uncertainty assessment. Uncertainty assessment represents a crucial factor, when communicating the results of simulation and futher incorporation of such models into for instance global circulation models. 2. The authors pointed out several obstacles, which prevent further development of soil erosion modelling research such as lack of uniform and reliable data on water erosion rates, lack of datasets providing distributed data on topography, soil, climate, land use

and field management at the field scale. Supplementary TableS5 contains the list of soil erosion measurement records, it would be good to add an information about the scale of measurement (plot, field, . . .) The article is of high scientific value and I recommend it for the publication without any substantial revision.

————————————————————

---

## Author Comment (AC1) · 12 May 2020

Thank you for your comment. The supplementary TableS5 includes a column specifying the measurement method of each record as either "experimental plot" or "radioisotopic". These values indicate that each record represents a rather small spatial scale typical for runoff plots or radioisotopic measurements, which are commonly derived from a transect of sampling points. It will be difficult to obtain the area of each plot or transect for most records. However, we suggest to rename the method value "radioisotopic" to "radioisotopic measurement transect" to clarify the spatial scale of CS-137 measurements to readers less familiar with the method.

---

## Referee Comment (RC2) · Anonymous Referee #2 · 17 Jun 2020

This appears to have serious methodological problems before one gets to the results.

The methods used have all been seriously criticised and in most cases the authors do not seem aware of these issues or chose to ignore them.

The EPIC model has been used to look at climate change impacts on crop yields and erosion rates e.g. Favis-Mortlock et al. (1991) and to model 7000 years of erosion under changing climates and land uses for a single field (Favis-Mortlock et al., 1997). It is stressed that EPIC needs calibration in order to give reasonable results. This is the very firm conclusion of the GCTE erosion model testing exercise (Favis-Mortlock, 1998; Boardman and Favis-Mortlock, 1998).

[Figure]

The authors claim that they are evaluating their results against field-scale measures (lines 84 and 95). This is not the case: they use 137Cs and erosion plot data (line 219). Erosion plot data cannot be up-scaled to field scale: it is useful for relative assessments e.g Cerdan et al. (2010). Extrapolation from12 plots in central Belgium to give an average rate of erosion for Europe is a well-known (?) example of misuse of experimental plot data (Boardman, 1998): the current paper is heading in that direction!

RUSLE is an unvalidated model and its problems and poor performance are reviewed in Evans and Boardman (2016a and b).

For a review of the general problems of using erosion models see Favis-Mortlock et al. (2001): in Harmon and Doe (ed) book.

137Cs has been seriously criticised recently (Parsons and Foster, 2011). The technique should not be used without dealing with these limitations. This problem is ignored in the paper.

It is simply not true to claim that there is a limited availability of field data and lack of long term measurements (lines 68-69). There are extensive data sets from Switzerland, north Germany and the UK. These could be used to validate the results of erosion models: see Boardman and Evans (2020: PPG) for a review of these methods of assessment of erosion at a field scale.

The method of deriving a common slope within an area of 9x56 km is not clear and seems rather dubious (line 122). Averaging slope from a large cell (eg. 1km2) is a common failing of erosion modelling exercises (e.g. Evans and Brazier, 2005).

One conclusion seems to be that wheat erodes are a greater average rate ((19t/ha) than maize (6t/ha) (line 244): this is contrary to all field evidence that I am aware of.

I strongly suggest that the methodological problems make this paper unfit for publication.

---

## Referee Comment (RC3) · Anonymous Referee #3 · 22 Jun 2020

Review of

Uncertainties, sensitivities and robustness of simulated water erosion in an EPIC-based global-gridded crop model

By T. W. Carr et al.

This manuscript describes a study to characterize global soil erosion rates on cropland using the exploration of a large parameter space of driver data and erosion models. Starting with global information on climate, soils, agricultural practices, and field properties, the authors calculate representative erosion rates. In a series of experiments, they show the sensitivity of the model to driving inputs and parameter assumptions.

[Figure]

They evaluate the model results against a large dataset of observed soil erosion data. The authors conclude that the model results are very sensitive to assumptions about management strategy, and the accuracy of the model is limited by a lack of field observations for calibration and evaluation.

In general this manuscript is well written and simple enough to understand. However some key information is lacking in the main text of the manuscript, and some of the results seem rather suspicious, possibly because of artifacts in the input data. In particular, the headline numbers for global soil erosion, and the mapped model output, appear to be strongly influenced by erosion in mountainous areas, where in reality land use for agriculture may be much more limited than the model assumes. These issues need to be addressed in a revision before the manuscript is ready for publication.

Looking at the model results in Figure 2a, what stands out immediately is that very high rates of erosion are plotted in many regions of the world where I would not be sure that there is any significant amount of agriculture, including the central highlands of Borneo, the Himalaya, eastern Madagascar, South Korea, and parts of the Alps. These are indeed high-rainfall/high slope regions and in some of the area agriculture is practiced. But where there is cropland, it almost certainly must be limited to valley bottoms or other low-slope areas, or only performed with substantial investment in erosion mitigation measures, such as terracing. Digging deep into the manuscript supplementary materials, I discovered that the actual crop distribution data used in this study (5') comes from Portmann et al. (2010). This citation, and explanation for how the crop areas were determined, must be moved to the main body of the text. It appears that Portmann et al. (2010) do not use slope or any other topographic characteristics in determining the spatial allocation of cropland in their crop area maps. Furthermore, 5' resolution is probably too coarse even in the authors' own admission to accurately determine appropriate mean slope classes for their soil erosion calculations.

These limitations mean that the headline numbers for erosion (e.g., lines 25-26 of the abstract), and much of the results are likely to be skewed by calculations that are not

realistic, because they are biased by high-slope/high-precipitation areas where in reality, agriculture is not practiced at all, or only in very limited and specialized forms, e.g., agroforestry, and perennial crops such as tea and orchards. This source of uncertainty needs to be addressed more thoroughly and the methods presented more transparently before this manuscript is suitable for publication.

Finally, it would be interesting if the authors performed a "reality check" on their erosion numbers. With some of the extreme values that they calculated, could agriculture be sustainable at all? How long would it take before most soil is completely eroded away?

Lines 122-123

What is the justification for choosing the "most common slope"? At the very least, wouldn't it make more sense to choose the lowest slope class in each 5' gridcell? At least until all of the area in the slope class is filled by agricultural land use before moving to the next steeper class? If not, the authors' choice of modal slope class should be justified with citations.

Lines 184-187

Again, where is the evidence that steeper slopes are actually cultivated, and on what basis are these P-factors selected? Were the parameters selected using empirical evidence, or a citation?

Lines 352-355 What is the evidence that any of these "cultivation areas with slopes steeper than 8

Lines 377-379

"...a significant share of the estimated 379 soil removal of 7 Gt a-1 originates from small wheat and maize fields on steep slopes with strong annual precipitation". So here the authors admit that the global numbers are skewed by extreme levels of simulated erosion. But more evidence that these fields actually exist needs to be provided.

Lines 391-392

How were the countries where "conservation agriculture... is likely" selected? What evidence is there for this?

Lines 423-425

That "... many older measurements are poorly accessible as they are not available online" seems to be a bit of a weak argument for not collecting more measurements on soil erosion. Can the authors elaborate a bit more in what kind of data are out there and precisely what it would take to utilize them for future studies?

Lines 466-467

Yes it seems clear that increased resolution would be important. Several datasets are already available however, including 100m agricultural cover fraction data (Buchhorn et al., 2019) and 90m topography from a range of different datasets, such as MERIT-Hydro (Yamazaki et al., 2019). Global climate and soils data are available at at least 1km resolution and could be downscaled (Fick  Hijmans, 2017; Hengl et al., 2017). Some more explanation as to why the authors were limited to 5' and more concrete recommendations for future research would be valuable.

Lines 473-474

As the high erosion "areas represent only a small fraction of global cropland 474 for wheat and maize", why not show median values as the headline results instead of means?

Lines 684-689; Figure 2

I would like to see the map and statistics separated out into two, one figure set each for maize and wheat. As the growing areas are different and only partially overlapping, it would be very helpful to see these individually in the main body of the manuscript.

Lines 706-709; Figure 7

I am quite suspicious that there is any substantial amount agriculture at all in the purple areas marked on the map, e.g., Borneo highlands, northern Laos, Himalayan front, western Madagascar, Korea, Japan. If there is, agriculture must be limited to valley bottoms that are not detected at 5' resolution or done with extreme terracing.

Lines 691-693; Figures 3 and 4

Would also be useful to see how much uncertainty is caused by the assumption of what slopes are being farmed, e.g., always lowest slopes first, mean slope, median slope, etc.

List of References

Buchhorn, M., Smets, B., Bertels, L., Lesiv, M., N.-E., T., Herold, M., Fritz, S. (2019). Copernicus Global Land Service: Land Cover 100m, epoch 2015, Globe (Version V2.0.2).

Fick, S. E., Hijmans, R. J. (2017). WorldClim 2: new 1-km spatial resolution climate surfaces for global land areas. Int J Climatol, 37(12), 4302-4315. doi:10.1002/joc.5086

Hengl, T., Mendes de Jesus, J., Heuvelink, G. B., Ruiperez Gonzalez, M., Kilibarda, M., Blagotic, A., Shangguan, W., Wright, M. N., Geng, X., Bauer-Marschallinger, B., Guevara, M. A., Vargas, R., MacMillan, R. A., Batjes, N. H., Leenaars, J. G., Ribeiro, E., Wheeler, I., Mantel, S., Kempen, B. (2017). SoilGrids250m: Global gridded soil information based on machine learning. Plos One, 12(2), e0169748. doi:10.1371/journal.pone.0169748

Portmann, F. T., Siebert, S., Döll, P. (2010). MIRCA2000-Global monthly irrigated and rainfed crop areas around the year 2000: A new high-resolution data set for agricultural and hydrological modeling. Global Biogeochem Cy, 24(1), n/a-n/a. doi:10.1029/2008gb003435

Yamazaki, D., Ikeshima, D., Sosa, J., Bates, P. D., Allen, G., Pavelsky, T. (2019). MERIT Hydro: A high-resolution global hydrography map based on latest topography
datasets. Water Resour Res. doi:10.1029/2019wr024873

---

## Author Comment (AC2) · 13 Jul 2020

**Uncertainties, sensitivities and robustness of simulated water erosion in an EPIC based global-gridded crop model**
By T. W. Carr et al.

**Reply to Anonymous Referee #2**

Dear reviewer,

Before we address each of your comments, we briefly clarify the main incentive of this study. Large-scale indicators about global-scale phenomena are needed to inform all major environmental and agricultural policies such as the European Union's Common Agricultural Policy (CAP), the United Nations Sustainable Development Goals (SDGs), the United Nations Convention to Combat Desertification (UNCCD) and the Intergovernmental Science-Policy Platform on Biodiversity and Ecosystem Services (IPBES). Water erosion will not be considered in any of these major environmental and agricultural policy programs without large-scale assessments (Alewell et al., 2019). Global-gridded crop models have the capacity to develop large-scale indicators and to inform about agricultural productivity in a transparent and consistent way across large areas (Mueller et al., 2017). This paper aims to address the gaps in the literature of the links between water erosion and crop cultivation in various large-scale and global impact assessments, as accurately as is currently feasible given data availability. Studies on large-scale and global climate change impacts in the agricultural sector lack representation of water erosion impacts on crops (Balkovič et al., 2018), studies on global terrestrial carbon fluxes do not account for carbon runoff from cropland through soil erosion (Chappell et al., 2016), and studies assessing large-scale and global market impacts of soil erosion rely on simple linear estimates of water erosion impacts on crop production (Panagos et al., 2018; Sartori et al., 2019). It is important, though, to understand the limitations of such assessments so that they can be improved in the future, and that is why we systematically test a number of approaches in our paper.

The model used in this study has been confirmed as a reliable tool for global crop yield projections and stands out against comparable global models due to its detailed representation of soil processes including water erosion and the impacts of tillage on soil properties. Therefore, a global-gridded EPIC model has the potential to deliver much needed indicators about relationships between erosion and crop productivity on large and global scales. This paper has the objective to support the ongoing development of large-scale and global model applications by analysing the robustness of average long-term water erosion estimates generated with global-gridded crop models. In other words, we do not attempt to reproduce soil loss rates measured in single fields but to analyse the robustness of large-scale water erosion estimates based on global-gridded crop model outputs. Moreover, we focus on the necessary improvements needed to account for water erosion in global models by analysing and discussing its robustness across global agro-environmental conditions, the importance of global input data on field management and the uncertainty resulting from different erosion equations.

Most of the criticism by the referee is directed against the lack of field data, model calibration, and general criticism on RUSLE-based erosion models and the $^{137}Cs$-method used for erosion measurements. Each point of criticism is addressed in the following (Referee comments are printed in purple and our replies are listed below):

*The EPIC model has been used to look at climate change impacts on crop yields and erosion rates e.g. Favis-Mortlock et al. (1991) and to model 7000 years of erosion under changing climates and land uses for a single field (Favis-Mortlock et al., 1997). It is stressed that EPIC needs calibration in order*

*to give reasonable results. This is the very firm conclusion of the GCTE erosion model testing exercise (Favis-Mortlock, 1998; Boardman and Favis-Mortlock, 1998).*

The first point of criticism is on the need to calibrate the EPIC model for reasonable results. The EPIC-IIASA model uses state-of-the-art global crop management and agro-environmental input data and has been positively evaluated for representing national average yields and inter-annual yield variability globally (Balkovič et al., 2014). It was used in several studies and its outputs have been compared to regional yield statistics and other global crop and land use models as a part of ISI-MIP and GGCMI model inter-comparison initiatives (Mueller et al., 2017). Global crop models are not calibrated to reproduce crop yields at field scale but rather to represent the crop yield patterns across regions and countries to address research questions that cannot be addressed through field scale studies. Following the same paradigm, we aim to analyse the robustness of EPIC to represent regional differences and regional spatial patterns in water erosion estimates rather than accurately reproduce erosion events occurring in the past in response to individual rain events of rainy seasons. We are aware that our approach would be inappropriate for the latter case. We are also aware that a proper model calibration is always needed to meet experimental data obtained from the field, foremost for a complex process like soil loss due to water erosion. At the same time, sound calibration for a wide range of global environments and crop management practices would require enormous capacity and work force while still facing a high degree of subjectivity in experimental data (e.g. Panagos et al. 2016). Given the current lack of consistent field measurements representing all global environments, it is not possible to produce plausible global erosion estimates using only bottom-up, field-scale modelling.

To further clarify the intention of this study, we add a reference clarifying the usefulness of large-scale models to line 57: "*Moreover, improving the representation of water erosion in large-scale models is urgently needed to inform major environmental and agricultural policy programs such as the European Union's Common Agricultural Policy (CAP), the United Nations Sustainable Development Goals (SDGs), the United Nations Convention to Combat Desertification (UNCCD) and the Intergovernmental Science-Policy Platform on Biodiversity and Ecosystem Services (IPBES) (Alewell et al., 2019).*"

We will further clarify the focus of this paper in the introduction, line 83: "*The overall aim of this study is: (i) to analyse the robustness of water erosion estimates in all global agro-environmental regions simulated with an EPIC-based global-gridded crop model; and, (ii) to discuss the main drivers affecting the robustness and the uncertainty of simulated water erosion rates on a global scale.*"

We will modify the last conclusion on line 474: "*The overlap of simulated and measured water erosion values in most environments used to produce wheat and maize underlines the robustness of an EPIC-based GGCM to simulate the differences in water erosion rates of major global crop production regions*"

*The authors claim that they are evaluating their results against field-scale measures (lines 84 and 95). This is not the case: they use 137Cs and erosion plot data (line 219). Erosion plot data cannot be up-scaled to field scale: it is useful for relative assessments e.g Cerdan et al. (2010). Extrapolation from12 plots in central Belgium to give an average rate of erosion for Europe is a well-known (?) example of misuse of experimental plot data (Boardman, 1998): the current paper is heading in that direction!*

The referee criticises the use of our erosion plot data. We do not extrapolate plot data to represent water erosion on continental scales as in the Belgium-example mentioned by the referee. As explained on line 235-239, we aggregate field data into groups with similar slope classes and precipitation classes

and compare the values to model outputs with the same slope and precipitation classes. Thereby, we analyse the robustness of model outputs for different environments characterised by the most important parameters affecting water erosion on a global scale. Slope and precipitation are the most sensitive parameters influencing model outputs and are the parameters found previously to be most critical for the robustness of RUSLE-based models. We chose this method to illustrate the varying robustness of our model outputs around the globe and we identified regions where the model performance is not sufficient, which is communicated in the discussion (line 367-370) and the conclusion (line 472 – 473).

More generally, plot and field scale are not exactly defined. These two categories can be overlapping as large plots can have similar slope length as fields. Plots of 20 m are most common, and if they are equipped by multislot divisors, typing buckets they can be up to 100 m long, or even several hundred meters if equipped by Coshocton wheels. We know about 100 m erosion plots from Slovakia and Austria or 30 m plots in Zimbabwe. Fields can also have slope length about 20-100 meters, especially if they are on slopes, contour oriented or when they belong to small family farms in developing countries. We have seen 30-50 m long slopes in Uganda, Madagascar, or Slovakia.

Regarding the real scale of our data, most of them (both 137Cs and erosion plots) represent slopes of 10-100 m so they are at the margin of plot and field scale. Therefore, in the paper, we will name the range of spatial scales of the field data on line 218 (we increased the field data sample as explained below). "*We compared our simulated water erosion rates with 606 soil erosion measurements on arable land from 39 countries representing plot, hillslope and field scale; 315 records were derived by the $^{137}$Cs method, 188 records from erosion plot experiments, and 103 from volumetric observations.* "

The term "*field-scale measurements*" on line 84 will be deleted as this sentence will be replaced with the sentence stating the overall aim of this study presented above.

The term "*field measurements*" on line 95 is not related to scale

*RUSLE is an unvalidated model and its problems and poor performance are reviewed in Evans and Boardman (2016a and b). For a review of the general problems of using erosion models see Favis-Mortlock et al. (2001): in Harmon and Doe (ed) book.*

Further criticism is focused on the validity of the RUSLE method in general, which is one out of seven similar erosion models included in our EPIC study. The RUSLE methodology is based on more than 10,000 plot-years of experiments and has been applied in more than 100 countries with varying robustness (Alewell et al., 2019; Renard et al., 1997; Wischmeier & Smith, 1978). As already disputed above, the limited availability of global input and experimental data requires simple erosion models for global studies. Therefore, RUSLE has been chosen by most studies focusing on global erosion (Borrelli et al., 2017; Doetterl et al., 2012; van Oost et al., 2007) and it is unfortunate that the referee disagrees with the approach adopted by most of the scientific community. We agree that the varying robustness of RUSLE-based methods around the world need to be considered, which is one of the main foci of this paper and has been addressed in the introduction (lines 66 – 68) and in the discussion (lines 360 – 373). Furthermore, we present the varying estimates of different water erosion equations and thereby demonstrate the uncertainty of relying on erosion estimates from a single erosion model (lines 268 – 271, 381 – 384, 409 – 417). The references used for criticising the RUSLE method by the referee promote field studies as alternatives to erosion models. However, this is not feasible for a range of research applications focusing on analysing scenarios at large and global scales (Alewell et al., 2019; Panagos et al., 2016), which is also a major purpose of global gridded crop models. Instead, focusing on the improvement of the application of simple erosion models such as RUSLE-based models

as intended with this paper supports the ongoing development of global erosion impact assessments (Naipal et al., 2015).

*137Cs has been seriously criticised recently (Parsons and Foster, 2011). The technique should not be used without dealing with these limitations. This problem is ignored in the paper.*

The objections of Parsons and Foster (2011) were discussed by Mabit et al. (2013) published as a direct answer to Parsons and Foster in the same journal. One of the most important confirmation of the usefulness of the 137Cs method given by Mabit et al (2013) are the positive results of a comparison between erosion values obtained with various measurement methods and erosion values obtained by 137Cs method: "*Several studies at various scales (from plot to watershed scales) have been conducted to compare the 137Cs based erosion rates obtained with direct erosion measurement approaches such as erosion plots and catchment sediment yields (e.g. Schuller et al., 2003; Porto et al., 2003; Porto et al., 2004; Stankoviansky et al., 2006;Mabit et al., 2009; Parsons et al., 2010; Ceaglio et al., 2012; Porto and Walling, 2012a, 2012b). In northern California, rates of erosion from accumulated pond sediment and soil lost from hillsides assessed through 137Cs agreed well (i.e. O'Farrell et al., 2007). In Italy, the reliability of the mass balance model at slope and catchment scale was verified and confirmed by comparing the basin net soil erosion value obtained by 137Cs measurements against the mean annual value of sediment yield measured at the basin outlet (i.e. Di Stefano et al., 2005). For three small catchments located in Southern Italy, measurements of sediment output validated 137Cs theoretical conversion models to estimate soil redistribution rates (i.e. Porto et al., 2004)*".

Another paper by Parsons (2019) also criticises erosion plots, modelling (several models) and volumetric measurements of rills. As our paper is focused on global erosion modelling, we did not include the extensive literature on the advantages and disadvantages of each erosion measurement method. A comprehensive discussion of the objections against the 137Cs method and other measurement methods would but too extensive for our paper. However, we will address Parsons and Fosters objections in this response:

a. Regional and local heterogeneity of 137Cs fallout and its local redistribution by vegetation, infiltration, bioturbation, etc.: this is well known fact but it is considered in the methodology and the solution is the selection of reference sites in immediate vicinity of study sites and using statistical criteria (variation coefficient) criteria for microvariability of 137Cs. The microvariability of 137Cs is similar to most other soil properties and the potential error is similar or smaller than for other erosion measuring methods (as it will be demonstrated further). There is set a limit for variation coefficient which reference site should not exceed.

b. Mobility of 137Cs: the references presented by Mabit et al. (2013) clearly demonstrate that 137Cs is strongly bind to colloids (with details on particular clay minerals and organic matter) and its mobility (washing by runoff during the deposition, leaching and plant uptake are really negligible (representing less than one percentage to very few percentages of 137Cs fallout). The ideas about mobility, leaching and plant uptake of 137Cs presented by Parsons and Foster are based mainly on laboratory experiments with Caesium which do not represent its real behaviour in nature because the laboratory conditions are artificial, and the used doses of Caesium are too high. Parsons and Foster admit that in their paper. If these ideas would be correct, we would frequently see leached Caesium in deeper part of soil profile or the whole inventories at sites undisturbed by erosion would be depleted by plant uptake. But this is never the case in natural soil.

c. Selective removal and sorting of the particles: This is well known by all authors using the 137Cs method and it is mentioned in all handbooks and conversion procedure has a parameter for that. But it is true that this factor is difficult to quantify. But similar weak points are common in most methods of erosion measurements as will be demonstrated below.

d. Conversion models: Indeed, procedures to calculate soil loss require some parameters which are not always available. The accuracy of calculation depends on quality of input data and is different in individual studies. But this is the case for all erosion measurements and all methods as it will be demonstrated below.

e. Sample preparation: the problems with coarse fraction, dry or wet sieving can occur in specific soils, especially when having porous coarse fraction and concretions containing clay or organic matter so that the coarse fraction has electrical charges. This of course should be understood by staff who is expected to have basic pedological education. It is true that not all these details are mentioned in every methodological guidance, but they are discussed by some authors.

f. Gammaspectroscopy: Criticism of gammaspectroscopy is not relevant at all. Indeed, there are possible different geometries of detectors and detectors have to be calibrated. But this is task for laboratory staff. Cs-method obviously require staff having background in nuclear physics. Each laboratory method whether physical or chemical require staff with appropriate qualification.

Finally, we accept that we could have mentioned the limitations of the field data we collected and referred to some of the existing literature. We will briefly mention the main limitations of the field data we collected on line 449: "*The accuracy of the $^{137}Cs$ method has been criticised as well as the subjectivity of choosing reference sites and sample points (Parsons and Foster, 2011). Similarly, the accuracy of erosion measurements derived from runoff plots has been challenged due to the complexity in designing plot experiments and collecting soil runoff data (Boix-Fayos et al., 2006). Also, volumetric surveys have been criticised due to their high degree of subjectivity (Panagos et al., 2016). Nevertheless, the usefulness of the $^{137}Cs$ method, plot experiments and volumetric surveys has been confirmed in various studies (Boardman and Evans, 2020; Cerdan et al., 2010; Mabit et al., 2013).*"

*It is simply not true to claim that there is a limited availability of field data and lack of long term measurements (lines 68-69). There are extensive data sets from Switzerland, north Germany and the UK. These could be used to validate the results of erosion models: see Boardman and Evans (2020: PPG) for a review of these methods of assessment of erosion at a field scale.*

The need for slope and precipitation information accompanying erosion measurements in our evaluation method narrows down suitable field datasets, as meta data such as slope steepness is often not available in published datasets. Moreover, when we refer to a lack of field data, we are talking about the global scale and especially the imbalance in data availability among world regions. We have addressed the uneven distribution of global field data in the introduction (line 68) and the skewed focus of field data on the United States and Europe in the discussion (line 426). Also, the difficulty of gathering field data from a very heterogenous mix of measurement methods is comprehensively addressed in the discussion (line 419-449). We attempted to gather field data from as many continents as possible to represent different global environments. Insufficient field data representing all global regions and the lack of sufficient metadata in available datasets to further improve erosion modelling on large and global scale is an important conclusion of this paper (line 471).

We will further clarify these issues in the discussion after line 425 "*A variety of factors influencing water erosion such as climate, field topography, soil properties and field management need to be considered when modelling water erosion but are often not reported in available field measurements*

*(García-Ruiz et al., 2015). This hampers a direct comparison between simulated and observed water erosion values. We demonstrated the varying match between measured and simulated water erosion using different tillage and cover crop scenarios. Metadata on field management often only provides the crop cultivated and therefore the conditions under which erosion was measured in the field are not known sufficiently to evaluate erosion values simulated under different field management scenarios. Similarly, information on field topography and soil properties is often not provided with recorded field measurements and thus their use is limited in an evaluation of water erosion estimates simulated in different global environments. Moreover, the geographical distribution of erosion data is unbalanced. Most data are concentrated in the United States, West Europe and the West Mediterranean (García-Ruiz et al., 2015). In summary, there is a lack of field data representing all needed regions, situations and scenarios (Alewell et al., 2019)."*

We increased the field data sample to 606 records using publicly available datasets from Germany and the UK provided by Auerswald et al. (2009) and Benaud et al. (2020).

We add a description of the additional field data after line 231: "*We use also erosion rate measurements on arable land based on plot experiments and volumetric surveys collected by Auerswald et al. (2009), Benaud et al. (2020) and García-Ruiz et al. (2015). Measurements of soil runoff from plots are useful to compare the relative changes in erosion under different topographic, climate and controlled field management conditions (Auerswald et al., 2009; Cerdan et al., 2010). In volumetric surveys erosion rates are derived from measuring the volume of visible traces of erosion such as rills, gullies and fans, and have been considered as a good method to monitor erosion in large fields (Boardman and Evans, 2020).*"

An attempt to further increase the field data sample using the sources listed in Boardman & Evans (2020) was not possible as only aggregated values are published in the listed papers, which present large field datasets.

Some more general remarks: Although huge effort was spent by the erosion community to generate an enormous number of data, there is a serious lack of useful data to evaluate large-scale models. It was very challenging to gather the amount of field data used in this study for the following reasons:

1. As we are working with USLE-derived models, we were looking for data from certain spatial extend only. The USLE was developed at short slopes and represents sheet erosion and initial stages of rill erosion. Therefore, we preferred data from ca 10-100 m long slopes. This is the case for field data derived with the 137Cs method and erosion plots. Initially, we decided against using data from long slopes (several hundred of meters), which is usually the case of volumetric measurements of rill erosion and hydrological measurements in small watersheds.

2. At the beginning of this project we tried to focus only on data derived by 137Cs method as different methods represent different erosion processes and are subject to different systematic errors which are presented in the following:

   **Erosion plots with total collection of sediment** have problems to collect great volumes of sediment in case of extreme rain events (the sediment may exceed the capacity of containers) and it can be difficult to determine the weight of sediment (when it is wet, the whole volume cannot be carried to the laboratory for drying so the quantity of soil in the collected mud is just estimated by taking a sample of the mud to measure concentration).

**Multislot divisors, tipping buckets and Coshocton wheels** have many technical problems (multislot divisors may split the sediment unequally if they are not fixed exactly horizontally, the tipping buckets and Coshocton wheels loose part of the sediment when they are tipping or when the stream is strong water is splashing out, if the stream is week the soil material is sedimenting immediately in tipping buckets and the sample is not representative, data loggers can break, etc.).

Studies with replicated plots showed great variability for replicas. Nearing et al. (1999) report from almost 800 replicated plot pairs/year data a coefficient of variation ranging between 14% and 150%. Variability was decreasing with increasing soil loss. The rates of 10 tons/ha had coefficient of variability of ca 40%.

**Geodetical method (erosion pins)** has much bigger error than erosion plots because it has poor resolution. If one mm of soil is removed, the change of surface is hardly seen. But this represents already 10 tons of soil per hectare. On arable land the geodetic method has problems to distinguish between erosion and compaction.

**Rill and gully volumetric measurements** (underline: preferred method in the reference provided by the referee (Boardman & Evans (2020))) neglects sheet erosion completely. The recalculation of obtained volumetric data to weight is problematic because of the limited information on soil bulk density and its vertical and horizontal variability. This is problematic as we need data in t/ha to compare with models. Usually it is not indicated whether rill measurement represent the whole year or only the vegetation season, whether they involve rills from snow melt or not, etc.

The measurement of rill volumes itself is a source of huge error. Authors who use this method know this (for example Evans, 2013, stated: "Mollenhauer notes (2002: 4) 'The measurement of lengths and cross-section areas of linear forms can be extremely error ridden'; and quotes from Ruttiman and Prasuhn's (1990) work in Switzerland that the 'total error for soil loss volume can amount to between 20 and 40%' (Mollenhauer, 2002: 4)" and further "The level of accuracy of field-based estimates depends on the amount of time spent in measuring/estimating the number, lengths and dimensions of rills and gullies and assessing volumes of depositional features such as fans. The larger an area surveyed inevitably means that cruder estimates of eroded amount will be made, for example, numbers, lengths and cross-sectional areas of channels will all have to be estimated rather than measured").

Measurements have very few traverses (sometimes only 4, Boardman, 2003), which is a huge source of error. Boardman even says that sometimes one traverse is enough (Boardman, 2003: "The number of traverses is clearly subject to the time and resources available and also the purpose of the survey as to how much detail and accuracy is required. In many situations, it is reasonable to undertake one traverse across the mid-point of the eroded slope and estimate total erosion based on the mean rill length."). Measurement based on one traverse compared to measurements based on 4 traverse revealed errors from -18,1% to +48,7%.

The method neglects interrill erosion which is an important portion of the whole erosion process. Estimates of the importance of interrill erosion differ significantly for different conditions (negligible amount of 0.3 m3/ha/y provided by Evans (1990, in Boardman 2003),

few % of total erosion: 5-11% (Morgan et al., 1987, in Evans, 2013), up to few tens of %: 25% (Prasuhn, 2011) and 10-30% (Zachar, 1982)).

Parsons (2019) emphasize that the volumetric measurements of rills severely underestimate overall erosion because rills also involve large quantity of material which was delivered by sheet erosion to rills and further transported by rills. These proportions can be 40% for rill erosion and 60% for interrill. Luk et al. (1993, in Parsons, (2019)) determined the portion of rill erosion ranging between 0 (when only sheet erosion develops) and 56%. Our own experience from Central Europe with more heavy rainfalls from own unpublished measurements is that at steep slopes (ca 8-12 degrees) it can be in some years 10-40 tons per ha. Govers and Poesen found that the proportion between rill and interrill erosion can change significantly with time and according to changes in physical properties of top layer and deeper layers either proportion of rill erosion can rise or the proportion of interrill erosion can rise. In their case study the proportion of interrill erosion was decreasing with time from 46 to 22%, but other authors found opposite trends. Therefore, estimating interrill erosion from rill erosion using fixed ratios is wrong. They also find, that interrill erosion has higher proportion on short slopes than on long slopes.

Sometimes the presented rill measurements are not real measurements but just very rough and brief estimation. The rills and their lengths are estimated from photos, where smaller rills might be difficult to detect (for example Boardman, 2003: "Ground-level photographs of rills and gullies may be used as a record of length and size; subsequent analysis shows them to be a reliable means of estimating soil losses (Watson and Evans, 1991)" or Evans, 2013: "In this review paper, 'direct' assessment of water erosion is taken to mean the mapping of erosion and deposition as evident in fields (Figures 1–9) or on ground or aerial photographs and then when possible estimating eroded volumes based on lengths and cross-sections of rills and gullies and areas and depths of deposition (Evans, 1988; Herweg, 1996; Stocking & Murnaghan, 2001)." Watson and Evans (1991) estimate the sizes of rills on photos comparing it with the widths of crop rows and height of crop and thickness of colluvial fans they estimate according to their colour. They compared the results of volumetric measurements in field and on photos (12 photos) and found ratios from 0.67 to 2.12, so in some cases the estimations on photos are higher than in field and in other cases it is opposite.

**Hydrological measurements** in elementary watershed do not represent erosion only from agricultural land but also bank erosion and road erosion, and both these can be significant.

**Sampling of suspended sediment** is not well representative, and samplers or data loggers can break. The range of discharge in small catchments is so huge that it makes instrumentation of hydrological profiles difficult.

For these various possible sources of errors, we did not want to mix up different methods. The optimal case would have been to only use field data derived from one method, but to increase the amount of data we decided to take 137Cs data and some selected erosion plots (to further increase our sample we included suitable data from volumetric surveys).

3. Large amount of existing data is not accessible for various reasons:
    a. Many older publications are in national languages

b. Many older publications are not on internet

c. Many measurements were published in grey literature, local conference proceedings, national acta of scientific institutions, unpublished reports, etc.

d. Many published data are hardly interpretable because metadata are lacking (slope lengths, or inclinations or crop cover, period of measurement is not recorded, geographical position of the sites is not recorded, many measurements were running only during vegetation period of studied crop so they do not represent annual erosion but just few months, etc.).

e. International journals do not have interest to publish usual case studies which present raw data. To get paper published the authors need to present some special objectives to follow some special goals or developing methodological innovations. Therefore, also many new data sets cannot be found online.

f. Even if paper is published, journals have usually size limitations. To save space the primary data are not presented, only the results of interpretation, statistical processing, etc. are there. In publications using 137Cs it is more common to find primary data than in studies using other methods.

Please consider that all methods have a lot of weak points, methodological shortcomings and sources of errors, uncertainties and variability and there is lack of reliable comparisons and comprehensive assessment of all methods which would be widely accepted by the whole erosion community. There are different schools and groups of researchers who use predominantly one method. One group uses 137Cs method, other groups prefer erosion plots, the next one focuses on elementary watersheds using hydrological methods based on discharge and sediment concentration sampling (or combination of plots and watersheds) and other group focuses on volumetric measurements of rills and gullies. Some researchers using certain method are very critical about other methods, but they are very tolerant regarding the shortcomings of their favoured method. Although each method has advantages and limitations and each group has success and achievements as well as challenges and failures, it is normal if individual researchers or teams prefer one particular approach. But they should respect also other approaches.

Our collected data set represents a reasonable compromise to achieve the objectives of this study. It is far beyond the capacities of the team and the objectives of this study to collect all existing erosion data. Such task would require years lasting international project with participation of research teams from most countries, so that each team would be able to revise data sources in his country and provide summary of data including those published in national language and unpublished reports. Most existing data have limited accuracy and representativeness, but we cannot wait until dense coverage of perfect data will cover the Globe. Erosion is running, agriculture is in troubles and we should proceed under existing circumstances and using available tools.

*The method of deriving a common slope within an area of 9x56 km is not clear and seems rather dubious (line 122). Averaging slope from a large cell (eg. 1km2) is a common failing of erosion modelling exercises (e.g. Evans and Brazier, 2005).*

The most common slope is determined by a slope class covering the largest area in each simulation grid. Slope classes are taken from a global terrain slope database (IIASA/FAO, 2012) and are based on a high-resolution 90 m SRTM digital elevation model. We assume that the slope class representing the largest area in each simulation unit is most likely covered by the largest area of cropland. This builds

on the idea that a spatially extensive and diverse landscape can be represented by a single "representative field" characterized by the prevailing combination of topography and soil condition found in the landscape. This method is designed to represent differences in large-scale global crop production with an emphasis on the most important global crop production regions.

We clarify the concept of the representative field on line 120: "*Each simulation unit is represented by a single field characterized by the prevailing combination of topography and soil conditions found in the landscape. Each representative field has a defined slope length (20 – 200 m) and field size (1- 10 ha) based on a set of rules for different slope classes (Table S1). The slope of each representative field is determined by the slope class covering the largest area in each simulation unit (Table S1). Slope classes are taken from a global terrain slope database (IIASA/FAO, 2012) and are based on a high-resolution 90-m SRTM digital elevation model.*"

A detailed discussion about the uncertainty in slope input data has been added in response to comments by the third referee (see response to referee #3).

*One conclusion seems to be that wheat erodes are a greater average rate ((19t/ha) than maize (6t/ha) (line 244): this is contrary to all field evidence that I am aware of.*

The criticised conclusion on falsely higher erosion rates in wheat fields compared to maize fields is based on a misunderstanding. The values represent a global average value (19 t/ha) and a median value (6 t/ha) of water erosion rates for both maize and wheat fields combined.

To avoid confusion, we will focus only on median values in the revised version as median values are less influenced by the skewed distribution of erosion values. In the discussion line 375 we mention both mean and median value to illustrate the skewed distribution of erosion rates due to very high values simulated on steep slopes.

We will present global median water erosion for both maize (7 t/ha) and wheat (5 t/ha) fields on lines 25,244 and 312, and will delete average values.

Global average water erosion values simulated under different management scenarios and different water erosion equations will be deleted on line 383 – 387 to focus only on median values.

**References**

Alewell, C., Borrelli, P., Meusburger, K., & Panagos, P. (2019). Using the USLE: Chances, challenges and limitations of soil erosion modelling. *International Soil and Water Conservation Research*, *7*(3), 203–225. https://doi.org/10.1016/j.iswcr.2019.05.004

Auerswald, K., Fiener, P., & Dikau, R. (2009). Rates of sheet and rill erosion in Germany - A meta-analysis. *Geomorphology*, *111*(3–4), 182–193. https://doi.org/10.1016/j.geomorph.2009.04.018

Balkovič, J., van der Velde, M., Skalský, R., Xiong, W., Folberth, C., Khabarov, N., et al. (2014). Global wheat production potentials and management flexibility under the representative concentration pathways. *Global and Planetary Change*, *122*, 107–121. https://doi.org/10.1016/j.gloplacha.2014.08.010

Balkovič, J., Skalský, R., Folberth, C., Khabarov, N., Schmid, E., Madaras, M., et al. (2018). Impacts and Uncertainties of +2°C of Climate Change and Soil Degradation on European Crop Calorie Supply. *Earth's Future*, *6*(3), 373–395. https://doi.org/10.1002/2017EF000629

Benaud, P., Anderson, K., Evans, M., Farrow, L., Glendell, M., James, M., et al. (2020). National-scale geodata describe widespread accelerated soil erosion. *Geoderma*, *371*(April), 114378. https://doi.org/10.1016/j.geoderma.2020.114378

Boardman J (1983) Soil erosion at Albourne, West Sussex, England. Applied Geography 3: 317–329.

Boardman, J., 2003. Soil erosion and flooding on the South Downs, southern England 1976–2001. Trans. Inst. Brit. Geogr. 28 (2), 176–196.

John Boardman, Robert Evans, 2019. The measurement, estimation and monitoring of soil erosion by runoff at the field scale: Challenges and possibilities with particular reference to Britain, Progress in Physical Geography, 1–19:

Boardman, J., & Evans, R. (2020). The measurement, estimation and monitoring of soil erosion by runoff at the field scale: Challenges and possibilities with particular reference to Britain. *Progress in Physical Geography*, *44*(1), 31–49. https://doi.org/10.1177/0309133319861833

Chappell, A., Baldock, J., & Sanderman, J. (2016). The global significance of omitting soil erosion from soil organic carbon cycling schemes. *Nature Climate Change*, *6*(2), 187–191. https://doi.org/10.1038/nclimate2829

Evans, R. 1988. Water Erosion in England and Wales 1982–1984.

Report for Soil Survey and Land Research Centre, Silsoe.

Evans, R., Brazier, R., 2005. Evaluation of modelled spatially distributed predictions of soil erosion by water versus field-based assessments. Environ. Sci. Pol. 8, 493–501.

Fischer FK, Kistler M, Brandhuber R, et al. (2017) Validation of official erosion modelling based on high resolution rain data by aerial photo erosion classification. Earth Surface Processes and Landforms. DOI: 10. 1002/esp.4216.

Fritz, S., See, L., Mccallum, I., You, L., Bun, A., Moltchanova, E., et al. (2015). Mapping global cropland and field size. *Global Change Biology*, *21*(5), 1980–1992. https://doi.org/10.1111/gcb.12838

Govers G and Poesen J (1988) Assessment of the interrill and rill contributions to a total soil loss from and upland field plot. Geomorphology 1: 343–354.

Hayward, J.A., 1968. The measurement of soil loss from fractional acre plots. Lincoln Papers in Water Resources. 5. New Zealand Agricultural Engineering Institute, Lincoln College, Canterbury, New Zealand. Hill, H.O., Mech, S.J., Pope, J.B., Progress report

Herweg, K. 1996. Assessment of Current Erosion Damage. Soil Conservation Research Programme, Ethiopia and Centre for Development and Environment, University of Berne, Berne, Switzerland.

Hudson, N.W., 1993. Field Measurement of Soil Erosion and Runoff. Soils Bulletin 68. FAO, Rome.

Lesiv, M., Laso Bayas, J. C., See, L., Duerauer, M., Dahlia, D., Durando, N., et al. (2019). Estimating the global distribution of field size using crowdsourcing. *Global Change Biology*, *25*(1), 174–186. https://doi.org/10.1111/gcb.14492

Luk, S.H., Abrahams, A.D., Parsons, A.J., 1993. Sediment sources and sediment transport by rill flow and interrill flow on a semi-arid piedmont slope southern Arizona. Catena 20, 93–111.

Morgan, R.P.C., Martin, L. & Noble, C.C. 1987. Soil Erosion in the United Kingdom: A Case Study from Mid-Bedfordshire. Occasional

paper No. 14, Silsoe College, Cranfield Institute of Technology, Silsoe.

Mueller, C., Elliott, J., Chryssanthacopoulos, J., Arneth, A., Balkovic, J., Ciais, P., et al. (2017). Global gridded crop model evaluation: Benchmarking, skills, deficiencies and implications. *Geoscientific Model Development*, *10*(4), 1403–1422. https://doi.org/10.5194/gmd-10-1403-2017

Naipal, V., Reick, C., Pongratz, J., & Van Oost, K. (2015). Improving the global applicability of the RUSLE model - Adjustment of the topographical and rainfall erosivity factors. *Geoscientific Model Development*, *8*(9), 2893–2913. https://doi.org/10.5194/gmd-8-2893-2015

Nearing, M.A., Govers, G., Norton, L.D., 1999. Variability in soil erosion data from replicated plots. Soil Sci. Soc. Am. J. 63, 1829–1835.

Prasuhn V (2011) Soil erosion in the Swiss midlands: Results of a 10-year field survey. Geomorphology 126: 32–41.

Panagos, P., Borrelli, P., Poesen, J., Meusburger, K., Ballabio, C., Lugato, E., et al. (2016). Reply to "The new assessment of soil loss by water erosion in Europe. Panagos P. et al., 2015 Environ. Sci. Policy 54, 438-447-A response" by Evans and Boardman [Environ. Sci. Policy 58, 11-15]. *Environmental Science and Policy*, *59*, 53–57. https://doi.org/10.1016/j.envsci.2016.02.010

Panagos, P., Standardi, G., Borrelli, P., Lugato, E., Montanarella, L., & Bosello, F. (2018). Cost of agricultural productivity loss due to soil erosion in the European Union: From direct cost evaluation approaches to the use of macroeconomic models. *Land Degradation and Development*, *29*(3), 471–484. https://doi.org/10.1002/ldr.2879

Renard, K., Foster, G., Weesies, G., McCool, D., & Yoder, D. (1997). Predicting soil erosion by water: a guide to conservation planning with the Revised Universal Soil Loss Equation (RUSLE). *Agricultural Handbook No. 703*. https://doi.org/DC0-16-048938-5 65–100.

Sartori, M., Philippidis, G., Ferrari, E., Borrelli, P., Lugato, E., Montanarella, L., & Panagos, P. (2019). A linkage between the biophysical and the economic: Assessing the global market impacts of soil erosion. *Land Use Policy*, *86*(December 2018), 299–312. https://doi.org/10.1016/j.landusepol.2019.05.014

Stocking, M.A. & Murnaghan, N. 2001. Handbook for the Field Assessment of Land Degradation. Earthscan, London.

Stroosnijder, L., 2005. Measurement of erosion: Is it possible? Catena, Volume 64, Issues 2–3, 30 December 2005, Pages 162-173, https://doi.org/10.1016/j.catena.2005.08.004

Watson A and Evans R (1991) A comparison of estimates of soil erosion made in the field and from photographs. Soil & Tillage Research 19: 17–27.

Wischmeier, W. H., & Smith, D. D. (1978). Predicting rainfall erosion losses. *Agriculture Handbook No. 537*, (537), 285–291. https://doi.org/10.1029/TR039i002p00285

---

## Author Comment (AC3) · 13 Jul 2020

**Uncertainties, sensitivities and robustness of simulated water erosion in an EPIC based global-gridded crop model**
By T. W. Carr et al.

**Reply to Anonymous Referee #3**

Dear reviewer, we appreciate your thoughtful comments and have responded to each in the following (Referee comments are printed in purple and are each addressed below).

*This manuscript describes a study to characterize global soil erosion rates on cropland using the exploration of a large parameter space of driver data and erosion models. Starting with global information on climate, soils, agricultural practices, and field properties, the authors calculate representative erosion rates. In a series of experiments, they show the sensitivity of the model to driving inputs and parameter assumptions. They evaluate the model results against a large dataset of observed soil erosion data. The authors conclude that the model results are very sensitive to assumptions about management strategy, and the accuracy of the model is limited by a lack of field observations for calibration and evaluation.*

*In general this manuscript is well written and simple enough to understand. However some key information is lacking in the main text of the manuscript, and some of the results seem rather suspicious, possibly because of artifacts in the input data. In particular, the headline numbers for global soil erosion, and the mapped model output, appear to be strongly influenced by erosion in mountainous areas, where in reality land use for agriculture may be much more limited than the model assumes. These issues need to be addressed in a revision before the manuscript is ready for publication.*

Our experience (not only from modelling, but also from field work, excursions and own observations and measurements) shows that in some mountainous areas the erosion rates reaches very high levels. But to some extent you are right and we considered your objection. Below, when answering other comments on this issue we will try to demonstrate in detail the situation in mountains. We will provide also some photos.

*Looking at the model results in Figure 2a, what stands out immediately is that very high rates of erosion are plotted in many regions of the world where I would not be sure that there is any significant amount of agriculture, including the central highlands of Borneo, the Himalaya, eastern Madagascar, South Korea, and parts of the Alps. These are indeed high-rainfall/high slope regions and in some of the area agriculture is practiced. But where there is cropland, it almost certainly must be limited to valley bottoms or other low-slope areas, or only performed with substantial investment in erosion mitigation measures, such as terracing.*

*Digging deep into the manuscript supplementary materials, I discovered that the actual crop distribution data used in this study (5') comes from Portmann et al. (2010). This citation, and explanation for how the crop areas were determined, must be moved to the main body of the text. It appears that Portmann et al. (2010) do not use slope or any other topographic characteristics in determining the spatial allocation of cropland in their crop area maps. Furthermore, 5' resolution is probably too coarse even in the authors' own admission to accurately determine appropriate mean slope classes for their soil erosion calculations.*

The reference to Portmann et al. (2010) is listed in Table 3 in the main text, which summarises the field management assumption and aggregation of model outputs.

We extend the reference to table 3 on line 191. "*Table 3 summarises the field management assumptions of the baseline scenario used to aggregate erosion rates in each simulation unit and region.*"

We agree, that in some mountainous areas the erosion rates obtained by modelling do not represent typical rates. However, it is not the case for all mountains. The values are overestimated most probably in mountains of temperate areas such as Europe (Alps), Korea and Japan and also in some tropical rice grooving regions on tropical Monsoon Asia (such as Borneo). However, in many mountainous areas in tropics the land is cultivated, and maize and wheat are grown even in very steep slopes and often without any soil conservation practices or conservation practices are used with insufficient efficiency. We attached a collection of photos demonstrating these phenomena. In the tropics, agriculture is very active in mountains especially for four reasons:

1. While in temperate areas such as Europe the temperature is limiting factor of agriculture and in higher altitudes (several hundred meters above the see level) it is too cold for most crops and if agriculture exist there, it is mainly grazing. In contrary, in tropics the temperature is sufficient also in high mountains.

2. The limiting factor in tropics is drought. Therefore, mountains having more rainfall and less heat are popular agricultural area. Very good example is Uganda, where the whole flat central part of the country is too dry and it is used only for grazing, while steep mountains in west and east peripheries of the country are intensively cultivated. The same is in Madagascar and in whole Latin America where mountains are more agriculturally exploited than lowlands.

3. In tropics the weathering crusts are very thick and so the loose materials constitute thick soils. Tropical soils are poor in organic matter so the difference between topsoil and subsoil is not very big. These soils can be exposed to extreme erosion rates much longer than thin soils of temperate areas, so farmers do not feel the decrease of fertility and production potential. It is decreasing slowly, and they do not realize the impact of erosion.

4. In developing countries, a great portion of land is still under hand management of small family subsistence farming. These farmers can cultivate steep slopes easier than farmers who use heavy machines.

Some examples from our own field work, where the extreme exploitation of steep slopes exists are following:

1. In Latin America, especially in mountains with volcanic rocks are cultivate even in extreme slopes and here maize is dominant crop (see photos from El Salvador).
2. Mountainous area of Sub-Saharan Africa: Extremely steep slopes are cultivated, many crops with low conserving efficiency are grown (such as cassava and other sweet potatoes, beans, maize, etc.), there are terraces, but these are not really horizontal but inclined so they reduce erosion partially but not much (see photos from Uganda and Madagascar)
3. South and East Asia: In this region the major crop is rice which is usually grown on paddy field with flood irrigation. Therefore, the large mountainous areas are well terraced and well protected from erosion. However, there are also very large mountainous areas which are not terraced at all and steep slopes are cultivated. They are growing various crops there such as

dryland rice, tea, fruit trees, sugar cane, etc. For example, in large part of southern China the rice terraces are only in valley bottoms occupying minor areas and all steep slopes occupying majority areas are used to grow sugar cane without any soil conservation.

An explanation and further discussion of the resolution of slope input and cropland distribution data follows below

*These limitations mean that the headline numbers for erosion (e.g., lines 25-26 of the abstract), and much of the results are likely to be skewed by calculations that are not realistic, because they are biased by high-slope/high-precipitation areas where in reality, agriculture is not practiced at all, or only in very limited and specialized forms, e.g., agroforestry, and perennial crops such as tea and orchards. This source of uncertainty needs to be addressed more thoroughly and the methods presented more transparently before this manuscript is suitable for publication.*

*Finally, it would be interesting if the authors performed a "reality check" on their erosion numbers. With some of the extreme values that they calculated, could agriculture be sustainable at all? How long would it take before most soil is completely eroded away?*

Unfortunately, in many mountainous areas especially in tropics conventional agriculture with very bad management is practiced and it has huge negative impact on land. We demonstrated it by photos. In many mountainous areas agriculture is not sustainable at all. But unfortunately, despite of that in many areas the destructive land management is going on and poor farmers are destroying the land completely. We even do not know how many cases like this occur. We have examples also from Slovakia, mainly historical but also recent. There are known cases that some slopes were cultivated just 10-20 years and then one extreme storm event removed all soil and the field was abandoned. In tropics frequently happens that when the field is destroyed, it is abandoned for 5-10 years being fallow and then cultivated again, but there are many cases also when slope is cultivated for 5-7 years and destroyed once for ever. See attached photos.

There are indications of this problems also in literature, for example Montgomery (2007) calculated mean erosion rate under conventional agriculture (n= 448) to be over 3.9 mm (what is ca 60 tons per hectare). Of course, such agriculture is not sustainable at all. He concluded: "A direct implication of the imbalance between agricultural soil loss and erosion under both native vegetation and geologic time is that, given time, continued soil loss will become a critical problem for global agricultural production under conventional upland farming practices." Catastrophic effects of agriculture on land discuss Pimentel and Burges (2013). They argue that annually 10 million ha of cropland is abandoned due to deteriorating production potential caused by erosion. Further, they estimate that recently the world cropland covers 1.5 billion ha but since the beginning agriculture people abandoned 2 billion ha of crop land. So, more soil is already destroyed and abandoned then what is still used.

*Lines 122-123*

*What is the justification for choosing the "most common slope"? At the very least, wouldn't it make more sense to choose the lowest slope class in each 5' gridcell? At least until all of the area in the slope class is filled by agricultural land use before moving to the next steeper class? If not, the authors' choice of modal slope class should be justified with citations.*

The most common slope is determined by the slope class covering the largest area in each simulation grid. Slope classes are taken from a global terrain slope database (IIASA/FAO, 2012) and are based on a high-resolution 90 m SRTM digital elevation model. We assume that the slope class representing the largest area in each simulation unit is most likely covered by the largest area of cropland. This builds on the idea that a spatially extensive and diverse landscape can be represented by a single "representative field" characterized by the prevailing combination of topography and soil condition found in the landscape. This method is designed to represent differences in large-scale global crop production with an emphasis on the most important global crop production regions.

We clarify the concept of the representative field on line 120: "*Each simulation unit is represented by a single field characterized by the prevailing combination of topography and soil conditions found in the landscape. Each representative field has a defined slope length (20 – 200 m) and field size (1 – 10 ha) based on a set of rules for different slope classes (Table S1). The slope of each representative field is determined by the slope class covering the largest area in each simulation unit (Table S1). Slope classes are taken from a global terrain slope database (IIASA/FAO, 2012) and are based on a high-resolution 90 m SRTM digital elevation model.*"

Slope input data is an important uncertainty for simulating global water erosion estimates as we cannot identify cultivated slopes on a global scale. This will be discussed in the revised paper by addressing: (i) the simulation of extreme values on steep slopes including the proposed 'reality check'; (ii) the intention behind using the most common slope; and, (iii) an ideal scenario, where cultivation is limited to the flattest terrain available. We used Italy as a test case for a comparison between water erosion rates simulated under the proposed ideal slope scenario and the slope scenario based on the most common slope. We chose Italy because it has large maize and wheat cultivation areas, which are located on both flat terrain in the north and in hilly regions in the south, and thus the country represents a diverse landscape with a wide range of possible water erosion rates. We add a comprehensive discussion addressing each slope related issues after line 380:

"*…High water erosion rates on steep slopes exceeding 100 t ha$^{-1}$ are also included in the field dataset compiled for this study. However, regional erosion assessments in mountainous cropland conclude that areas with extreme water erosion rates are mainly limited to marginal steep land cultivated by smallholders (Haile and Fetene, 2012; Long et al., 2006; Nyssen et al., 2019). Moreover, efforts to remove marginal farmlands from agricultural production and programs to improve land management on steep slopes have reduced high water erosion rates in several mountainous regions (Deng et al., 2012; Nyssen et al., 2015). Nevertheless, in many regions upland farming still produces extremely high erosion rates and recent pressure through increasing population and crop production demands has resulted in re-cultivation of hillslopes and a reduction of fallow periods, which limits the recovery of eroded soil (Turkelboom et al., 2008; Valentin et al., 2008). In an ideal scenario where farmers cultivate the flattest land first before moving to the next higher slope, unsustainable water erosion rates in mountainous regions would be substantially reduced. However, next to topography, the distribution of cropland is determined by climatic factors and various socio-economic factors such as competing land use and land tenure (Hazell and Wood, 2008; Nyssen et al., 2019).*

*To analyse the sustainability of simulated maize and wheat cultivation systems exposed to high erosion rates, we compared simulated annual eroded soil depth with a global dataset on modelled sedimentary deposit thickness (Pelletier et al., 2016). The comparison shows that at 4 % of grid cells permanent maize and wheat cultivation would not be sustainable as the whole soil profile would be eroded at the end of the simulation period (Fig. S11). Most of the unsustainable agriculture is simulated on steep slopes. Although we account for conservation techniques and cover crops, we do not imitate the highly complex farming practices involving intercropping techniques and fallow periods, which are common*

*on hillslopes typically manged by smallholders (Turkelboom et al., 2008). Moreover, we assume that the slope class representing the largest area in each simulation unit is most likely covered by the largest cultivated area. This builds on the idea that a spatially extensive and diverse landscape can be represented by a single "representative field" characterized by the prevailing combination of topography and soil condition found in the landscape. This method is designed to analyse the differences in large-scale global crop production systems with an emphasis on the most important crop production regions. However, this setup might not capture the complex distribution patterns of cropland in mountainous terrain.*

*The uncertainty in cropland distribution can partly be reduced by developing a higher resolution global gridded data infrastructure, which is currently not available for EPIC-IIASA. However, due to the large uncertainty in global land cover maps (Fritz et al., 2015; Lesiv et al., 2019), an explicit spatial link between cropland distribution and the corresponding slope category cannot be established without on-site observations. We test the impact of this uncertainty for erosion estimates in Italy, where large maize and wheat cultivation areas are distributed on both flat terrain in the north and mountainous regions in the south. In an ideal scenario where cropland is limited to flattest land available per grid cell, median simulated water erosion in Italy would be reduced to tolerable levels below 1 t ha$^{-1}$. However, in a scenario, where the most common slopes per grid cell are cultivated, median simulated water erosion increases to 14 t ha$^{-1}$ due to high water erosion simulated in Italy's mountainous regions (Fig. S12). This suggests a high uncertainty in global water erosion estimates due to uncertain spatial links between maize and wheat cultivation areas and different slope categories."*

The following two figures addressed in the discussion will be added to the supplementary information.

[Figure]

*Figure 1:Simulated years left until the whole soil profile is eroded in each simulation unit. Calculated as a ratio of the sedimentary deposit thickness [m] (Pelletier et al., 2016) and the eroded soil depth per year (water erosion [t ha$^{-1}$ a$^{-1}$] x bulk density [g m$^{-3}$]).*

[Figure]

*Figure 2: Comparison of slope inputs and simulated water erosion outputs between the cropland distribution scenario using the most common slopes and the cropland distribution scenario using the flattest terrain available in Italy. (a, b) distribution of the cropland share* (Portmann et al., 2010) *per slope class. (c, d) distribution of simulation units per slope class. (e) Simulated water erosion for Italy using both cropland distribution scenarios. Midlines visualise median values, boxes include values from the 25th to the 75th percentiles and whiskers bracket values between the 10th and the 90th percentiles.*

*Line 184-187*

*Again, where is the evidence that steeper slopes are actually cultivated, and on what basis are these P-factors selected? Were the parameters selected using empirical evidence, or a citation?*

As described above, we cannot be certain that steep slopes are cultivated, but we assume that steep slopes are only cultivated with conservation techniques to reduce high water erosion values. The P-values for contouring and terracing are within the range of the values reported by Morgan (2005), which are presented on lines 151 -157.

We modify line 185 accordingly: "*To account for erosion control measures reducing high water erosion on steep slopes, we use a conservation P-factor of 0.5 on slopes steeper than 16 %, and a P-factor of 0.15 on slopes steeper than 30 % to simulate contouring and terracing based on the range of P-factors presented in Morgan (2005).*"

*Lines 377-379*

*"…a significant share of the estimated soil removal of 7 Gt a-1 originates from small wheat and maize fields on steep slopes with strong annual precipitation". So here the authors admit that the global*

*numbers are skewed by extreme levels of simulated erosion. But more evidence that these fields actually exist needs to be provided.*

See comments and discussion above about the high uncertainty in cropland distribution on steep slopes.

We delete the total soil loss value from the abstract, line 26, as it is significantly influenced by extreme water erosion rates from mountainous regions. But we will keep the value in the discussion, where we address the uncertainty of the global soil loss value.

*Lines 391-392*
*How were the countries where "conservation agriculture… is likely" selected? What*
*evidence is there for this?*

We selected countries where conservation agriculture is most likely based on the share of conservation agriculture reported by AQUASTAT (2005-2014). The criteria is presented on lines 178-179 and table 3.

We will refer to AQUASTAT (FAO, 2016) on line 392.

*Lines 423-425*
*That "…many older measurements are poorly accessible as they are not available*
*online" seems to be a bit of a weak argument for not collecting more measurements*
*on soil erosion. Can the authors elaborate a bit more in what kind of data are out there*
*and precisely what it would take to utilize them for future studies?*

Indeed, this is true. There was huge amount of erosion measurements at experimental plots in many countries. For example, in USA first measurements started in 1915 and when Wischmeier and Smith were developing their equation they had about 10000 erosion plot/year data and this was in 1970ies. These data are archived by USDA but they are not directly accessible on internet. When in Germany Schwertmann was verifying USLE for Germany he used about 2500 plot/year data, but they are not available on internet and only small part was published and it was in German. We know situation mainly in central Europe. In Slovakia we have about 50 plot/year data published in Slovak language, we know about erosion plot measurements in Hungary, a lot of old data are in Czechia (starting with measurements by Maran in 1950ies, and Poland (starting by Gerlach in 1950ties), significant data set is in Austria, whole book about long term measurements (ca 20 years) was published in Croatia in Croatian language, there was extensive measurement programme in Yugoslavia (Serbia, Gavrilovic, Djorovic,). A lot of data exists in China, Japan, UK and Russia. In Africa we know about data from Uganda and Zimbabwe, most data from Francophone Africa are in French, from Latin America in Spanish, etc. There are five major reasons why most data are not available:

1. Many older publications are in national languages
2. Many older publications are not on internet
3. Many measurements were published in grey literature, local conference proceedings, national acta of scientific institutions, unpublished reports, etc.
4. Many published data are hardly interpretable because metadata are lacking (slope lengths, or inclinations or crop cover, period of measurement is not recorded, geographical position of the sites is not recorded, many measurements were running only during vegetation period of studied crop so they do not represent annual erosion, etc.).

5. International journals do not have interest to publish usual case studies which present raw data. To get paper published the authors need to present some special objectives to follow some special goals or developing methodological innovations. Therefore, also many new data sets cannot be found online.
6. Even if paper is published, journals have usually size limitations. To save space the primary data are not presented, only the results of interpretation, statistical processing, etc. are there.

The collected data set represents a reasonable compromise to achieve the objectives of this study. It is far beyond the capacities of the team and the objectives of this study to collect all existing erosion data. Such task would require 3-5 years lasting international project with participation of research teams from most countries, so that each team would be able to revise data sources in his country and provide summary of data including those published in national language and unpublished reports.

We added a more comprehensive discussion to available field data to the response addressing the second referee.

We increased the field data sample from 473 to 606 following a comment by the second referee. We will change the values and the presentation of the field data accordingly on lines 218 – 220, 232 – 234, 312 – 314.

Criteria for the appropriate selection of field data are addressed on lines 426 – 449. We will further clarify the field data needs in the discussion after line 421: "*The main reasons for the low availability of suitable data to evaluate simulated water erosion rates are twofold: (i) erosion monitoring is expensive, time consuming and labour demanding; and, (ii) primary data and metadata of measurement sites accompanying final results are often not available and many older measurements are poorly accessible as they are not available online (Benaud et al., 2020). A variety of factors influencing water erosion such as climate, field topography, soil properties and field management need to be considered when modelling water erosion but are often not reported in available field measurements (García-Ruiz et al., 2015). This hampers a direct comparison between simulated and observed water erosion values. We demonstrated the varying match between measured and simulated water erosion using different tillage and cover crop scenarios. Metadata on field management often only provides the crop cultivated and therefore the conditions under which erosion was measured in the field are not known sufficiently to evaluate erosion values simulated under different field management scenarios. Similarly, information on field topography and soil properties is often not provided with recorded field measurements and thus their use is limited in an evaluation of water erosion estimates simulated in different global environments. Moreover, the geographical distribution of erosion data is unbalanced. Most data are concentrated in the United States, West Europe and the West Mediterranean (García-Ruiz et al., 2015). In summary, there is a lack of field data representing all needed regions, situations and scenarios (Alewell et al., 2019).*"

We will additionally mention "*the lack of sufficient metadata accompanying erosion measurements*" on line 235.

We will add two sentences comparing the high variability within field data with the deviation between simulated values and measured values based on the evaluation results to Line 331: "*Outside locations combining steep slopes and strong precipitation, median deviation between simulated and measured data is lower than the variability within the field data.*" Line 451: "*In most environments relevant for maize and wheat cultivation the deviation between simulated and measured water erosion values is lower than the variability within the field data.*"

*Lines 466-467*

*Yes, it seems clear that increased resolution would be important. Several datasets are already available however, including 100m agricultural cover fraction data (Buchhorn et al., 2019) and 90m topography from a range of different datasets, such as MERITHydro (Yamazaki et al., 2019). Global climate and soils data are available at at least 1km resolution and could be downscaled (Fick Hijmans, 2017; Hengl et al., 2017). Some more explanation as to why the authors were limited to 5' and more concrete recommendations for future research would be valuable.*

We rely on the existing data infrastructure of the EPIC-IIASA model, which has been constructed and evaluated for large-scale and global crop yield projections. The EPIC-IIASA model uses state-of-the-art global crop management and agro-environmental input data and has been positively evaluated for representing national average yields and inter-annual yield variability globally (Balkovič et al., 2014). It was used in several studies and its outputs have been compared to regional yield statistics and other global crop and land use models as a part of ISI-MIP and GGCMI model inter-comparison initiatives (Mueller et al., 2017). One of the main goals of this study is to analyse if EPIC-IIASA can account for relationships between water erosion and crop cultivation. Therefore, we rely on the existing model setup and data infrastructure of EPIC-IIASA, which has been confirmed as a reliable model to simulate daily crop growth on a global scale. The Input data for EPIC-IIASA originally available at different scales were aggregated at 5' resolution grid. In EPIC-IIASA, each simulation grid is represented by a representative field (1 to 10 hectares, depending on the prevailing slope category) while the field topography was calculated as a "dominant combination" from the high-resolution 90-m SRTM digital elevation model. Given the large uncertainty in land cover maps (Fritz et al., 2015; Lesiv et al., 2019), EPIC-IIASA does not provide an explicit link between land cover category, such as cropland, and the dominant fields. Instead, an area share of each land cover category per simulation grid is provided based on the GLC2000 land cover map with 1x1 km spatial resolution.

As mentioned above a discussion on the uncertainty in cropland distribution and slope input data will be added, as well as an explanation of the concept of the "representative field".

We will further clarify the focus of this paper in the introduction, line 83: "*The overall aim of this study is: (i) to analyse the robustness of water erosion estimates in all global agro-environmental regions simulated with an EPIC-based global-gridded crop model; and, (ii) to discuss the main drivers affecting the robustness and the uncertainty of simulated water erosion rates on a global scale.*"

We further highlight the model's weakness in conclusion line 471: "*Using existing field data, we were able to identify specific environmental characteristics for which we have lower confidence in the modelled erosion rates. These are mainly found in the tropics and mountainous regions due to the high sensitivity of simulated water erosion to slope steepness and precipitation strength, and the complexity of agricultural systems in mountainous regions.*"

*Lines 473-474*
*As the high erosion "areas represent only a small fraction of global cropland for wheat and maize", why not show median values as the headline results instead of means?*

We agree that the presentation of both mean and median values can be confusing. In the revised version we will focus only on median values. However, in the discussion, line 375 we mention both mean and median value to demonstrate the skewed distribution of erosion rates due to extreme values simulated on steep slopes.

We present global median water erosion in maize (7 t/ha) and wheat (5 t/ha) fields on line 25,244, 312 and delete average values.

Global average water erosion values simulated under different management scenarios and different water erosion equations will be deleted on line 483 – 387 to focus only on median values.

We add a row to Table 3 clarifying that the median is used to aggregate water erosion values simulated under all management scenarios for simulation units and regions.

*Lines 684-689; Figure 2*
*I would like to see the map and statistics separated out into two, one figure set each*
*for maize and wheat. As the growing areas are different and only partially overlapping,*
*it would be very helpful to see these individually in the main body of the manuscript.*

We will present two maps for maize and wheat respectively in the revised paper (see figure below).

We will group bars by crop in the revised version (see figure below).

The explanation that water erosion is presented as a weighted average from maize and wheat fields will be deleted in table 3.

We include a note in the figure label explaining that pixel cells in figures do not indicate cropland sizes. *"Each pixel cell illustrates the median relative water erosion of one representative field. The extent of cropland areas is not considered in pixel cell size. "*

[Figure]

*Figure 3: Soil loss due to water erosion in maize (a) and wheat (b) fields simulated with the baseline scenario. Each pixel cell illustrates the median relative water erosion of one representative field. The extent of cropland areas is not considered in pixel cell size. The bars in the bottom plot (c) illustrate median soil removal for major world regions simulated under maize and wheat cultivation. The lines and whiskers illustrate 25th and 75th percentile values. The classification of world regions is illustrated in Fig. S4. Due to the large gap between aggregated values, all values in the bottom plot have been log-transformed to facilitate the visual comparison.*

*Lines 706-709; Figure 7*

*I am quite suspicious that there is any substantial amount agriculture at all in the purple areas marked on the map, e.g., Borneo highlands, northern Laos, Himalayan front, western Madagascar, Korea, Japan. If there is, agriculture must be limited to valley*

*bottoms that are not detected at 5' resolution or done with extreme terracing.*

Each pixel in the maps illustrates the median erosion rate of one representative field. The pixel cells in each map do not indicate total cropland area. In other words, most of the pixel in mountainous regions represent a very small cultivated area. Table 3 lists details on how erosion rates in each pixel are aggregated.

*Lines 691-693; Figures 3 and 4*
*Would also be useful to see how much uncertainty is caused by the assumption of what slopes are being farmed, e.g., always lowest slopes first, mean slope, median slope, etc.*

We will address slope uncertainty in an extended discussion (see comments above)

**References**

Alewell, C., Borrelli, P., Meusburger, K., & Panagos, P. (2019). Using the USLE: Chances, challenges and limitations of soil erosion modelling. *International Soil and Water Conservation Research*, *7*(3), 203–225. https://doi.org/10.1016/j.iswcr.2019.05.004

Balkovič, J., van der Velde, M., Skalský, R., Xiong, W., Folberth, C., Khabarov, N., et al. (2014). Global wheat production potentials and management flexibility under the representative concentration pathways. *Global and Planetary Change*, *122*, 107–121. https://doi.org/10.1016/j.gloplacha.2014.08.010

Benaud, P., Anderson, K., Evans, M., Farrow, L., Glendell, M., James, M., et al. (2020). National-scale geodata describe widespread accelerated soil erosion. *Geoderma*, *371*(April), 114378. https://doi.org/10.1016/j.geoderma.2020.114378

FAO. (2016). AQUASTAT Main Database. Retrieved July 1, 2020, from http://www.fao.org/nr/water/aquastat/data/query/index.html?lang=en

Fritz, S., See, L., Mccallum, I., You, L., Bun, A., Moltchanova, E., et al. (2015). Mapping global cropland and field size. *Global Change Biology*, *21*(5), 1980–1992. https://doi.org/10.1111/gcb.12838

Lesiv, M., Laso Bayas, J. C., See, L., Duerauer, M., Dahlia, D., Durando, N., et al. (2019). Estimating the global distribution of field size using crowdsourcing. *Global Change Biology*, *25*(1), 174–186. https://doi.org/10.1111/gcb.14492

Montgomery, D. R., 2007. Soil erosion and agricultural sustainability. Proceedings of the National Academy of Sciences of the United States of America, 104(33), 13268–72.

Morgan, R. P. C. (2005). *Soil erosion and conservation* (3rd ed.). Blackwell Science Ltd.

Mueller, C., Elliott, J., Chryssanthacopoulos, J., Arneth, A., Balkovic, J., Ciais, P., et al. (2017). Global gridded crop model evaluation: Benchmarking, skills, deficiencies and implications. *Geoscientific Model Development*, *10*(4), 1403–1422. https://doi.org/10.5194/gmd-10-1403-2017

Pelletier, J. D., Broxton, P. D., Hazenberg, P., Zeng, X., Troch, P. A., Niu, G.-Y., et al. (2016). A gridded

global data set of soil, intact regolith, and sedimentary deposit thicknesses for regional and global land surface modeling. *Journal of Advances in Modeling Earth Systems*, *8*(1), 41–65. https://doi.org/10.1002/2015MS000526

Portmann, F. T., Siebert, S., & Döll, P. (2010). MIRCA2000—Global monthly irrigated and rainfed crop areas around the year 2000: A new high-resolution data set for agricultural and hydrological modeling. *Global Biogeochemical Cycles*, *24*(1). https://doi.org/10.1029/2008GB003435

Wischmeier, W. H., & Smith, D. D. (1978). Predicting rainfall erosion losses. *Agriculture Handbook No. 537*, (537), 285–291. https://doi.org/10.1029/TR039i002p00285

**Latin America**

Mountain agriculture in El Salvador: Volcanic soils are fertile, climate is warm and moist, so even extremely steep slopes are cultivated. There is a lot of volcanic bombs and boulders but the cultivation is possible because they use hand labour (working with hoes). Maize is absolutely dominant crop.

[Figure]

Recent maize fields

Abandoned field or temporary fallows

New maize fields

Maize fields

**Latin America**

**Mountain agriculture in El Salvador**

[Figure]

Mixture of active and abandoned slope maize fields and terracese, eroded slopes and land slides

Recent maize field

Abandoned maize field

**Latin America**

Mountain agriculture in El Salvador: Land is used sometimes as cultivated maize fields, sometimes as pasture land (also after maize harvest there is grazing. Most slopes has dense microrelief of livestock pats and biological erosion. From distance it looks like pasture land but it is maize field. Very typical feature in El Salvador

[Figure]

Maize

Maize crop rezidues

**Latin America**
**Mountain agriculture in El Salvador**

[Figure]

**Latin America**

**Mountain agriculture in El Salvador**

[Figure]

Forest burned for new maize fields

Maize field full of volcanic bombs (affected by wind erosion)

**Latin America**

**Mountain agriculture in El Salvador**

[Figure]

Maize field full of volcanic bombs (affected by wind erosion)

Maize fields, crop residues visible in fromt

**Latin America**

Mountain agriculture in El Salvador

[Figure]

Maize fields with crop residues and livestock paths

Maize

Maize

**Latin America**

Mountain agriculture in El Salvador

[Figure]

Maize

**Africa**

Mountain agriculture in Uganda with terraces
but strongly affected by tills and gullies

[Figure]

**Africa**

The areas with young eucalyptus forests (in all next figures are extremely devastated and abandoned fields which were reforested with funding of World Bank

[Figure]

**Africa**

Mountain agriculture in Uganda

[Figure]

**Africa**

Mountain agriculture in Uganda

[Figure]

**Africa**

Mountain agriculture in Uganda

[Figure]

**Africa**

Mountain agriculture in Uganda

[Figure]

**Africa**

Mountain agriculture in Uganda

[Figure]

**Africa**

Mountain agriculture in Uganda

[Figure]

**Africa**

Mountain agriculture in Uganda: uppermost fields strongly degraded
by erosion and abandoned few years ago, recently used for grazing

[Figure]

**Africa**

Mountain agriculture in Uganda: maize is common crop on slopes.

**Africa**

Mountain agriculture in Uganda: maize is common crop on slopes.

[Figure]

**Africa**

Mountain agriculture in Uganda: maize is common crop on slopes.

[Figure]

[Figure]

**Africa**

Mountain agriculture in Uganda: maize is common crop on slopes.

Maize

Maize

Maize

**Africa**

Mountain agriculture in Uganda: maize is common crop on slopes.

[Figure]

**Africa**

Mountain agriculture in Uganda: maize is common crop on slopes.

[Figure]

Maize and cassawa

**Africa**

Mountain agriculture in Uganda: maize is common crop on slopes.

[Figure]

[Figure]

**Africa**

Mountain agriculture in Uganda: maize is common crop on slopes.

[Figure]

**Africa**

Mountain agriculture in Uganda: maize is common crop on slopes.

[Figure]

Maize

Sloping terraces on hills

Paddy fields with rice in valleys

**Africa**

Mountain agriculture in Madagascar

[Figure]

[Figure]

**Africa**

Mountain agriculture in Madagascar

[Figure]

**Asia**

**Mountain agriculture in South China**

Reforestation by eucalyptus

Sugar cane

Sugar cane

Sugar cane on slopes

**Asia**

Mountain agriculture in South China

sugar cane

Reforested abandoned field

sugar cane

sugar cane

sugar cane

Devastated field with sugar cane to be abandoned in next few years

**Asia**

Mountain agriculture in South China

[Figure]

**Mountain agriculture in South China**

Sheet erosion under sugar cane

[Figure]

Mountain agriculture in South China    Asia

Sheet erosion under sugar cane

**Asia**

**Slope agriculture in Loess Plateau, Northern China**

[Figure]

Maize

Maize

**Mountain Agriculture Sri Lanka**

**Asia**

**Deforestation in Vietnam**

[Figure]

[Figure]

**Central Europe**

Mountainous agriculture in Slovakia:

It began in 17th century but most intensive was in 19th century. Many fields were already devastated in last quarter of 19th century and first half of 20th century and farmers were migrating to America.

[Figure]

Completely devastated field at foot slope, recently still cultivated

[Figure]

**Central Europe**

**Mountainous agriculture in Slovakia**

In 17th – 19th century large areas of footslopes of Carpathians were devastated by huge gullies, recently mostly self-stabilised.

[Figure]

**Central Europe**

Mountainous agriculture in Slovakia

Recent ploughing in subsoil - dolomitic rock

---

## Author Response (AR1)

London, 15th August 2020

Dear Fortunat Joos (editor),

Please find our revised manuscript including tracked changes below. We also included a revised version of the supplementary information as we added two additional figures discussed in the response to referee #3 and updated supplementary information describing the field data, which we modified during the reviewing process.

Following the revised documents, we included our responses to the three reviewers and highlighted the suggested relevant changes. The photos added to the response to referee #3 are excluded in this document to reduce the file size.

Kind regards,

Tony Carr on behalf of the authors

[revised manuscript text omitted]
 PARM | SI | MUSL PARM | SI | MUSS PARM | SI | MUST PARM | SI | RUSLE2 PARM | SI | RUSLE PARM | SI | USLE PARM | SI |
|---|---|---|---|---|---|---|---|---|---|---|---|---|---|---|
| 1 | SLP | 0.47 | SLP | 0.47 | SLP | 0.46 | SLP | 0.48 | SLP | 0.46 | SLP | 0.50 | SLP | 0.54 |
| 2 | PRCP | 0.13 | PRCP | 0.10 | PRCP | 0.12 | PRCP | 0.09 | PRCP | 0.16 | PRCP | 0.20 | PRCP | 0.18 |
| 3 | HSG | 0.03 | HSG | 0.04 | HSG | 0.05 | HSG | 0.04 | HSG | 0.03 | SAND | 0.05 | SILT | 0.02 |
| 4 | SILT | 0.02 | LUN | 0.02 | LUN | 0.02 | LUN | 0.02 | SAND | 0.01 | TMX | 0.01 | TMX | 0.01 |
| 5 | LUN | 0.01 | SILT | 0.02 | S301 | 0.01 | SILT | 0.02 | LUN | 0.01 | ORHI | 0.01 | ORHI | 0.01 |
| … | … | … | … | … | … | … | … | … | … | … | … | … | … | … |
| sum | | 0.69 | | 0.68 | | 0.71 | | 0.69 | | 0.71 | | 0.78 | | 0.77 |

Table 5: Total-order sensitivity indices (SI) ranking for the five most sensitive input parameters (PARM) for each
water erosion equation including slope steepness (SLP), daily precipitation (PRCP), soil hydrologic group (HSG),
land use number (LUN), soil silt content (SILT), soil sand content (SAND), maximum air temperature (TMX)
and crop residues left after harvest (ORHI). The sensitivity indices of the remaining parameters are presented in
Table S3.

| rank | AOF PARM | SI | MUSL PARM | SI | MUSS PARM | SI | MUST PARM | SI | RUSLE2 PARM | SI | RUSLE PARM | SI | USLE PARM | SI |
|---|---|---|---|---|---|---|---|---|---|---|---|---|---|---|
| 1 | SLP | 0.68 | SLP | 0.68 | SLP | 0.63 | SLP | 0.68 | SLP | 0.66 | SLP | 0.69 | SLP | 0.75 |
| 2 | PRCP | 0.28 | PRCP | 0.23 | PRCP | 0.22 | PRCP | 0.21 | PRCP | 0.32 | PRCP | 0.36 | PRCP | 0.36 |
| 3 | HSG | 0.09 | HSG | 0.12 | HSG | 0.13 | HSG | 0.12 | HSG | 0.08 | SAND | 0.12 | SILT | 0.05 |
| 4 | SILT | 0.07 | LUN | 0.07 | LUN | 0.07 | LUN | 0.07 | LUN | 0.05 | TMX | 0.02 | TMX | 0.02 |
| 5 | LUN | 0.05 | SILT | 0.07 | SILT | 0.05 | SILT | 0.07 | SAND | 0.04 | ORHI | 0.01 | SAND | 0.01 |
| … | … | … | … | … | … | … | … | … | … | … | … | … | … | … |
| sum | | 1.29 | | 1.30 | | 1.25 | | 1.27 | | 1.34 | | 1.27 | | 1.27 |

**Text S1.**

The following equations describe the calculation of the cover and management factor, the soil erodibility factor and the topographic factor of each water erosion equation:

The **cover and management factor** is calculated the same way for each equation:

$C = FRSD * FBIO * FRUF$     (1)

where FRSD is the crop residue factor, FBIO is the growing biomass factor and FRUF is the soil random roughness factor, which are calculated with the following equations (Wang et al., 2011; Williams et al., 2012):

$FRSD = exp(-P23 * CVRS)$    (2)

$FBIO = 1 - \frac{STL}{(STL+exp(SCRP1(23)-SCRP2(23)*STL))} * exp(-P26 * CPHT)$   (3)

$FRUF = exp(-0.026 * (RR - 6.1))$    (4)

where P23 is an exponential coefficient ranging from 0.01-0.5, CVRS is the amount of above ground crop residue [t ha$^{-1}$], STL is the amount of standing live biomass of the crop [t ha$^{-1}$], SCRP1(23) and SCRP2(23) are coefficients defining an S-shaped growth curve used to estimate the fraction of the ground covered by the plant as a function of the Leaf Area Index, P26 is an exponential coefficient ranging from 0.01-0.2, CPHT is the crop height [m] and RR is the soil surface random roughness [mm].

The **soil erodibility factor** is calculated the same way for the USLE, AOF, MUSLE, MUST and MUSS

equation using a function of sand, silt, clay and organic carbon contents in the soil:

$K = X1 * X2 * X3 * X4$       (5)

$X1 = 0.2 + 0.3 * exp(-0.0256 * SAND * (1 - 0.01 * SILT))$  6)

$X2 = \left(\frac{SILT}{CLAY+SILT}\right)^{0.3}$    (7)

$X3 = \frac{1-0.25*OC}{OC+exp(3.718-2.947*OC)}, IF\ OC \leq 5$      (8)

$X3 = 0.75, IF\ OC > 5$       (9)

$X4 = \frac{1-0.7*SN1}{SN1+exp(-5.509+22.899*SN1)}$         (10)

$SN1 = 1 - 0.01 * SAND$       (11)

Where SAND, SILT, CLAY, and OC are the sand, silt, clay, and organic carbon contents of the soil in %. For the RUSLE and RUSLE2 method soil erodibility is calculated without the organic carbon contents of the soil using the following equation:

$KR = 9.811 * \left(0.0034 + 0.0405 * exp\left(-0.5 * \left(\frac{Log10(DG)+1.659}{0.7101}\right)^2\right)\right)$   (12)

$DG = exp\,(SUM)$     (13)

$SUM = \dfrac{SAND*0.0247 - SILT*3.65 - CLAY*6.908}{100}$     (14)

The **topographic factor** is calculated the same way for the USLE, AOF, MUSLE, MUST and MUSS equation using a function of slope length and slope steepness:

$LS = \left(\dfrac{SLPL}{22.127}\right)^{XM} * (SLP * (65.41 * SLP + 4.56) + 0.065)$     (15)

$XM = 0.3 * \dfrac{SLP}{SLP + exp\,(-1.47 - 61.09*SLP)} + 0.2$     (16)

Where SLPL is the slope length in m, SLP is the land surface slope in m/m and XM is an exponent dependent upon slope. The topographic factor for the RUSLE method is calculated using a function of slope length and slope steepness as well:

$LSR = RSF * RLF$     (17)

$RSF = 10.8 * SLP + 0.03,\, IF\ SLPL > 4.57\ \&\ SLP < 0.09$     (18)

$RSF = 16.8 * SLP - 0.5,\, IF\ SLPL > 4.57\ \&\ SLP > 0.09$ (19)

$RSF = X1,\, IF\ SLPL < 4.57$     (20)

$X1 = 3 * SLP^{0.8} + 0.56$     (21)

$RLF = \dfrac{SLPL}{22.127}^{RXM}$     (22)

$RXM = \dfrac{B}{1+B}$     (23)

$B = \dfrac{SLP}{0.0896*X1}$     (24)

Where SLPL is slope length in m and SLP is land surface slope in m/m. The slope steepness factor RSF is adjusted for different slope steepness and slope length thresholds based on experimental data (Renard et al.,

1997). The slope length factor RLF includes an exponent RXM, which is a function of the ratio B of rill erosion caused by flow and interill erosion caused by raindrop impact (USDA-ARC, 2013). B reflects how steepness affects rill erosion differently than it does interrill erosion. Rill erosion is assumed to vary linearly with steepness. The topographic factor for the RUSLE2 method is calculated the same way than for the RUSLE

equation if the transport capacity determined by a function of flow rate and slope steepness exceeds sediment load. When sediment load exceeds transport capacity RUSLE2 computes deposition. Interrill erosion is assumed to occur even when RUSLE2 computes deposition, which can be calculated without a distance term as detachment is solely caused by impacting raindrops (USDA-ARC, 2013). Therefore, the slope length factor is not considered in the RUSLE2 equation when deposition occurs.

[Figure]

**Figure S1**. Main climate zones using the updated Koeppen-Geiger climate classification (Peel et al., 2007).

[Figure]

**Figure S2**. Grid cells with irrigated and rainfed wheat and maize cultivation around the year 2000 (Portmann et al., 2010).

[Figure]

**Figure S3.** Distribution of average water erosion values from 1980 – 2010 simulated with the baseline scenario and weighted for each simulation grid. The dashed vertical line illustrates the median of the distribution, which represents global median water erosion of 6 t ha⁻¹ a⁻¹. Average water erosion at each grid and the global average water erosion of 19 t ha⁻¹ a⁻¹ has been calculated as a weighted average based on the distribution of irrigated and rainfed maize and wheat acreage (Portmann et al., 2010).

[Figure]

**Figure S3**. World regions classified using the United Nations geoscheme  (UN, 1999) with minor
modifications : Melanesia has been added to South eastern Asia and the Caribbean has been added
to Central America.

[Figure]

method
● Radioisotopic   ● Runoff and sediment collection   ● Volumetric survey

**Figure S4.** Locations of water erosion field data from cropland where coordinates were recorded (n=554).

[Figure]

[Figure]

**Figure** S5. Distribution of erosion (t ha[-1]) values measured in agricultural fields using runoff and sediment
collection  (n = 188, Mean =  21 t ha[-1]; Median = 4 t ha[-1]) [137]Cs method (n =
315, Mean =  24 t ha[-1]; Median =  18 t ha[-1]) and volumetric surveys (n = 103, Mean = 2 t ha[-1]; Median =
0.1 t ha[-1]).

[Figure]

**Figure S6**. (a) Distribution of slope steepness (%) records for measured erosion values (n = 606; Mean =
16 %; Median = 11 %). (b) Distribution of annual precipitation (mm) records for measured erosion
values (n = 606; Mean = 879 mm; Median = 774 mm). (c) Distribution of recorded measurement
periods for soil loss experiments excluding radioisotopic methods (n = 95; Mean = 15 a; Median = 1 a).

[Figure]

**Figure S7S7**. (a) Number of measured water erosion records (n=473606) per country (n=39376). (b) Methods
used to measure water erosion in agricultural fields (n=473). (c) Soil texture recorded at sites of water erosion
measurement (n = 473).

[Figure]

method
● experimental plot  ● radioisotopic

**Figure S8.** Locations of water erosion field data from cropland where coordinates were recorded (n=468).

[Figure]

**Figure** S8. Median deviation (MD) in t ha⁻¹ between measured and simulated water erosion using the baseline
scenario with different water erosion equations. Measured and simulated medians were calculated for different
slope and precipitation classes.

[Figure]

**Figure S3**. Distribution of average water erosion values from 1980 – 2010 simulated with the baseline scenario
and weighted for each simulation grid. The dashed vertical line illustrates the median of the distribution, which
represents global median water erosion of 6 t ha$^{-1}$ a$^{-1}$. Average water erosion at each grid and the global average
water erosion of 19 t ha$^{-1}$ a$^{-1}$ has been calculated as a weighted average based on the distribution of irrigated and
rainfed maize and wheat acreage (Portmann et al., 2010).

[Figure]

**Figure S10**. Simulated years left until the whole soil profile is eroded under permanent maize and wheat cultivation. Calculated as a ratio of the sedimentary deposit thickness [m] (Pelletier et al., 2016) and the eroded soil depth per year (water erosion [t ha$^{-1}$ a$^{-1}$] x bulk density [g m$^{-3}$]).

[Figure]

**Figure S11**. Comparison of slope inputs and simulated water erosion outputs between the cropland distribution scenario using the most common slopes and the cropland distribution scenario using the flattest terrain available in Italy. (a, b) distribution of the cropland share (Portmann et al., 2010) per slope class. (c, d) distribution of grid cells per slope class. (e) Simulated water erosion for Italy using both cropland distribution scenarios. Midlines visualise median values, boxes include values from the 25th to the 75th percentiles and whiskers bracket values between the 10th and the 90th percentiles.

| Dominant slope class | Lower value (%) | Upper value (%) | Mid value (%) | Slope length (m) | Field size (ha) |
|---|---|---|---|---|---|
| 1 | 0 | 0.5 | 0.25 | 200 | 10 |
| 2 | 0.5 | 2 | 1.25 | 200 | 10 |
| 3 | 2 | 5 | 3.5 | 200 | 10 |
| 4 | 5 | 8 | 6.5 | 200 | 10 |
| 5 | 8 | 16 | 12 | 100 | 5 |
| 6 | 16 | 30 | 18 | 75 | 5 |
| 7 | 30 | 45 | 35.5 | 50 | 1 |
| 8 | 45 | 100 | 60 | 20 | 1 |

**Table S1**. A set of rules for field size and slope length estimation for each dominant slope class. The
area/dominant slope class was assigned to each grid cell from a global slope and terrain dataset (Fisher et al.,
2007) providing 3 arc-sec spatial resolution distributions of nine slope gradient classes: 0–0.5%, 0.5–2%, 2–5%,
5–8%, 8–16%, 16–30%, 30–45%, and > 45% interpreted from SRTM elevation data (CGIAR-CSI, 2006). Mid-
interval value of the dominant slope class was used as an input for EPIC.

**Table S2**. Input parameters for the sensitivity analysis of the water erosion equations. Random values assigned
to each input parameter in the sensitivity analysis are defined by a range of discrete values or a triangular
distribution defined by the values given in the table.

**Table S3**. First- and total-order sensitivity indices (SI) ranking for 30 input parameters for each water erosion
equation.

**Table S4**. Spearman coefficients explaining the positive or negative correlation between the first- and total-
order sensitivity indices of the input parameters from each equation and the amount of annual rainfall at a
location.

**Table S5**. Measured water erosion values collected from 1013 studies. The reference list of each study is
available at TWCarr-si02.docx.

**Uncertainties, sensitivities and robustness of simulated water erosion in an EPIC based global-gridded crop model**
By T. W. Carr et al.

**Reply to Anonymous Referee #1**

Referee comment is printed in purple and is addressed below

Despite the decades of research, modelling spatially distributed phenomena such as soil water erosion, still represents very challenging job. The biggest challenge lies in comparing modelled and measured soil erosion rates, especially in case of global scaled models such EPIC. The main added values of presented paper are: 1. Evaluation of simulation results against field data and uncertainty assessment. Uncertainty assessment represents a crucial factor, when communicating the results of simulation and further incorporation of such models into for instance global circulation models. 2. The authors pointed out several obstacles, which prevent further development of soil erosion modelling research such as lack of uniform and reliable data on water erosion rates, lack of datasets providing distributed data on topography, soil, climate, land use and field management at the field scale. Supplementary TableS5 contains the list of soil erosion measurement records, it would be good to add an information about the scale of measurement (plot, field, : : :) The article is of high scientific value and I recommend it for the publication without any substantial revision.

Thank you for your positive comment. We added a column to the supplementary TableS5 specifying the scale of the erosion measurement as "Hillslope", "Plot" and "Field". A more detailed discussion about the field data scale can be found in the reply to referee #2.

**Uncertainties, sensitivities and robustness of simulated water erosion in an EPIC based global-gridded crop model**
By T. W. Carr et al.

**Reply to Anonymous Referee #2**

Dear reviewer,

Before we address each of your comments, we briefly clarify the main incentive of this study. Large-scale indicators about global-scale phenomena are needed to inform all major environmental and agricultural policies such as the European Union's Common Agricultural Policy (CAP), the United Nations Sustainable Development Goals (SDGs), the United Nations Convention to Combat Desertification (UNCCD) and the Intergovernmental Science-Policy Platform on Biodiversity and Ecosystem Services (IPBES). Water erosion will not be considered in any of these major environmental and agricultural policy programs without large-scale assessments (Alewell et al., 2019). Global-gridded crop models have the capacity to develop large-scale indicators and to inform about agricultural productivity in a transparent and consistent way across large areas (Mueller et al., 2017). This paper aims to address the gaps in the literature of the links between water erosion and crop cultivation in various large-scale and global impact assessments, as accurately as is currently feasible given data availability. Studies on large-scale and global climate change impacts in the agricultural sector lack representation of water erosion impacts on crops (Balkovič et al., 2018), studies on global terrestrial carbon fluxes do not account for carbon runoff from cropland through soil erosion (Chappell et al., 2016), and studies assessing large-scale and global market impacts of soil erosion rely on simple linear estimates of water erosion impacts on crop production (Panagos et al., 2018; Sartori et al., 2019). It is important, though, to understand the limitations of such assessments so that they can be improved in the future, and that is why we systematically test a number of approaches in our paper.

The model used in this study has been confirmed as a reliable tool for global crop yield projections and stands out against comparable global models due to its detailed representation of soil processes including water erosion and the impacts of tillage on soil properties. Therefore, a global-gridded EPIC model has the potential to deliver much needed indicators about relationships between erosion and crop productivity on large and global scales. This paper has the objective to support the ongoing development of large-scale and global model applications by analysing the robustness of average long-term water erosion estimates generated with global-gridded crop models. In other words, we do not attempt to reproduce soil loss rates measured in single fields but to analyse the robustness of large-scale water erosion estimates based on global-gridded crop model outputs. Moreover, we focus on the necessary improvements needed to account for water erosion in global models by analysing and discussing its robustness across global agro-environmental conditions, the importance of global input data on field management and the uncertainty resulting from different erosion equations.

Most of the criticism by the referee is directed against the lack of field data, model calibration, and general criticism on RUSLE-based erosion models and the $^{137}$Cs-method used for erosion measurements. Each point of criticism is addressed in the following (Referee comments are printed in purple and our replies are listed below):

*The EPIC model has been used to look at climate change impacts on crop yields and erosion rates e.g. Favis-Mortlock et al. (1991) and to model 7000 years of erosion under changing climates and land uses for a single field (Favis-Mortlock et al., 1997). It is stressed that EPIC needs calibration in order*

*to give reasonable results. This is the very firm conclusion of the GCTE erosion model testing exercise (Favis-Mortlock, 1998; Boardman and Favis-Mortlock, 1998).*

The first point of criticism is on the need to calibrate the EPIC model for reasonable results. The EPIC-IIASA model uses state-of-the-art global crop management and agro-environmental input data and has been positively evaluated for representing national average yields and inter-annual yield variability globally (Balkovič et al., 2014). It was used in several studies and its outputs have been compared to regional yield statistics and other global crop and land use models as a part of ISI-MIP and GGCMI model inter-comparison initiatives (Mueller et al., 2017). Global crop models are not calibrated to reproduce crop yields at field scale but rather to represent the crop yield patterns across regions and countries to address research questions that cannot be addressed through field scale studies. Following the same paradigm, we aim to analyse the robustness of EPIC to represent regional differences and regional spatial patterns in water erosion estimates rather than accurately reproduce erosion events occurring in the past in response to individual rain events of rainy seasons. We are aware that our approach would be inappropriate for the latter case. We are also aware that a proper model calibration is always needed to meet experimental data obtained from the field, foremost for a complex process like soil loss due to water erosion. At the same time, sound calibration for a wide range of global environments and crop management practices would require enormous capacity and work force while still facing a high degree of subjectivity in experimental data (e.g. Panagos et al. 2016). Given the current lack of consistent field measurements representing all global environments, it is not possible to produce plausible global erosion estimates using only bottom-up, field-scale modelling.

To further clarify the intention of this study, we add a reference clarifying the usefulness of large-scale models to line 59: "*Moreover, improving the representation of water erosion in large-scale models is urgently needed to inform major environmental and agricultural policy programs such as the European Union's Common Agricultural Policy (CAP), the United Nations Sustainable Development Goals (SDGs), the United Nations Convention to Combat Desertification (UNCCD) and the Intergovernmental Science-Policy Platform on Biodiversity and Ecosystem Services (IPBES) (Alewell et al., 2019).*"

We will further clarify the focus of this paper in the introduction, line 91: "*The overall aim of this study is: (i) to analyse the robustness of water erosion estimates in all global agro-environmental regions simulated with an EPIC-based global-gridded crop model; and, (ii) to discuss the main drivers affecting the robustness and the uncertainty of simulated water erosion rates on a global scale.*"

We will modify the last conclusion on line 606: "*The overlap of simulated and measured water erosion values in most environments used to produce wheat and maize underlines the robustness of an EPIC-based GGCM to simulate the differences in water erosion rates of major global crop production regions*"

*The authors claim that they are evaluating their results against field-scale measures (lines 84 and 95). This is not the case: they use 137Cs and erosion plot data (line 219). Erosion plot data cannot be up-scaled to field scale: it is useful for relative assessments e.g Cerdan et al. (2010). Extrapolation from12 plots in central Belgium to give an average rate of erosion for Europe is a well-known (?) example of misuse of experimental plot data (Boardman, 1998): the current paper is heading in that direction!*

The referee criticises the use of our erosion plot data. We do not extrapolate plot data to represent water erosion on continental scales as in the Belgium-example mentioned by the referee. As explained on line 235-239, we aggregate field data into groups with similar slope classes and precipitation classes and compare the values to model outputs with the same slope and precipitation classes. Thereby, we analyse the robustness of model outputs for different environments characterised by the most important parameters affecting water erosion on a global scale. Slope and precipitation are the most sensitive parameters influencing model outputs and are the parameters found previously to be most critical for the robustness of RUSLE-based models. We chose this method to illustrate the varying robustness of our model outputs around the globe and we identified regions where the model performance is not sufficient, which is communicated in the discussion (line 367-370) and the conclusion (line 472 – 473).

More generally, plot and field scale are not exactly defined. These two categories can be overlapping as large plots can have similar slope length as fields. Plots of 20 m are most common, and if they are equipped by multislot divisors, typing buckets they can be up to 100 m long, or even several hundred meters if equipped by Coshocton wheels. We know about 100 m erosion plots from Slovakia and Austria or 30 m plots in Zimbabwe. Fields can also have slope length about 20-100 meters, especially if they are on slopes, contour oriented or when they belong to small family farms in developing countries. We have seen 30-50 m long slopes in Uganda, Madagascar, or Slovakia.

Regarding the real scale of our data, most of them (both 137Cs and erosion plots) represent slopes of 10-100 m so they are at the margin of plot and field scale. Therefore, in the paper, we will name the range of spatial scales of the field data on line 239 (we increased the field data sample as explained below). “*We compared our simulated water erosion rates with 606 soil erosion measurements on arable land from 36 countries representing plot and field scale. Most of the selected erosion rates are based on the $^{137}$Cs method. In addition, data from erosion plots and volumetric measurements of rills collected by Auerswald et al. (2009), Benaud et al. (2020) and García-Ruiz et al. (2015) were used. In total, 315 records were derived by the 137Cs method, 188 records from runoff plots, and 103 records from volumetric measurements of rills.* “

The term “*field-scale measurements*” on line 95 will be deleted as this sentence will be replaced with the sentence stating the overall aim of this study presented above.

The term “*field measurements*” on line 106 is not related to scale.

*RUSLE is an unvalidated model and its problems and poor performance are reviewed in Evans and Boardman (2016a and b). For a review of the general problems of using erosion models see Favis-Mortlock et al. (2001): in Harmon and Doe (ed) book.*

Further criticism is focused on the validity of the RUSLE method in general, which is one out of seven similar erosion models included in our EPIC study. The RUSLE methodology is based on more than 10,000 plot-years of experiments and has been applied in more than 100 countries with varying robustness (Alewell et al., 2019; Renard et al., 1997; Wischmeier & Smith, 1978). As already disputed above, the limited availability of global input and experimental data requires simple erosion models for global studies. Therefore, RUSLE has been chosen by most studies focusing on global erosion (Borrelli et al., 2017; Doetterl et al., 2012; van Oost et al., 2007) and it is unfortunate that the referee disagrees with the approach adopted by most of the scientific community. We agree that the varying robustness of RUSLE-based methods around the world need to be considered, which is one of the main foci of this paper and has been addressed in the introduction and in the discussion. Furthermore, we present the varying estimates of different water erosion equations and thereby demonstrate the uncertainty of relying on erosion estimates from a single model. The references used for criticising the RUSLE method by the referee promote field studies as alternatives to erosion models. However, this is not feasible for a range of research applications focusing on analysing scenarios at large and global scales (Alewell et al., 2019; Panagos et al., 2016), which is also a major purpose of global gridded crop models. Instead, focusing on the improvement of the application of simple erosion models such as RUSLE-based models as intended with this paper supports the ongoing development of global erosion impact assessments (Naipal et al., 2015).

*137Cs has been seriously criticised recently (Parsons and Foster, 2011). The technique should not be used without dealing with these limitations. This problem is ignored in the paper.*

The objections of Parsons and Foster (2011) were discussed by Mabit et al. (2013) published as a direct answer to Parsons and Foster in the same journal. One of the most important confirmation of the usefulness of the 137Cs method given by Mabit et al (2013) are the positive results of a comparison between erosion values obtained with various measurement methods and erosion values obtained by 137Cs method: "*Several studies at various scales (from plot to watershed scales) have been conducted to compare the 137Cs based erosion rates obtained with direct erosion measurement approaches such as erosion plots and catchment sediment yields (e.g. Schuller et al., 2003; Porto et al., 2003; Porto et al., 2004; Stankoviansky et al., 2006;Mabit et al., 2009; Parsons et al., 2010; Ceaglio et al., 2012; Porto and Walling, 2012a, 2012b). In northern California, rates of erosion from accumulated pond sediment and soil lost from hillsides assessed through 137Cs agreed well (i.e. O'Farrell et al., 2007). In Italy, the reliability of the mass balance model at slope and catchment scale was verified and confirmed by comparing the basin net soil erosion value obtained by 137Cs measurements against the mean annual value of sediment yield measured at the basin outlet (i.e. Di Stefano et al., 2005). For three small catchments located in Southern Italy, measurements of sediment output validated 137Cs theoretical conversion models to estimate soil redistribution rates (i.e. Porto et al., 2004)*".

Another paper by Parsons (2019) also criticises erosion plots, modelling (several models) and volumetric measurements of rills. As our paper is focused on global erosion modelling, we did not include the extensive literature on the advantages and disadvantages of each erosion measurement method. A comprehensive discussion of the objections against the 137Cs method and other measurement methods would but too extensive for our paper. However, we will address Parsons and Fosters objections in this response:

   a. Regional and local heterogeneity of 137Cs fallout and its local redistribution by vegetation, infiltration, bioturbation, etc.: this is well known fact but it is considered in the methodology and the solution is the selection of reference sites in immediate vicinity of study sites and using statistical criteria (variation coefficient) criteria for microvariability of 137Cs. The microvariability of 137Cs is similar to most other soil properties and the potential error is similar or smaller than for other erosion measuring methods (as it will be demonstrated further). There is set a limit for variation coefficient which reference site should not exceed.
   b. Mobility of 137Cs: the references presented by Mabit et al. (2013) clearly demonstrate that 137Cs is strongly bind to colloids (with details on particular clay minerals and organic matter) and its mobility (washing by runoff during the deposition, leaching and plant uptake are negligible (representing less than one percentage to very few percentages of 137Cs fallout). The ideas about mobility, leaching and plant uptake of 137Cs presented by Parsons and Foster are based mainly on laboratory experiments with Caesium which do not represent its real behaviour in nature because the laboratory conditions are artificial, and the used doses of Caesium are too high. Parsons and Foster admit that in their paper. If these ideas would be correct, we would frequently see leached Caesium in deeper part of soil profile or the whole inventories at sites undisturbed by erosion would be depleted by plant uptake. But this is never the case in natural soil.

c. Selective removal and sorting of the particles: This is well known by all authors using the 137Cs method and it is mentioned in all handbooks and conversion procedure has a parameter for that. But it is true that this factor is difficult to quantify. But similar weak points are common in most methods of erosion measurements as will be demonstrated below.

d. Conversion models: Indeed, procedures to calculate soil loss require some parameters which are not always available. The accuracy of calculation depends on quality of input data and is different in individual studies. But this is the case for all erosion measurements and all methods as it will be demonstrated below.

e. Sample preparation: the problems with coarse fraction, dry or wet sieving can occur in specific soils, especially when having porous coarse fraction and concretions containing clay or organic matter so that the coarse fraction has electrical charges. This of course should be understood by staff who is expected to have basic pedological education. It is true that not all these details are mentioned in every methodological guidance, but they are discussed by some authors.

f. Gammaspectroscopy: Criticism of gammaspectroscopy is not relevant at all. Indeed, there are possible different geometries of detectors and detectors have to be calibrated. But this is task for laboratory staff. Cs-method obviously require staff having background in nuclear physics. Each laboratory method whether physical or chemical require staff with appropriate qualification.

Finally, we accept that we could have mentioned the limitations of the field data we collected and referred to some of the existing literature. We will mention the main limitations of the field data we collected on line 552:

*"The 137Cs method was criticised by Parsons and Foster (2013), who questioned assumptions about the 137Cs behaviour in the environment (variability of the 137Cs input by wet fallout, its microspatial variability at reference sites, its possible mobility in certain soils, the 137Cs uptake by plants and other aspects of 137Cs behaviour in soil). To confront the criticism against the 137Cs method, Mabit et al. (2013) discussed all objections raised by Parsons and Foster (2013) and confirmed its accuracy by listing several studies, in which 137Cs based erosion rates are compared with erosion rates derived from direct measurements. The 137Cs method is based on a set of presumptions which should be met to produce useful results and thus careful interpretation of the obtained results is needed (Fulajtar et al., 2017; Mabit et al., 2014; Zapata, 2002).*

*Similarly, erosion rates obtained by volumetric measurements require careful interpretation as they are exposed to various potential sources of errors and do not account for interill erosion. Although the latter can be neglected under certain circumstances, studies from Europe and semiarid areas of the USA have reported that interill erosion contributed significantly to the amount of soil eroded in fields (Boardman and Evans, 2020; Parsons, 2019). Further, measuring the lengths and cross-sections of rills during field surveys or on terrestrial and aerial photos can be very subjective (Panagos et al., 2016). Different approaches used to detect and measure rills in fields can cause variability in calculated erosion volumes up to a factor of two (Boardman and Evans, 2020; Casali et al., 2006; Watson and Evans, 1991). In order to obtain soil erosion rates in weight units, soil volumes need to be converted using the soil bulk density, which is often based on estimates (Evans and Brazier, 2005).*

*The shortcomings of erosion plot measurements were discussed by several authors (Auerswald et al., 2009; Brazier, 2004; Evans, 1995, 2002; Loughran et al., 1988). Erosion plots have various sizes and shapes (few meters to few hundreds of meters) and various approaches of sediment recording are used (total collection, multislot divisors, tipping buckets, Coshocton wheels), which all involve significant uncertainties. Although some long-term plot experiments exist, many plot measurements fail to cover the whole year erosion cycle (Auerswald et al., 2009). Often, they have to be removed during land*

*management operations such as seeding, ploughing, or they are too expensive and labour demanding."*

*It is simply not true to claim that there is a limited availability of field data and lack of long term measurements (lines 68-69). There are extensive data sets from Switzerland, north Germany and the UK. These could be used to validate the results of erosion models: see Boardman and Evans (2020: PPG) for a review of these methods of assessment of erosion at a field scale.*

The need for slope and precipitation information accompanying erosion measurements in our evaluation method narrows down suitable field datasets, as meta data such as slope steepness is often not available in published datasets. Moreover, when we refer to a lack of field data, we are talking about the global scale and especially the imbalance in data availability among world regions. We have addressed the uneven distribution of global field data in the introduction and the skewed focus of field data on the United States and Europe in the discussion. Also, the difficulty of gathering field data from a very heterogenous mix of measurement methods is comprehensively addressed in the discussion. We attempted to gather field data from as many continents as possible to represent different global environments. Insufficient field data representing all global regions and the lack of sufficient metadata in available datasets to further improve erosion modelling on large and global scale is an important conclusion of this paper.

We will further clarify these issues in the discussion after line 514

*"A variety of factors influencing water erosion such as climate, field topography, soil properties and field management need to be considered when modelling water erosion but are often not reported in available field measurements (García-Ruiz et al., 2015). This hampers a direct comparison between simulated and observed water erosion values. We demonstrated the varying match between measured and simulated water erosion using different tillage and cover crop scenarios. Metadata on field management often only provides the crop cultivated and therefore the conditions under which erosion was measured in the field are not known sufficiently to evaluate erosion values simulated under different field management scenarios. Similarly, information on field topography and soil properties is often not provided with recorded field measurements and thus their use is limited in an evaluation of water erosion estimates simulated in different global environments. Moreover, most data are concentrated in the United States, West Europe and the West Mediterranean (García-Ruiz et al., 2015). In summary, there is a lack of field data representing all needed regions, situations and scenarios (Alewell et al., 2019)."*

We increased the field data sample to 606 records using publicly available datasets from Germany and the UK provided by Auerswald et al. (2009) and Benaud et al. (2020).

We add a description of the additional field data after line 256:

*"Bounded plots are the most commonly used method of erosion measurements. They were introduced in the USA in the 1920s (Hudson, 1993) and were used for the development of USLE and WEPP models (Brazier, 2004). Eroded soil material can be quantified with erosion plots in different ways (total collection of sediment, fractioned collection of sediments using multislott divisors, measurement of discharge and sediment concentration by tipping buckets and Coshocton wheels). The overview of this method is provided by Cerdan et al. (2010); Hudson (1993); Mutchler et al. (1994); De Ploey and Gabriels (1980) and Zachar (1982).*

*The volumetric measurements of rill erosion are used since approximately the 1940s in the USA (Kaiser, 1978 in Evans, 2013) and the 1950s in Europe (Lobotka, 1955), usually at field scale (Boardman, 1990,*

*2003; Boardman and Evans, 2020; Brazier, 2004; Evans, 2002, 2013; Herweg, 1988; Zachar, 1982). The volume of erosion rills is derived from their lengths and profile cross-section areas, which are measured in field or from terrestrial and aerial photos (Evans, 1986, 1988; Watson and Evans, 1991)."*

An attempt to further increase the field data sample using the source provided by the referee ( Boardman & Evans (2020)) was not possible as only aggregated values are provided for the large datasets listed in Boardman & Evans (2020).

Some more general remarks: Although huge effort was spent by the erosion community to generate an enormous number of data, there is a serious lack of useful data to evaluate large-scale models. It was very challenging to gather the amount of field data used in this study for the following reasons:

1. As we are working with USLE-derived models, we were looking for data from certain spatial extend only. The USLE was developed at short slopes and represents sheet erosion and initial stages of rill erosion. Therefore, we preferred data from ca 10-100 m long slopes. This is the case for field data derived with the 137Cs method and erosion plots. Initially, we decided against using data from long slopes (several hundred of meters), which is usually the case of volumetric measurements of rill erosion and hydrological measurements in small watersheds.

2. At the beginning of this project we tried to focus only on data derived by 137Cs method as different methods represent different erosion processes and are subject to different systematic errors which are presented in the following:

   **Erosion plots with total collection of sediment** have problems to collect great volumes of sediment in case of extreme rain events (the sediment may exceed the capacity of containers) and it can be difficult to determine the weight of sediment (when it is wet, the whole volume cannot be carried to the laboratory for drying so the quantity of soil in the collected mud is just estimated by taking a sample of the mud to measure concentration).

   **Multislot divisors, tipping buckets and Coshocton wheels** have many technical problems (multislot divisors may split the sediment unequally if they are not fixed exactly horizontally, the tipping buckets and Coshocton wheels loose part of the sediment when they are tipping or when the stream is strong water is splashing out, if the stream is weak the soil material is sedimenting immediately in tipping buckets and the sample is not representative, data loggers can break, etc.).

   Studies with replicated plots showed great variability for replicas. Nearing et al. (1999) report from almost 800 replicated plot pairs/year data a coefficient of variation ranging between 14% and 150%. Variability was decreasing with increasing soil loss. The rates of 10 tons/ha had coefficient of variability of ca 40%.

   **Geodetical method (erosion pins)** has much bigger error than erosion plots because it has poor resolution. If one mm of soil is removed, the change of surface is hardly seen. But this represents already 10 tons of soil per hectare. On arable land the geodetic method has problems to distinguish between erosion and compaction.

   **Rill and gully volumetric measurements** (underline: preferred method in the reference provided by the referee (Boardman & Evans (2020))) neglects sheet erosion completely. The recalculation of obtained volumetric data to weight is problematic because of the limited information on soil bulk density and its vertical and horizontal variability. This is problematic as we need data in t/ha to compare with models. Usually it is not indicated whether rill measurement represent the whole year or only the vegetation season, whether they involve rills from snow melt or not, etc.

The measurement of rill volumes itself is a source of huge error. Authors who use this method know this (for example Evans, 2013, stated: "Mollenhauer notes (2002: 4) 'The measurement of lengths and cross-section areas of linear forms can be extremely error ridden'; and quotes from Ruttiman and Prasuhn's (1990) work in Switzerland that the 'total error for soil loss volume can amount to between 20 and 40%' (Mollenhauer, 2002: 4)" and further "The level of accuracy of field-based estimates depends on the amount of time spent in measuring/estimating the number, lengths and dimensions of rills and gullies and assessing volumes of depositional features such as fans. The larger an area surveyed inevitably means that cruder estimates of eroded amount will be made, for example, numbers, lengths and cross-sectional areas of channels will all have to be estimated rather than measured").

Measurements have very few traverses (sometimes only 4, Boardman, 2003), which is a huge source of error. Boardman even says that sometimes one traverse is enough (Boardman, 2003: "The number of traverses is clearly subject to the time and resources available and also the purpose of the survey as to how much detail and accuracy is required. In many situations, it is reasonable to undertake one traverse across the mid-point of the eroded slope and estimate total erosion based on the mean rill length."). Measurement based on one traverse compared to measurements based on 4 traverse revealed errors from -18,1% to +48,7%.

The method neglects interrill erosion which is an important portion of the whole erosion process. Estimates of the importance of interrill erosion differ significantly for different conditions (negligible amount of 0.3 m3/ha/y provided by Evans (1990, in Boardman 2003), few % of total erosion: 5-11% (Morgan et al., 1987, in Evans, 2013), up to few tens of %: 25% (Prasuhn, 2011) and 10-30% (Zachar, 1982)).

Parsons (2019) emphasize that the volumetric measurements of rills severely underestimate overall erosion because rills also involve large quantity of material which was delivered by sheet erosion to rills and further transported by rills. These proportions can be 40% for rill erosion and 60% for interrill. Luk et al. (1993, in Parsons, (2019)) determined the portion of rill erosion ranging between 0 (when only sheet erosion develops) and 56%. Our own experience from Central Europe with more heavy rainfalls from own unpublished measurements is that at steep slopes (ca 8-12 degrees) it can be in some years 10-40 tons per ha. Govers and Poesen found that the proportion between rill and interrill erosion can change significantly with time and according to changes in physical properties of top layer and deeper layers either proportion of rill erosion can rise or the proportion of interrill erosion can rise. In their case study the proportion of interrill erosion was decreasing with time from 46 to 22%, but other authors found opposite trends. Therefore, estimating interrill erosion from rill erosion using fixed ratios is wrong. They also find, that interrill erosion has higher proportion on short slopes than on long slopes.

Sometimes the presented rill measurements are not real measurements but just very rough and brief estimation. The rills and their lengths are estimated from photos, where smaller rills might be difficult to detect (for example Boardman, 2003: "Ground-level photographs of rills and gullies may be used as a record of length and size; subsequent analysis shows them to be a reliable means of estimating soil losses (Watson and Evans, 1991)" or Evans, 2013: "In this review paper, 'direct' assessment of water erosion is taken to mean the mapping of erosion and deposition as evident in fields (Figures 1–9) or on ground or aerial photographs and then when possible estimating eroded volumes based on lengths and cross-sections of rills and gullies and areas and depths of deposition (Evans, 1988; Herweg, 1996; Stocking & Murnaghan, 2001)." Watson and Evans (1991) estimate the sizes of rills on photos comparing it with the widths of crop rows and height of crop and thickness of colluvial fans they estimate according to their colour. They compared the results of volumetric measurements in field and on photos (12 photos) and found ratios from 0.67 to 2.12.

**Hydrological measurements** in elementary watershed do not represent erosion only from agricultural land but also bank erosion and road erosion, and both these can be significant.

**Sampling of suspended sediment** is not well representative, and samplers or data loggers can break. The range of discharge in small catchments is so huge that it makes instrumentation of hydrological profiles difficult.

For these various possible sources of errors, we did not want to mix up different methods. The optimal case would have been to only use field data derived from one method, but to increase the amount of data we decided to take 137Cs data and some selected erosion plots (to further increase our sample we included suitable data from volumetric surveys).

3. Large amount of existing data is not accessible for various reasons:
   a. Many older publications are in national languages
   b. Many older publications are not on internet
   c. Many measurements were published in grey literature, local conference proceedings, national acta of scientific institutions, unpublished reports, etc.
   d. Many published data are hardly interpretable because metadata are lacking (slope lengths, or inclinations or crop cover, period of measurement is not recorded, geographical position of the sites is not recorded, many measurements were running only during vegetation period of studied crop so they do not represent annual erosion but just few months, etc.).
   e. International journals do not have interest to publish usual case studies which present raw data. To get paper published the authors need to present some special objectives to follow some special goals or developing methodological innovations. Therefore, also many new data sets cannot be found online.
   f. Even if paper is published, journals have usually size limitations. To save space the primary data are not presented, only the results of interpretation, statistical processing, etc. are there. In publications using 137Cs it is more common to find primary data than in studies using other methods.

Please consider that all methods have a lot of weak points, methodological shortcomings and sources of errors, uncertainties and variability and there is lack of reliable comparisons and comprehensive assessment of all methods which would be widely accepted by the whole erosion community. There are different schools and groups of researchers who use predominantly one method. One group uses 137Cs method, other groups prefer erosion plots, the next one focuses on elementary watersheds using hydrological methods based on discharge and sediment concentration sampling (or combination of plots and watersheds) and other group focuses on volumetric measurements of rills and gullies. Some researchers using certain method are very critical about other methods, but they are very tolerant regarding the shortcomings of their favoured method. Although each method has advantages and limitations and each group has success and achievements as well as challenges and failures, it is normal if individual researchers or teams prefer one particular approach. But they should respect also other approaches.

Our collected data set represents a reasonable compromise to achieve the objectives of this study. It is far beyond the capacities of the team and the objectives of this study to collect all existing erosion data. Such task would require years lasting international project with participation of research teams from most countries, so that each team would be able to revise data sources in his country and provide summary of data including those published in national language and unpublished reports. Most existing data have limited accuracy and representativeness, but we cannot wait until dense coverage of perfect data will cover the Globe. Erosion is running, agriculture is in troubles and we should proceed under existing circumstances and using available tools.

*The method of deriving a common slope within an area of 9x56 km is not clear and seems rather dubious (line 122). Averaging slope from a large cell (eg. 1km2) is a common failing of erosion modelling exercises (e.g. Evans and Brazier, 2005).*

The most common slope is determined by a slope class covering the largest area in each simulation grid. Slope classes are taken from a global terrain slope database (IIASA/FAO, 2012) and are based on a high-resolution 90 m SRTM digital elevation model. We assume that the slope class representing the largest area in each grid cell is most likely covered by the largest area of cropland. This builds on the idea that a spatially extensive and diverse landscape can be represented by a single "representative field" characterized by the prevailing combination of topography and soil condition found in the landscape. This method is designed to represent differences in large-scale global crop production with an emphasis on the most important global crop production regions.

We clarify the concept of the representative field on line 134: "*Each grid cell is represented by a single field characterized by the combination of topography and soil conditions prevailing in this landscape unit. Each representative field has a defined slope length (20 – 200 m) and field size (1 - 10 ha) based on a set of rules for different slope classes (Table S1). The slope of each representative field is determined by the slope class covering the largest area in each grid cell (Table S1). Slope classes are taken from a global terrain slope database (IIASA/FAO, 2012) and are based on a high-resolution 90 m SRTM digital elevation model.*"

A detailed discussion about the uncertainty in slope input data has been added in response to comments by the third referee (see response to referee #3).

*One conclusion seems to be that wheat erodes are a greater average rate ((19t/ha) than maize (6t/ha) (line 244): this is contrary to all field evidence that I am aware of.*

The criticised conclusion on falsely higher erosion rates in wheat fields compared to maize fields is based on a misunderstanding. The values represent a global average value (19 t/ha) and a median value (6 t/ha) of water erosion rates for both maize and wheat fields combined.

To avoid confusion, we will focus only on median values in the revised version as median values are less influenced by the skewed distribution of erosion values. In the discussion line 418 we mention both mean and median value to illustrate the skewed distribution of erosion rates due to very high values simulated on steep slopes.

We will present global median water erosion for both maize (7 t/ha) and wheat (5 t/ha) fields on lines 26, 280 and 349, and will delete average values.

Global average water erosion values simulated under different management scenarios and different water erosion equations are deleted on line 466 – 468 and 497 to focus only on median values.

**Reply to Anonymous Referee #3**

Dear reviewer, we appreciate your thoughtful comments and have responded to each in the following (Referee comments are printed in purple and are each addressed below).

*This manuscript describes a study to characterize global soil erosion rates on cropland using the exploration of a large parameter space of driver data and erosion models. Starting with global information on climate, soils, agricultural practices, and field properties, the authors calculate representative erosion rates. In a series of experiments, they show the sensitivity of the model to driving inputs and parameter assumptions. They evaluate the model results against a large dataset of observed soil erosion data. The authors conclude that the model results are very sensitive to assumptions about management strategy, and the accuracy of the model is limited by a lack of field observations for calibration and evaluation.*

*In general this manuscript is well written and simple enough to understand. However some key information is lacking in the main text of the manuscript, and some of the results seem rather suspicious, possibly because of artifacts in the input data. In particular, the headline numbers for global soil erosion, and the mapped model output, appear to be strongly influenced by erosion in mountainous areas, where in reality land use for agriculture may be much more limited than the model assumes. These issues need to be addressed in a revision before the manuscript is ready for publication.*

Our experience (not only from modelling, but also from field work, excursions and own observations and measurements) shows that in some mountainous areas the erosion rates reaches very high levels. But to some extent you are right and we considered your objection. Below, when answering other comments on this issue we will try to demonstrate in detail the situation in mountains. We will provide also some photos.

*Looking at the model results in Figure 2a, what stands out immediately is that very high rates of erosion are plotted in many regions of the world where I would not be sure that there is any significant amount of agriculture, including the central highlands of Borneo, the Himalaya, eastern Madagascar, South Korea, and parts of the Alps. These are indeed high-rainfall/high slope regions and in some of the area agriculture is practiced. But where there is cropland, it almost certainly must be limited to valley bottoms or other low-slope areas, or only performed with substantial investment in erosion mitigation measures, such as terracing.*

*Digging deep into the manuscript supplementary materials, I discovered that the actual crop distribution data used in this study (5') comes from Portmann et al. (2010). This citation, and explanation for how the crop areas were determined, must be moved to the main body of the text. It appears that Portmann et al. (2010) do not use slope or any other topographic characteristics in determining the spatial allocation of cropland in their crop area maps. Furthermore, 5' resolution is probably too coarse even in the authors' own admission to accurately determine appropriate mean slope classes for their soil erosion calculations.*

The reference to Portmann et al. (2010) is listed in Table 3 in the main text, which summarises the field management assumption and aggregation of model outputs.

We extend the reference to table 3 on line 212. *"Table 3 summarises the field management assumptions of the baseline scenario used to aggregate erosion rates in each grid cell and region."*

We agree, that in some mountainous areas the erosion rates obtained by modelling do not represent typical rates. However, it is not the case for all mountains. The values are overestimated most probably in mountains of temperate areas such as Europe (Alps), Korea and Japan and also in some tropical rice grooving regions on tropical Monsoon Asia (such as Borneo). However, in many mountainous areas in tropics the land is cultivated, and maize and wheat are grown even in very steep slopes and often without any soil conservation practices or conservation practices are used with insufficient efficiency. We attached a collection of photos demonstrating these phenomena (**photos are attached in the pdf-file including the direct response to referee #3**). In the tropics, agriculture is very active in mountains especially for four reasons:

1. While in temperate areas such as Europe the temperature is limiting factor of agriculture and in higher altitudes (several hundred meters above the see level) it is too cold for most crops and if agriculture exist there, it is mainly grazing. In contrary, in tropics the temperature is sufficient also in high mountains.

2. The limiting factor in tropics is drought. Therefore, mountains having more rainfall and less heat are popular agricultural area. Very good example is Uganda, where the whole flat central part of the country is too dry and it is used only for grazing, while steep mountains in west and east peripheries of the country are intensively cultivated. The same is in Madagascar and in whole Latin America where mountains are more agriculturally exploited than lowlands.

3. In tropics the weathering crusts are very thick and so the loose materials constitute thick soils. Tropical soils are poor in organic matter so the difference between topsoil and subsoil is not very big. These soils can be exposed to extreme erosion rates much longer than thin soils of temperate areas, so farmers do not feel the decrease of fertility and production potential. It is decreasing slowly, and they do not realize the impact of erosion.

4. In developing countries, a great portion of land is still under hand management of small family subsistence farming. These farmers can cultivate steep slopes easier than farmers who use heavy machines.

Some examples from our own field work, where the extreme exploitation of steep slopes exists are following:

1. In Latin America, especially in mountains with volcanic rocks are cultivate even in extreme slopes and here maize is dominant crop (see photos from El Salvador).
2. Mountainous area of Sub-Saharan Africa: Extremely steep slopes are cultivated, many crops with low conserving efficiency are grown (such as cassava and other sweet potatoes, beans, maize, etc.), there are terraces, but these are not really horizontal but inclined so they reduce erosion partially but not much (see photos from Uganda and Madagascar)
3. South and East Asia: In this region the major crop is rice which is usually grown on paddy field with flood irrigation. Therefore, the large mountainous areas are well terraced and well protected from erosion. However, there are also very large mountainous areas which are not terraced at all and steep slopes are cultivated. They are growing various crops there such as dryland rice, tea, fruit trees, sugar cane, etc. For example, in large part of southern China the rice terraces are only in valley bottoms occupying minor areas and all steep slopes occupying majority areas are used to grow sugar cane without any soil conservation.

An explanation and further discussion of the resolution of slope input and cropland distribution data follows below

*These limitations mean that the headline numbers for erosion (e.g., lines 25-26 of the abstract), and much of the results are likely to be skewed by calculations that are not realistic, because they are biased by high-slope/high-precipitation areas where in reality, agriculture is not practiced at all, or only in very limited and specialized forms, e.g., agroforestry, and perennial crops such as tea and orchards. This source of uncertainty needs to be addressed more thoroughly and the methods presented more transparently before this manuscript is suitable for publication.*

*Finally, it would be interesting if the authors performed a "reality check" on their erosion numbers. With some of the extreme values that they calculated, could agriculture be sustainable at all? How long would it take before most soil is completely eroded away?*

Unfortunately, in many mountainous areas especially in tropics conventional agriculture with very bad management is practiced and it has huge negative impact on land. We demonstrated it by photos. In many mountainous areas agriculture is not sustainable at all. But unfortunately, despite of that in many areas the destructive land management is going on and poor farmers are destroying the land completely. We even do not know how many cases like this occur. We have examples also from Slovakia, mainly historical but also recent. There are known cases that some slopes were cultivated just 10-20 years and then one extreme storm event removed all soil and the field was abandoned. In tropics frequently happens that when the field is destroyed, it is abandoned for 5-10 years being fallow and then cultivated again, but there are many cases also when slope is cultivated for 5-7 years and destroyed once for ever. See attached photos.

There are indications of this problems also in literature, for example Montgomery (2007) calculated mean erosion rate under conventional agriculture (n= 448) to be over 3.9 mm (what is ca 60 tons per hectare). Of course, such agriculture is not sustainable at all. He concluded: "A direct implication of the imbalance between agricultural soil loss and erosion under both native vegetation and geologic time is that, given time, continued soil loss will become a critical problem for global agricultural production under conventional upland farming practices." Catastrophic effects of agriculture on land discuss Pimentel and Burges (2013). They argue that annually 10 million ha of cropland is abandoned due to deteriorating production potential caused by erosion. Further, they estimate that recently the world cropland covers 1.5 billion ha but since the beginning agriculture people abandoned 2 billion ha of crop land. So, more soil is already destroyed and abandoned then what is still used.

*Lines 122-123*

*What is the justification for choosing the "most common slope"? At the very least, wouldn't it make more sense to choose the lowest slope class in each 5' gridcell? At least until all of the area in the slope class is filled by agricultural land use before moving to the next steeper class? If not, the authors' choice of modal slope class should be justified with citations.*

The most common slope is determined by the slope class covering the largest area in each simulation grid. Slope classes are taken from a global terrain slope database (IIASA/FAO, 2012) and are based on a high-resolution 90 m SRTM digital elevation model. We assume that the slope class representing the largest area in each grid cell is most likely covered by the largest area of cropland. This builds on the idea that a spatially extensive and diverse landscape can be represented by a single "representative field" characterized by the prevailing combination of topography and soil condition found in the landscape. This method is designed to represent differences in large-scale global crop production with an emphasis on the most important global crop production regions.

We clarify the concept of the representative field on line 134:

*"Each grid cell is represented by a single field characterized by the combination of topography and soil conditions prevailing in this landscape unit. Each representative field has a defined slope length (20 – 200 m) and field size (1 - 10 ha) based on a set of rules for different slope classes (Table S1). The slope of each representative field is determined by the slope class covering the largest area in each grid cell (Table S1). Slope classes are taken from a global terrain slope database (IIASA/FAO, 2012) and are based on a high-resolution 90 m SRTM digital elevation model."*

Slope input data is an important uncertainty for simulating global water erosion estimates as we cannot identify cultivated slopes on a global scale. This will be discussed in the revised paper by addressing: (i) the simulation of extreme values on steep slopes including the proposed 'reality check'; (ii) the consequences of using the most common slope; and, (iii) an ideal scenario, where cultivation is limited to the flattest terrain available. We used Italy as a test case for a comparison between water erosion rates simulated under the proposed ideal slope scenario and the slope scenario based on the most common slope. We chose Italy because it has large maize and wheat cultivation areas, which are located on both flat terrain in the north and in hilly regions in the south, and thus the country represents a diverse landscape with a wide range of possible water erosion rates. We add a comprehensive discussion addressing each slope related issues after line 426:

[revised manuscript text omitted]

The following two figures addressed in the discussion will be added to the supplementary information.

[Figure]

*Figure 2:Simulated years left until the whole soil profile is eroded under permanent maize and wheat cultivation. Calculated as a ratio of the sedimentary deposit thickness [m]* (Pelletier et al., 2016) *and the eroded soil depth per year (water erosion [t ha$^{-1}$ a$^{-1}$] x bulk density [g m$^{-3}$]).*

[Figure]

*Figure 3: Comparison of slope inputs and simulated water erosion outputs between the cropland distribution scenario using the most common slopes and the cropland distribution scenario using the flattest terrain available in Italy. (a, b) distribution of the cropland share (Portmann et al., 2010) per slope class. (c, d) distribution of grid cells per slope class. (e) Simulated water erosion for Italy using both cropland distribution scenarios. Midlines visualise median values, boxes include values from the 25th to the 75th percentiles and whiskers bracket values between the 10th and the 90th percentiles.*

*Line 184-187*

*Again, where is the evidence that steeper slopes are actually cultivated, and on what basis are these P-factors selected? Were the parameters selected using empirical evidence, or a citation?*

As described above, we cannot be certain that steep slopes are cultivated, but we assume that steep slopes are only cultivated with conservation techniques to reduce high water erosion values. The P-values for contouring and terracing are within the range of the values reported by Morgan (2005).

We modify line 205 accordingly: *"To account for erosion control measures reducing high water erosion on steep slopes, we use a conservation P-factor of 0.5 on slopes steeper than 16 %, and a P-factor of 0.15 on slopes steeper than 30 % to simulate contouring and terracing based on the range of P-factors presented by Morgan (2005)."*

*Lines 377-379*

*"...a significant share of the estimated soil removal of 7 Gt a-1 originates from small wheat and maize fields on steep slopes with strong annual precipitation". So here the authors admit that the global numbers are skewed by extreme levels of simulated erosion. But more evidence that these fields actually exist needs to be provided.*

See comments and discussion above about the high uncertainty in cropland distribution on steep slopes.

==We delete the total soil loss value from the abstract, line 27, as it is significantly influenced by extreme water erosion rates from mountainous regions. But we will keep the value in the discussion, where we address the uncertainty of the global soil loss value.==

*Lines 391-392*
*How were the countries where "conservation agriculture… is likely" selected? What*
*evidence is there for this?*

We selected countries where conservation agriculture is most likely based on the share of conservation agriculture reported by AQUASTAT (2005-2014). The criteria is presented on lines 184-185 and table 3.

==We will refer to AQUASTAT (FAO, 2016) on line 477.==

*Lines 423-425*
*That "…many older measurements are poorly accessible as they are not available*
*online" seems to be a bit of a weak argument for not collecting more measurements*
*on soil erosion. Can the authors elaborate a bit more in what kind of data are out there*
*and precisely what it would take to utilize them for future studies?*

Indeed, this is true. There was huge amount of erosion measurements at experimental plots in many countries. For example, in USA first measurements started in 1915 and when Wischmeier and Smith were developing their equation they had about 10000 erosion plot/year data and this was in 1970s. These data are archived by USDA but they are not directly accessible on internet. When in Germany Schwertmann was verifying USLE for Germany he used about 2500 plot/year data, but they are not available on internet and only small part was published and it was in German. We know situation mainly in central Europe. In Slovakia we have about 50 plot/year data published in Slovak language, we know about erosion plot measurements in Hungary, a lot of old data are in Czechia (starting with measurements by Maran in 1950ies, and Poland (starting by Gerlach in 1950ties), significant data set is in Austria, whole book about long term measurements (ca 20 years) was published in Croatia in Croatian language, there was extensive measurement programme in Yugoslavia (Serbia, Gavrilovic, Djorovic,). A lot of data exists in China, Japan, UK and Russia. In Africa we know about data from Uganda and Zimbabwe, most data from Francophone Africa are in French, from Latin America in Spanish, etc. There are five major reasons why most data are not available:

1. Many older publications are in national languages
2. Many older publications are not on internet
3. Many measurements were published in grey literature, local conference proceedings, national acta of scientific institutions, unpublished reports, etc.
4. Many published data are hardly interpretable because metadata are lacking (slope lengths, or inclinations or crop cover, period of measurement is not recorded, geographical position of the sites is not recorded, many measurements were running only during vegetation period of studied crop so they do not represent annual erosion, etc.).
5. International journals do not have interest to publish usual case studies which present raw data. To get paper published the authors need to present some special objectives to follow some special goals or developing methodological innovations. Therefore, also many new data sets cannot be found online.

6. Even if paper is published, journals have usually size limitations. To save space the primary data are not presented, only the results of interpretation, statistical processing, etc. are there.

The collected data set represents a reasonable compromise to achieve the objectives of this study. It is far beyond the capacities of the team and the objectives of this study to collect all existing erosion data. Such task would require 3-5 years lasting international project with participation of research teams from most countries, so that each team would be able to revise data sources in his country and provide summary of data including those published in national language and unpublished reports.

We added a more comprehensive discussion to available field data to the response addressing the second referee.

We increased the field data sample from 473 to 606 following a comment by the second referee. We will change the values and the presentation of the field data accordingly on lines 239-244, 349 – 351.

We will further clarify the field data needs in the discussion line 510:

"*The main reasons for the low availability of suitable data to evaluate simulated water erosion rates are twofold: (i) erosion monitoring is expensive, time consuming and labour demanding; and, (ii) primary data and metadata of measurement sites accompanying final results are often not available and many older measurements are poorly accessible as they are not available online (Benaud et al., 2020). A variety of factors influencing water erosion such as climate, field topography, soil properties and field management need to be considered when modelling water erosion but are often not reported in available field measurements (García-Ruiz et al., 2015). This hampers a direct comparison between simulated and observed water erosion values. We demonstrated the varying match between measured and simulated water erosion using different tillage and cover crop scenarios. Metadata on field management often only provides the crop cultivated and therefore the conditions under which erosion was measured in the field are not known sufficiently to evaluate erosion values simulated under different field management scenarios. Similarly, information on field topography and soil properties is often not provided with recorded field measurements and thus their use is limited in an evaluation of water erosion estimates simulated in different global environments. Moreover, most data are concentrated in the United States, West Europe and the West Mediterranean (García-Ruiz et al., 2015). In summary, there is a lack of field data representing all needed regions, situations and scenarios (Alewell et al., 2019).*"

We additionally mention "*the lack of sufficient metadata accompanying erosion measurements*" on line 270.

We add two sentences comparing the high variability within field data with the deviation between simulated values and measured values based on the evaluation results to Line 371: "*Outside locations combining steep slopes and strong precipitation, median deviation between simulated and measured data is lower than the variability within the field data.*" Line 579: "*In most environments relevant for maize and wheat cultivation the deviation between simulated and measured water erosion values is lower than the variability within the field data.*"

*Lines 466-467*
*Yes, it seems clear that increased resolution would be important. Several datasets are*
*already available however, including 100m agricultural cover fraction data (Buchhorn*
*et al., 2019) and 90m topography from a range of different datasets, such as MERITHydro*
*(Yamazaki et al., 2019). Global climate and soils data are available at at least*
*1km resolution and could be downscaled (Fick Hijmans, 2017; Hengl et al., 2017).*
*Some more explanation as to why the authors were limited to 5' and more concrete*

*recommendations for future research would be valuable.*

We rely on the existing data infrastructure of the EPIC-IIASA model, which has been constructed and evaluated for large-scale and global crop yield projections. The EPIC-IIASA model uses state-of-the-art global crop management and agro-environmental input data and has been positively evaluated for representing national average yields and inter-annual yield variability globally (Balkovič et al., 2014). It was used in several studies and its outputs have been compared to regional yield statistics and other global crop and land use models as a part of ISI-MIP and GGCMI model inter-comparison initiatives (Mueller et al., 2017). One of the main goals of this study is to analyse if EPIC-IIASA can account for relationships between water erosion and crop cultivation. Therefore, we rely on the existing model setup and data infrastructure of EPIC-IIASA, which has been confirmed as a reliable model to simulate daily crop growth on a global scale. The Input data for EPIC-IIASA originally available at different scales were aggregated at 5' resolution grid. In EPIC-IIASA, each simulation grid is represented by a representative field (1 to 10 hectares, depending on the prevailing slope category) while the field topography was calculated as a "dominant combination" from the high-resolution 90-m SRTM digital elevation model. Given the large uncertainty in land cover maps (Fritz et al., 2015; Lesiv et al., 2019), EPIC-IIASA does not provide an explicit link between land cover category, such as cropland, and the dominant fields. Instead, an area share of each land cover category per simulation grid is provided based on the GLC2000 land cover map with 1x1 km spatial resolution.

As mentioned above a discussion on the uncertainty in cropland distribution and slope input data will be added, as well as an explanation of the concept of the "representative field".

We will further clarify the focus of this paper in the introduction, line 91: "*The overall aim of this study is: (i) to analyse the robustness of water erosion estimates in all global agro-environmental regions simulated with an EPIC-based global-gridded crop model; and, (ii) to discuss the main drivers affecting the robustness and the uncertainty of simulated water erosion rates on a global scale.*"

We further highlight the model's weakness in conclusion line 600: "*Using existing field data, we were able to identify specific environmental characteristics for which we have lower confidence in the modelled erosion rates. These are mainly found in the tropics and mountainous regions due to the high sensitivity of simulated water erosion to slope steepness and precipitation strength, and the complexity of agricultural systems in mountainous regions.*"

*Lines 473-474*
*As the high erosion "areas represent only a small fraction of global cropland for wheat and maize", why not show median values as the headline results instead of means?*

We agree that the presentation of both mean and median values can be confusing. In the revised version we will focus only on median values. However, in the discussion, line 391 we mention both mean and median value to demonstrate the skewed distribution of erosion rates due to extreme values simulated on steep slopes.

We present global median water erosion in maize (7 t/ha) and wheat (5 t/ha) fields on line 26, 280, 349 and delete average values.

Global average water erosion values simulated under different management scenarios and different water erosion equations will be deleted on line 470 – 472 and 497 to focus only on median values.

We add a row to Table 3 clarifying that the median is used to aggregate water erosion values simulated under all management scenarios for grid cells and regions.

*Lines 684-689; Figure 2*
*I would like to see the map and statistics separated out into two, one figure set each*
*for maize and wheat. As the growing areas are different and only partially overlapping,*
*it would be very helpful to see these individually in the main body of the manuscript.*

We will present two maps for maize and wheat respectively in the revised paper (see figure below).

We will group bars by crop in the revised version (see figure below).

The explanation that water erosion is presented as a weighted average from maize and wheat fields will be deleted in table 3.

We include a note in the figure label explaining that pixel cells in figures do not indicate cropland sizes. *"Each pixel cell illustrates the median relative water erosion of one representative field. The extent of cropland areas is not considered in pixel cell size. "*

[Figure]

*Figure 4: Soil loss due to water erosion in maize (a) and wheat (b) fields simulated with the baseline scenario. Each pixel cell illustrates the median relative water erosion of one representative field. The extent of cropland areas is not considered in pixel cell size. The bars in the bottom plot (c) illustrate median soil removal for major world regions simulated under maize and wheat cultivation. The lines and whiskers illustrate 25th and 75th percentile values. The classification of world regions is illustrated in Fig. S4. Due to the large gap between aggregated values, all values in the bottom plot have been log-transformed to facilitate the visual comparison.*

*Lines 706-709; Figure 7*

*I am quite suspicious that there is any substantial amount agriculture at all in the purple areas marked on the map, e.g., Borneo highlands, northern Laos, Himalayan front, western Madagascar, Korea, Japan. If there is, agriculture must be limited to valley*

*bottoms that are not detected at 5' resolution or done with extreme terracing.*

Each pixel in the maps illustrates the median erosion rate of one representative field. The pixel cells in each map do not indicate total cropland area. In other words, most of the pixel in mountainous regions represent a very small cultivated area. Table 3 lists details on how erosion rates in each pixel are aggregated.

*Lines 691-693; Figures 3 and 4*
*Would also be useful to see how much uncertainty is caused by the assumption of what slopes are being farmed, e.g., always lowest slopes first, mean slope, median slope, etc.*

We will address slope uncertainty in an extended discussion (see comments above)

---

## Author Response (AR2)

London, 22$^{nd}$ September 2020

Dear Fortunat Joos (editor),

We included our responses to reviewer #3 below. We did not include a separate reply to reviewer #1, as the reviewer accepted the manuscript without changes.

Please find our revised manuscript including tracked changes below our response to reviewer #3. We added four additional photos to the supplementary information. Additionally, we added the four photos of the main text to the supplementary information to make them available in higher detail.

Thank you for your support in publishing our research at Biogeosciences.

Kind regards,

Tony Carr on behalf of the authors

**Uncertainties, sensitivities and robustness of simulated water erosion in an EPIC based global-gridded crop model**
By T. W. Carr et al.

**2nd Reply to Anonymous Referee #3**

Dear reviewer,

Thank you again for your very helpful suggestions to improve our paper.

We added four additional photos to the supplementary information in addition to the photos of the main text. We appreciate your interest in the photos. However, we did not include all available photos to save a selection for additional publications. We added the location and slope information to each photo as detailed as possible, where it was recorded. But some photos were taken on excursions, in which we did not have the opportunity to record details about coordinates and topography.

Following your last suggestion, we added two sentences to line 548 addressing the need to collate soil erosion records for future research and referred to a recent publication focusing on this issue:

*"Moreover, the accessibility of field data should be improved as raw data is often not published or needs to be collected from numerous publications, grey literature and conference proceedings to obtain the large amount of data necessary for regional or global erosion studies. Therefore, we support recent efforts to collate erosion measurements and metadata from existing studies (Benaud et al., 2020) as we believe that the availability of field data through a single platform will greatly benefit future modelling studies and the understanding of soil erosion at all scales."*

The reviewer's comment is copied in purple below.

*Review of*

*Uncertainties, sensitivities and robustness of simulated water erosion in an EPIC-based global-gridded crop model*

*By T. W. Carr et al.*

*In general the authors have done an admirable job responding to the reviewers' comments and the current manuscript is greatly improved compared to the previous version. I would be pleased to recommend this manuscript for publication following minor revision.*

*I particularly liked the authors additional sensitivity analyses concerning assumptions about which slopes are farmed. The test study for Italy showed clearly that if only low-slope landscapes are cultivated, very little soil erosion occurs. This result may form a basis for policymaking in some cases.*

*Responding to on my comments from the previous version, Figure 8 is a fantastic addition to the manuscript. I particularly appreciated the authors sharing of personal observations of (frequently unsustainable) cultivation practices in tropical mountains, and the illustration with photos. The figure showing years until total depletion of the soil profile (Fig. S10) is also very helpful.*

*I realize figure 8 is only a fraction of the photos you must have, and I liked seeing all of the photos that were provided in the response to reviewers document. I would be pleased if all of these additional photos could be provided in a supplementary materials document to accompany the final paper.*

*Furthermore, in figure 8, please provide approximate geographic coordinates and approximate elevation and slope for the sites shown in each of the four pictures. Exact numbers are not necessary, but it would be very helpful for future studies to examine these regions in more detail, e.g., using satellite and updated terrain data.*

*I realize that collation of a large database of soil erosion measurements would be beyond the scope of the current manuscript, and I fully appreciate the authors explanation of the challenges and effort that would be required to do this. I found the six points listed in the response to reviewers helpful information that could be used to form a basis for future international research initiatives. These may already be well known in the community. I would anyway request that the authors include a 1-2 sentence summary of the need and requirements for synthesizing soil erosion records in the final version of the paper, as this would be a very helpful citation, e.g., for anyone preparing a proposal to initiate or coordinate a project on the topic.*

[revised manuscript text omitted]

Text S1.

The following equations describe the calculation of the cover and management factor, the soil erodibility factor and the topographic factor of each water erosion equation:

The **cover and management factor** is calculated the same way for each equation:

$C = FRSD * FBIO * FRUF$      (1)

where FRSD is the crop residue factor, FBIO is the growing biomass factor and FRUF is the soil random roughness factor, which are calculated with the following equations (Wang et al., 2011; Williams et al., 2012):

$FRSD = exp(-P23 * CVRS)$    (2)

$FBIO = 1 - \frac{STL}{(STL + exp(SCRP1(23) - SCRP2(23)*STL))} * exp(-P26 * CPHT)$   (3)

$FRUF = exp(-0.026 * (RR - 6.1))$    (4)

where P23 is an exponential coefficient ranging from 0.01-0.5, CVRS is the amount of above ground crop residue [t ha$^{-1}$], STL is the amount of standing live biomass of the crop [t ha$^{-1}$], SCRP1(23) and SCRP2(23) are coefficients defining an S-shaped growth curve used to estimate the fraction of the ground covered by the plant as a function of the Leaf Area Index, P26 is an exponential coefficient ranging from 0.01-0.2, CPHT is the crop height [m] and RR is the soil surface random roughness [mm].

The **soil erodibility factor** is calculated the same way for the USLE, AOF, MUSLE, MUST and MUSS

equation using a function of sand, silt, clay and organic carbon contents in the soil:

$K = X1 * X2 * X3 * X4$          (5)

$X1 = 0.2 + 0.3 * exp(-0.0256 * SAND * (1 - 0.01 * SILT))$   6)

$X2 = \left(\frac{SILT}{CLAY + SILT}\right)^{0.3}$     (7)

$X3 = \frac{1 - 0.25 * OC}{OC + exp(3.718 - 2.947 * OC)}, IF\ OC \leq 5$        (8)

$X3 = 0.75, IF\ OC > 5$        (9)

$X4 = \frac{1 - 0.7 * SN1}{SN1 + exp(-5.509 + 22.899 * SN1)}$           (10)

$SN1 = 1 - 0.01 * SAND$        (11)

Where SAND, SILT, CLAY, and OC are the sand, silt, clay, and organic carbon contents of the soil in %. For the RUSLE and RUSLE2 method soil erodibility is calculated without the organic carbon contents of the soil using the following equation:

$KR = 9.811 * \left(0.0034 + 0.0405 * exp\left(-0.5 * \left(\frac{Log10(DG) + 1.659}{0.7101}\right)^2\right)\right)$   (12)

$DG = exp\,(SUM)$          (13)

$SUM = \frac{SAND*0.0247 - SILT*3.65 - CLAY*6.908}{100}$          (14)

The **topographic factor** is calculated the same way for the USLE, AOF, MUSLE, MUST and MUSS equation using a function of slope length and slope steepness:

$LS = \left(\frac{SLPL}{22.127}\right)^{XM} * (SLP * (65.41 * SLP + 4.56) + 0.065)$          (15)

$XM = 0.3 * \frac{SLP}{SLP + exp\,(-1.47 - 61.09*SLP)} + 0.2$          (16)

Where SLPL is the slope length in m, SLP is the land surface slope in m/m and XM is an exponent dependent upon slope. The topographic factor for the RUSLE method is calculated using a function of slope length and slope steepness as well:

$LSR = RSF * RLF$          (17)

$RSF = 10.8 * SLP + 0.03, IF\ SLPL > 4.57\ \&\ SLP < 0.09$          (18)

$RSF = 16.8 * SLP - 0.5, IF\ SLPL > 4.57\ \&\ SLP > 0.09$  (19)

$RSF = X1, IF\ SLPL < 4.57$          (20)

$X1 = 3 * SLP^{0.8} + 0.56$          (21)

$RLF = \frac{SLPL}{22.127}^{RXM}$          (22)

$RXM = \frac{B}{1+B}$      (23)

$B = \frac{SLP}{0.0896*X1}$     (24)

Where SLPL is slope length in m and SLP is land surface slope in m/m. The slope steepness factor RSF is adjusted for different slope steepness and slope length thresholds based on experimental data (Renard et al.,

1997). The slope length factor RLF includes an exponent RXM, which is a function of the ratio B of rill erosion caused by flow and interill erosion caused by raindrop impact (USDA-ARC, 2013). B reflects how steepness affects rill erosion differently than it does interill erosion. Rill erosion is assumed to vary linearly with steepness. The topographic factor for the RUSLE2 method is calculated the same way than for the RUSLE

equation if the transport capacity determined by a function of flow rate and slope steepness exceeds sediment load. When sediment load exceeds transport capacity RUSLE2 computes deposition. Interill erosion is assumed to occur even when RUSLE2 computes deposition, which can be calculated without a distance term as detachment is solely caused by impacting raindrops (USDA-ARC, 2013). Therefore, the slope length factor is not considered in the RUSLE2 equation when deposition occurs.

[Figure]

Figure S1: Main climate zones using the updated Koeppen-Geiger climate classification (Peel et al., 2007).

[Figure]

Figure S2: Grid cells with irrigated and rainfed wheat and maize cultivation around the year 2000 (Portmann et al., 2010).

[Figure]

Figure S3: World regions classified using the United Nations geoscheme (UN, 1999) with minor modifications:
Melanesia has been added to Southeastern Asia and the Caribbean has been added to Central America.

[Figure]

method
● Radioisotopic   ● Runoff and sediment collection   ● Volumetric survey

Figure S4: Locations of water erosion field data from cropland where coordinates were recorded (n=554).

[Figure]

Figure S5: Distribution of erosion values (t ha$^{-1}$) measured in agricultural fields using $^{137}$Cs method (n = 315,
Mean = 24 t ha$^{-1}$; Median = 18 t ha$^{-1}$), runoff and sediment collection (n = 188, Mean = 21 t ha$^{-1}$; Median = 4 t
ha$^{-1}$) and volumetric surveys (n = 103, Mean = 2 t ha$^{-1}$; Median = 0.1 t ha$^{-1}$).

[Figure]

Figure S6: (a) Distribution of slope steepness (%) for measured erosion values (n = 606; Mean = 16 %; Median = 11 %). (b) Distribution of annual precipitation (mm) for measured erosion values (n = 606; Mean = 879 mm; Median = 774 mm). (c) Distribution of recorded measurement periods for soil loss experiments excluding radioisotopic methods (n = 95; Mean = 15 a; Median = 1 a).

Figure S7: Number of measured water erosion records (n=606) per country (n=36).

[Figure]

Figure S8: Median deviation (MD) in t ha[-1] between measured and simulated water erosion using the baseline
scenario with different water erosion equations. Measured and simulated medians were calculated for different
slope and precipitation classes.

[Figure]

Figure S9: Distribution of average water erosion values from 1980 – 2010 simulated with the baseline scenario
and weighted for each simulation grid. The dashed vertical line illustrates the median of the distribution, which
represents global median water erosion of 6 t ha$^{-1}$ a$^{-1}$. Average water erosion at each grid and the global average
water erosion of 19 t ha$^{-1}$ a$^{-1}$ has been calculated as a weighted average based on the distribution of irrigated and
rainfed maize and wheat acreage (Portmann et al., 2010).

[Figure]

Figure S10: Sugar cane cultivation on steep slopes in South China (Nanning, Guangxi Zhuang Autonomous
Region). The steepest slopes are already abandoned and reforested by eucalyptus trees.

[Figure]

Figure S11: Cultivated slopes and rice terraces in South China (Nanning, Guangxi Zhuang Autonomous
Region).

[Figure]

Figure S12: Maize cultivation on strongly eroded slopes (30 – 60 %) in South West Uganda (Kigwa, Kabale District).

[Figure]

Figure S13: Abandoned fields and maize cultivation on a steep slope (30 – 60 %) in South West Uganda (Kigwa, Kabale District).

[Figure]

Figure S14: Maize cultivation in South West Uganda (Kigwa, Kabale District).

[Figure]

Figure S15: Degraded and abandoned maize fields on steep slopes (20 – 60 %) in Northern El Salvador (San
Ignacio, Chalatenango Department).

[Figure]

Figure S16: Degraded land in Northern El Salvador (Monte Redondo, Chalatenango Province).

[Figure]

Figure S17: Gully erosion on arable land with slopes up to 20 – 30% in Slovakia (Figa, Rimavská Sobota
District).

[Figure]

Figure S18: Simulated years left until the whole soil profile is eroded under permanent maize and wheat
cultivation. Calculated as a ratio of the sedimentary deposit thickness [m] (Pelletier et al., 2016) and the eroded
soil depth per year (water erosion [t ha$^{-1}$ a$^{-1}$] x bulk density [g m$^{-3}$]).

[Figure]

Figure S19: Comparison of slope inputs and simulated water erosion outputs between the cropland distribution
scenario using the most common slopes and the cropland distribution scenario using the flattest terrain available
in Italy. (a, b) distribution of the cropland share (Portmann et al., 2010) per slope class. (c, d) distribution of grid
cells per slope class. (e) Simulated water erosion for Italy using both cropland distribution scenarios. Midlines
visualise median values, boxes include values from the 25th to the 75th percentiles and whiskers bracket values
between the 10th and the 90th percentiles.

| Dominant slope class | Lower value (%) | Upper value (%) | Mid value (%) | Slope length (m) | Field size (ha) |
|---|---|---|---|---|---|
| 1 | 0 | 0.5 | 0.25 | 200 | 10 |
| 2 | 0.5 | 2 | 1.25 | 200 | 10 |
| 3 | 2 | 5 | 3.5 | 200 | 10 |
| 4 | 5 | 8 | 6.5 | 200 | 10 |
| 5 | 8 | 16 | 12 | 100 | 5 |
| 6 | 16 | 30 | 18 | 75 | 5 |
| 7 | 30 | 45 | 35.5 | 50 | 1 |
| 8 | 45 | 100 | 60 | 20 | 1 |

Table S1. A set of rules for field size and slope length estimation for each dominant slope class. The
area/dominant slope class was assigned to each grid cell from a global slope and terrain dataset (Fisher et al.,
2007) providing 3 arc-sec spatial resolution distributions of nine slope gradient classes: 0–0.5%, 0.5–2%, 2–5%,
5–8%, 8–16%, 16–30%, 30–45%, and > 45% interpreted from SRTM elevation data (CGIAR-CSI, 2006). Mid-
interval value of the dominant slope class was used as an input for EPIC.

Table S2. Input parameters for the sensitivity analysis of the water erosion equations. Random values assigned
to each input parameter in the sensitivity analysis are defined by a range of discrete values or a triangular
distribution defined by the values given in the table.

Table S3. First- and total-order sensitivity indices (SI) ranking for 30 input parameters for each water erosion
equation.

Table S4. Spearman coefficients explaining the positive or negative correlation between the first- and total-order
sensitivity indices of the input parameters from each equation and the amount of annual rainfall at a location.

Table S5. Measured water erosion values collected from 113 studies. The reference list of each study is
available at TWCarr-si02.docx.